# Biosensor-integrated transposon mutagenesis reveals *rv0158* as a coordinator of redox homeostasis in *Mycobacterium tuberculosis*

**Somnath Shee[1,2], Reshma T Veetil[3], Karthikeyan Mohanraj[4], Mayashree Das[1,2], Nitish Malhotra[3], Devleena Bandopadhyay[5], Hussain Beig[1,2], Shalini Birua[1,2], Shreyas Niphadkar[6], Sathya Narayanan Nagarajan[1,2], Vikrant Kumar Sinha[5], Chandrani Thakur[7], Raju S Rajmani[2], Nagasuma Chandra[7], Sunil Laxman[6], Mahavir Singh[5], Areejit Samal[4], Aswin N Seshasayee[3], Amit Singh[1,2]***

[1]Department of Microbiology and Cell Biology, Indian Institute of Science Bangalore, Bangalore, India; [2]Centre for Infectious Disease Research, Indian Institute of Science Bangalore, Karnataka, India; [3]National Centre for Biological Sciences, Bangalore, India; [4]The Institute of Mathematical Sciences, A CI of Homi Bhabha National Institute, Chennai, India; [5]Molecular Biophysics Unit, Indian Institute of Science Bangalore, Bangalore, India; [6]Institute for Stem Cell Science and Regenerative Medicine, Bangalore, India; [7]Department of Biochemistry, Indian Institute of Science Bangalore, Bangalore, India

*For correspondence:
asingh@iisc.ac.in

**Competing interest:** The authors declare that no competing interests exist.

**Abstract** *Mycobacterium tuberculosis* (*Mtb*) is evolutionarily equipped to resist exogenous reactive oxygen species (ROS) but shows vulnerability to an increase in endogenous ROS (eROS). Since eROS is an unavoidable consequence of aerobic metabolism, understanding how *Mtb* manages eROS levels is essential yet needs to be characterized. By combining the Mrx1-roGFP2 redox biosensor with transposon mutagenesis, we identified 368 genes (redoxosome) responsible for maintaining homeostatic levels of eROS in *Mtb*. Integrating redoxosome with a global network of transcriptional regulators revealed a hypothetical protein (Rv0158) as a critical node managing eROS in *Mtb*. Disruption of *rv0158* (*rv0158* KO) impaired growth, redox balance, respiration, and metabolism of *Mtb* on glucose but not on fatty acids. Importantly, *rv0158* KO exhibited enhanced growth on propionate, and the Rv0158 protein directly binds to methylmalonyl-CoA, a key intermediate in propionate catabolism. Metabolite profiling, ChIP-Seq, and gene-expression analyses indicate that Rv0158 manages metabolic neutralization of propionate toxicity by regulating the methylcitrate cycle. Disruption of *rv0158* enhanced the sensitivity of *Mtb* to oxidative stress, nitric oxide, and anti-TB drugs. Lastly, *rv0158* KO showed poor survival in macrophages and persistence defect in mice. Our results suggest that Rv0158 is a metabolic integrator for carbon metabolism and redox balance in *Mtb*.

## Editor's evaluation

This manuscript is important as it describes the powerful combination of TnSeq approaches with a reporter for mycothiol redox potential to identify genes in *Mycobacterium tuberculosis* that are required to maintain the reducing environment of the cell. Through this study, the authors provide compelling data that Rv0158 functions as a critical regulator of bacterial responses to endogenous reactive oxygen species. This study will be important to investigators interested in bacterial physiology and how this can impact pathogenesis.

**Table 1.** List of redox-related proteins in *Mtb* which are either membrane bound or secreted out into periplasm/ extracellular space.

| | Protein/ small MW thiols | Rv ID | | References |
|---|---|---|---|---|
| 1 | KatG | Rv1908c | | *Braunstein et al., 2003*; *Tucci et al., 2020* |
| 2 | SodA | Rv3846 | | *Braunstein et al., 2003*, *Vargas-Romero et al., 2016*, *Tucci et al., 2020* |
| 3 | AhpC | Rv2428 | Antioxidant enzyme | *Nieto R et al., 2016*, *Tucci et al., 2020* |
| 4 | Tpx | Rv1932 | Probable thiol peroxidase | *Tucci et al., 2020* |
| 5 | TrxB2 | Rv3913 | Thioredoxin reductase | *Wong et al., 2018* |
| 6 | TrxC | Rv3914 | Thioredoxin | *Wong et al., 2018*, *Tucci et al., 2020* |
| 7 | | Rv0526 | Possible thioredoxin | *Ke et al., 2018* |
| 8 | ThiX | Rv0816 | Probable thioredoxin | *Ke et al., 2018* |
| 9 | Ergothioneine | | | *Sao Emani et al., 2013* |

## Introduction

*Mycobacterium tuberculosis* (*Mtb*) is the etiological agent of tuberculosis (TB), a disease that results in approximately 1.6 million deaths annually (*Global tuberculosis report, 2022*). Two major concerns: the emergence of drug-resistant strains and few anti-TB drugs in the pipeline, necessitate the discovery of novel drug targets. *Mtb* requires oxygen ($O_2$) for growth (*Wayne and Hayes, 1996*; *Boshoff and Barry, 2005*). Low $O_2$ levels, as encountered in hypoxic granulomas, trigger a transition from actively growing to a non-replicating persistent state in *Mtb* (*Boshoff and Barry, 2005*). While this metabolic adaptation is an important contributor toward tolerance against host-induced stressors and anti-TB drugs, it obstructs the completion of the *Mtb* life cycle (*Ernst, 2012*). Therefore, for successful progression and subsequent disease transmission, *Mtb* requires $O_2$. The immediate consequence of an aerobic lifestyle is the generation of endogenous reactive oxygen species (eROS) as metabolic by-products when $O_2$ adventitiously interacts with and extracts an electron from quinones, flavins, and transition metal centres distributed throughout the cell (*Imlay, 2013*; *Imlay, 2009*; *Imlay, 2003*). eROS can cause oxidative damage to DNA, proteins, and lipids, driving the bacteria to deploy detoxification and repair machinery to keep eROS-induced deleterious effects below a threshold level (*Imlay, 2009*). While multiple reports investigate this cellular requirement (*Ayer et al., 2012*; *Brynildsen et al., 2013*; *Radzinski et al., 2018*), an understanding of global pathways that keep *Mtb*- eROS levels in check, is absent.

The primary niche of *Mtb* is inside phagocytic cells, where the bacteria are exposed to superoxide ($O_2.^-$), nitric oxide (NO), their reaction product peroxynitrite (ONOO.), and hypochlorite stress (*Ehrt and Schnappinger, 2009*; *Philips and Ernst, 2012*). In response to host-induced redox stress, *Mtb* deploys antioxidant enzymes catalase-peroxidase (KatG), superoxide dismutase (Fe-SodA), thioredoxin reductase (TrxB2), alkyl hydroperoxide reductase (AhpC), and redox buffers- thioredoxin (trxC), and ergothioneine (*Table 1*). In addition to this, *Mtb* maintains a reduced cytosol by producing mycothiol (MSH; a low-molecular-weight sugar thiol) in millimolar amounts (*Kumar et al., 2011*). Interestingly, the cytoplasmic MSH-mediated redox buffering is intricately linked to the radical detoxification system by a membrane-associated oxidoreductase complex (MRC; SodA, DoxX, and SseA) in *Mtb* (*Nambi et al., 2015*). Disruption of any MRC-component impedes mycothiol recycling, increases oxidative damages, and enhances susceptibility to thiol stress (*Nambi et al., 2015*). Furthermore, *Mtb* expresses virulence factors that modulate host immune signalling and ROS- generating machinery to reduce the concentrations of ROS encountered by the bacteria inside the host (*Chai et al., 2020*; *Köster et al., 2017*). Additionally, the thick cell envelope consisting of cyclopropanated mycolic acids, phthiocerol dimycocerosates (PDIM), lipoarabinomannan, and phenolic glycolipid I (oxygen radical scavengers) confers an excellent anatomical barrier to and detoxification of exogenous oxidants

(*Hatfull and Jacobs, 2014*; *Tyagi et al., 2015*). Also, sulfur metabolism pathways (*Hatzios and Bertozzi, 2011*; *Kunota et al., 2021*), RHOCS (a redox homeostatic system; PknG, ribosomal protein L13, and RenU) (*Wolff et al., 2015*), DNA repair, and redox sensors (two-component systems such as DosR/S/T, WhiB family of transcriptional factors *Hatfull and Jacobs, 2014*; *Kumar et al., 2011*, and SigH-AosR/RshA *Khan et al., 2021*) allow *Mtb* to adapt and survive in response to exogenous redox stressors. While *Mtb* has evolutionarily adapted to counteract the host-generated exogenous redox stress, the pathogen is markedly sensitive to cell-permeable molecules that artificially elevate ROS and RNI inside the bacteria (*Singh et al., 2008*; *Tyagi et al., 2015*). For example, *Mtb* retains viability under high millimolar (mM) concentrations of exogenous ROS and RNI (*Voskuil et al., 2011*). However, it displays exceptional sensitivity to low micromolar (μM) concentrations of endogenous superoxide (2, 3-Dihydro-1,4, naphthoquinone) (*Tyagi et al., 2015*) and NO (PA-824) donors (*Singh et al., 2008*). In this context, PA-824 or Pretomanid, which releases NO inside *Mtb,* is lethal against both replicating and non-replicating *Mtb* (*Singh et al., 2008*) and has been approved to treat Multi-drug resistant (MDR) tuberculosis as B**Pa**LM regimen (*World Health Organization, 2022*). Taken together, these data indicate that disrupting intracellular redox homeostasis by enhancing eROS levels can be a promising strategy for identifying novel targets to kill *Mtb*.

Genetic determinants of redox homeostasis can coordinate the response to anti-TB drugs inside macrophages (*Mishra et al., 2019*; *Mishra et al., 2017*). Using a genetic biosensor (Mrx1-roGFP2) of MSH redox potential ($E_{MSH}$), we found that heterogeneity in $E_{MSH}$ of *Mtb* drives multidrug tolerance inside macrophages (*Bhaskar et al., 2014*; *Mishra et al., 2019*). Interestingly, several central metabolic enzymes (*e.g.*, isocitrate lyase; *Icl*) and redox regulators are required to maintain $E_{MSH}$, ergothioneine biogenesis, NAD$^+$/NADH poise, and to counter the lethal action of antibiotics in *Mtb* (*Chawla et al., 2012*; *Mishra et al., 2017*; *Nandakumar et al., 2014*; *Saini et al., 2016*; *Singh et al., 2009*). Indeed, the link between metabolism and antibiotic-mediated killing is further substantiated by studies showing that sensitivity to anti-TB drugs (bedaquiline, Q203, moxifloxacin, and clofazimine) is influenced by mycobacterial respiration and carbon catabolism (*Lamprecht et al., 2016*; *Mackenzie et al., 2020*; *Shee et al., 2022*). It appears that *Mtb* repurposes pathways involved in metabolism and redox homeostasis to mitigate immune and antibiotic pressures. These findings underscore the need for a comprehensive understanding of redox homeostasis in *Mtb*.

In this work, we repurposed transposon mutagenesis- deep sequencing (TnSeq) by combining it with the Mrx1-roGFP2 redox biosensor to discover novel pathways controlling basal $E_{MSH}$ of *Mtb*: –280±5 mV (*Figure 1A*). We built a Himar1-based saturated Tn mutant library (*Sassetti et al., 2003*; *Sassetti et al., 2001*; *Sassetti and Rubin, 2003*) in *Mtb* expressing Mrx1-roGFP2 (*Mtb*-roGFP2) and exploited FACS to reliably separate mutants displaying a shift from basal to oxidative $E_{MSH}$ during aerobic growth. We then developed a next-generation sequencing (NGS) and analysis pipeline for constructing a system-level network of genetic factors essential for maintaining redox homeostasis under aerobic growth in *Mtb*. We established the effectiveness of our strategy by discovering the role of an unknown transcriptional regulator encoded by *rv0158* in integrating the metabolism of carbon source with redox balance and bioenergetics in *Mtb*.

## Results

### Himar1-based transposon mutagenesis in *Mtb*-roGFP2

A library of transposon mutants (Tn-Library) comprising >$10^5$ independent insertion events was generated in *Mtb*-roGFP2 (*Figure 1A*). The pool was allowed to grow to mid-log-phase (OD$_{600\,nm}$ = 0.6–0.8), and genomic DNA was isolated for deep sequencing of transposon insertion (TnSeq) by Illumina technology. Using Transposon-directed insertion-site sequencing (TraDIS; *Barquist et al., 2016*), we identified the location of Tn insertion sites in the Tn-Library and detected insertion at ≈ 66% (49623 out of 74604) of TA sites (*Supplementary file 1*). Next, we quantified the frequency of Tn insertions at each open-reading frame (ORF) of *Mtb* (*Supplementary file 1*). Eighty-five percent of coding-DNA sequences incorporated at least 10 Tn insertion events in our Tn-Library. Consistent with previous studies, we observed a lack of Tn insertion in the TA sites of essential ORFs (≈11%) in the *Mtb* genome (*DeJesus et al., 2017*). Therefore, we conclude that the density of Tn insertion reached near saturation in the Tn-Library.

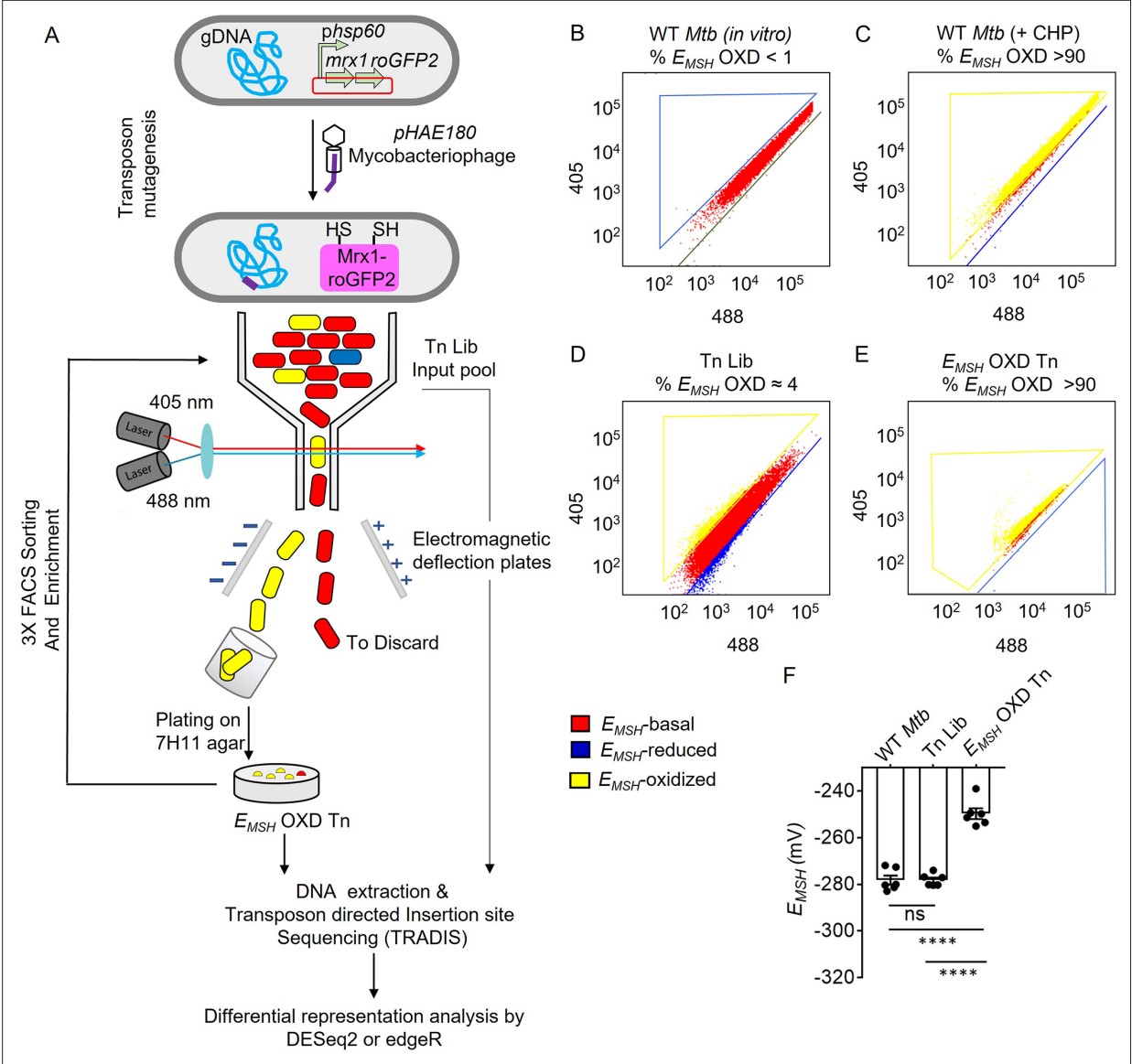

**Figure 1.** Detection and enrichment of $E_{MSH}$-oxidized *Mtb*-Tn mutants under *in vitro* growing conditions using FACS. (**A**) A saturated transposon mutant library in *Mtb*-roGFP2 was generated using a genetically engineered, temperature-sensitive mycobacteriophage *pHAE180*. Transposon mutants of *Mtb* were selected on 7H11 agar containing hygromycin and kanamycin. Genomic DNA was isolated and TraDIS (TnSeq) protocol was utilized to identify the transposon insertion site within the *Mtb* genome in the library of *Mtb*-transposon mutants. *Mtb* and *Mtb* Tn- Library expressing Mrx1-roGFP2 were grown in 7H9 supplemented with OADC and analyzed by flow-cytometry by exciting with 405 and 488 nm lasers at a constant emission (510 nm). The program BD FACS-Suite software was used to analyze the population distribution of bacteria, and a unique colour represented each population. $E_{MSH}$-basal, $E_{MSH}$-reduced, and $E_{MSH}$-oxidized subpopulations are shown in red, blue, and yellow, respectively. FACS Dot plot of (**B**) untreated *Mtb* expressing Mrx1-roGFP2 grown *in vitro* is shown in red. (**C**) *Mtb* cells treated with an oxidant- 10 mM cumene hydroperoxide (CHP) (shown in yellow). (**D**) Tn-Library (Tn Lib). The subpopulation indicated in blue is the $E_{MSH}$-reduced population as determined by *Mtb* cells treated with 20 mM dithiothreitol. (**E**) The $E_{MSH}$-oxidized subpopulation ($\approx$ 4%; $E_{MSH}$ OXD) was isolated by sorting, regrown on 7H11 agar, and resorted for three cycles to obtain $E_{MSH}$-oxidized Tn mutants. 10,000 events per sample were analyzed. (**F**) The calculated $E_{MSH}$ of WT *Mtb*, input Tn Lib, and flow-sorted $E_{MSH}$-oxidized Tn mutants. The data are means ± SEM of two independent experiments (n=6). (p>0.05: ns, p<0.0001: ****, one-way ANOVA with Tukey's multiple comparisons test).

The online version of this article includes the following source data and figure supplement(s) for figure 1:

**Source data 1.** FACS dot plots and $E_{MSH}$ numerical values of WT *Mtb*, input Tn Lib, and flow-sorted $E_{MSH}$-oxidized Tn mutants.

**Figure supplement 1.** Flow cytometry gating strategy to sort and enrich $E_{MSH}$-oxidized Tn mutants from the input Tn-Library (Tn Lib).

**Figure supplement 2.** Enriching the $E_{MSH}$-oxidized Tn mutants from the input Tn-Library (Tn Lib) by repeated cycles of sorting.

**Figure supplement 2—source data 1.** Numerical values indicating % population of $E_{MSH}$-oxidized Tn mutants after each round of sorting by FACS.

*Figure 1 continued on next page*

*Figure 1 continued*

**Figure supplement 3.** Accumulation of endogenous ROS inside $E_{MSH}$-oxidized *Mtb*-transposon mutant pool under *in vitro* growing conditions is dissipated by addition of multiple antioxidants, as detected by CellROX Deep Red dye.

**Figure supplement 3—source data 1.** Median fluorescence intensity values of CellROX Deep Red dye in WT *Mtb*, input Tn Lib, and flow-sorted $E_{MSH}$-oxidized Tn mutants.

## Flow-cytometry-based selection of *Mtb*-Tn mutants with altered basal $E_{MSH}$

We defined basal $E_{MSH}$ as the $E_{MSH}$ of mid-log phase *Mtb* grown in 7H9+ADS under standard aerobic culture conditions (180 RPM, 37 °C) (*Figure 1B and F*, *Figure 1—figure supplement 1b–f*). *Mtb* exhibited a basal $E_{MSH}$ of -278.2±4.7 mV (*Figure 1F*). We aimed to identify genes that, when mutated by transposon insertion, increase eROS and induce a shift in the basal $E_{MSH}$ of *Mtb* towards oxidative. The Tn-mutants displaying oxidative-$E_{MSH}$ were detected by quantifying the Mrx1-roGFP2 fluorescence ratio (excitation: 405 nm /488 nm; emission: 510 nm) of the library using flow-cytometry (*Bhaskar et al., 2014*). To sort out $E_{MSH}$-oxidized Tn-mutants, we utilized Fluorescence-activated Cell Sorting (FACS). To set the gates for sorting $E_{MSH}$-oxidized Tn-mutants by flow-cytometry, we treated wild-type (WT) *Mtb* H37Rv expressing Mrx1-roGFP2 with an oxidant, cumene hydroperoxide (CHP). Treatment with 10 mM CHP for 5 min maximally increased the 405/488 ratio (*Bhaskar et al., 2014*; *Shee et al., 2022*), indicating 100% oxidation of biosensor in WT *Mtb* ($E_{MSH}$-oxidized; –240 mV; *Figure 1C*, *Figure 1—figure supplement 1g*). Next, we subjected the Tn-mutant library (without CHP treatment) to FACS and selected the fraction of mutants displaying an increase in the biosensor ratio (*Figure 1D*). The biosensor ratio of this untreated Tn-mutant fraction overlapped with the ratiometric increase induced in WT *Mtb* by CHP, confirming the reliability of the sorting strategy (*Figure 1D*). Therefore, our strategy allowed the gating and isolation of Tn-mutants that were basally oxidized without exposure to oxidants such as CHP.

Using this approach, we consistently identified ≈ 4% of Tn-Library displaying oxidative-$E_{MSH}$ under aerobic culture conditions without exposure to exogenous oxidants (*Figure 1D*, *Figure 1—figure supplement 1i*). This mutant fraction was purified using FACS (*Figure 1—figure supplement 1j*). To enhance selectivity, sorted mutants from the first round were recultured on 7H11 agar plates, colonies were pooled, and re-sorted by flow-cytometry. This cycle was repeated thrice, improving the sensitivity and reliability of our selection by gradually enriching the minor fraction of $E_{MSH}$-oxidized Tn-mutants with each passage to >90% (*Figure 1E*, *Figure 1—figure supplement 2*). As expected, the enriched fraction of Tn-mutants showed a higher 405/488 ratio (0.7–0.9), which corresponded to an oxidative $E_{MSH}$ of –249±6.2 mV (*Figure 1F*). Consistent with this, measurement using the ROS-sensitive dye CellROX Deep Red showed that $E_{MSH}$-oxidized Tn-mutants accumulated ≈10-fold higher eROS than WT *Mtb* and the input pool of the Tn-Library (*Figure 1—figure supplement 3a*). In addition to this, co-treatment of $E_{MSH}$-oxidized Tn mutants with antioxidant molecules (*Shee et al., 2022*; combination of catalase (17.5 U/mL)+thiourea (1 mM)+bipyridyl (250 μM)) significantly decreased eROS levels (*Figure 1—figure supplement 3b*). Thus, by combining Tn mutagenesis with a redox biosensor, we reliably isolated redox-altered mutants of *Mtb* at a genomic scale.

## The $E_{MSH}$-oxidized Tn-mutant pool exhibited hallmarks of oxidative damage

The oxidative $E_{MSH}$ and increased eROS of Tn-mutants imply that this mutant pool experiences oxidative stress despite favorable aerobic growth conditions. Moreover, while each mutant is likely to get oxidized to a different extent, we expect that there will be an overall increase in the hallmarks of oxidative stress in the $E_{MSH}$-oxidized Tn-mutants fraction. To examine this, we assessed various oxidative stress markers, such as DNA damage, lipid peroxidation, and expression of antioxidant genes in $E_{MSH}$-oxidized Tn-mutants. The 8-hydroxy deoxyguanosine (an oxidative product of deoxyguanosine *David et al., 2007*; 8-OH dG) levels and lipid peroxidative products were ≈two-fold higher in $E_{MSH}$-oxidized Tn-mutants as compared to the Tn-Library and WT *Mtb*, indicating increased DNA and lipid damage (*Figure 2A and B*). As a positive control, we treated WT *Mtb* with CHP and found a similar increase in DNA and lipid damage (*Figure 2A and B*). Another signature of oxidative stress inside bacteria is the upregulation of oxidative stress-responsive genes. We found that antioxidant enzymes,

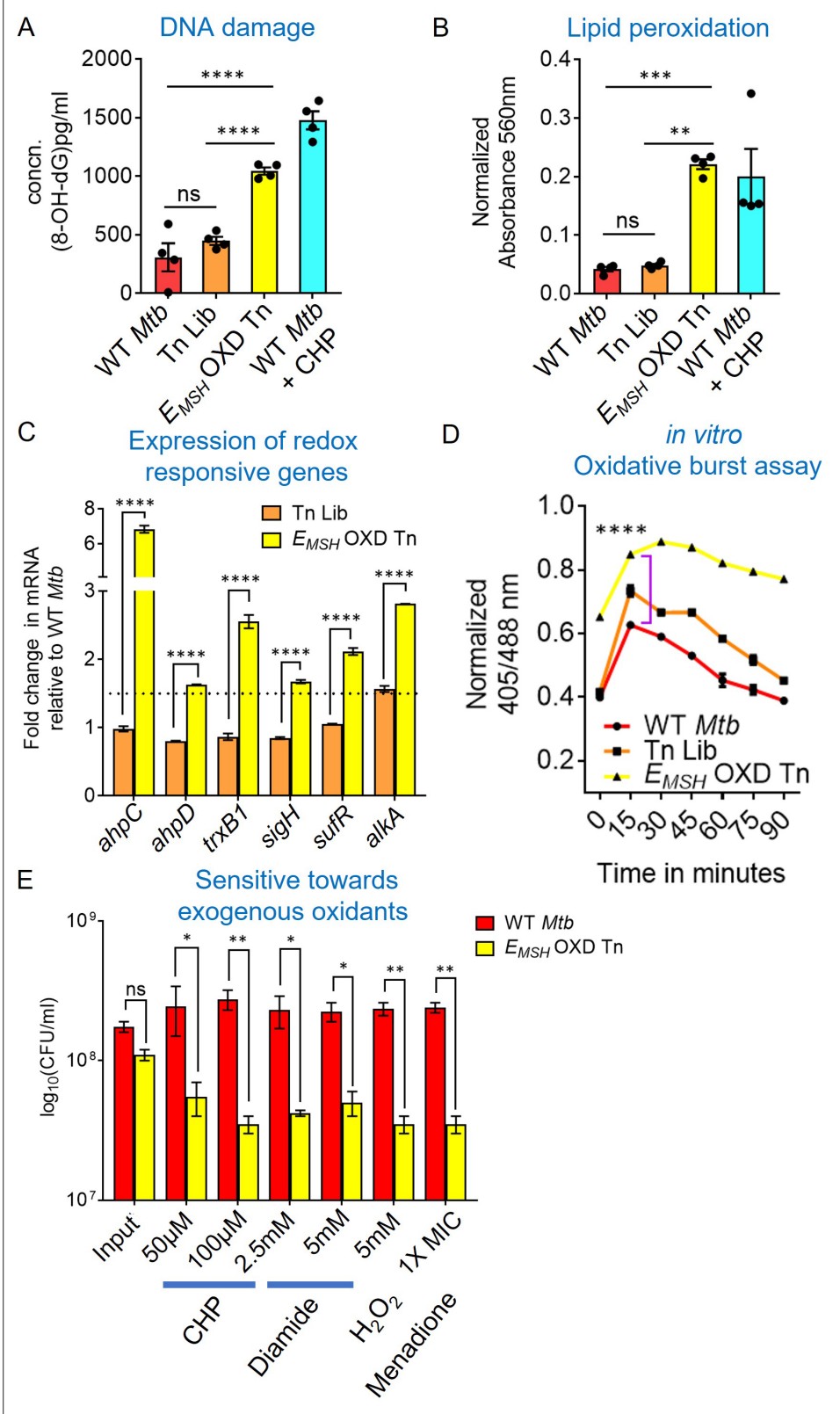

**Figure 2.** $E_{MSH}$-oxidized Tn mutants displayed signature of oxidative damage under *in vitro* growth conditions. Bacterial cultures were grown to log phase (OD$_{600}$ nm ≈ 0.6) (**A**) 8-hydroxy 2-deoxy guanosine, a major product of oxidative DNA damage was quantified using DNA/RNA Oxidative Damage ELISA Kit (Cayman chemicals). (**B**) Cellular lipid-hydroperoxides generation was measured by FOX2 assay. WT *Mtb* treated with 500 µM of cumene

*Figure 2 continued on next page*

*Figure 2 continued*

hydroperoxide (CHP) for 2 hr is used as a positive control. The data are means ± SEM of two independent experiments (n=4). (p>0.05: ns, p<0.01: **, p<0.001: ***, p<0.0001: ****, one-way ANOVA with Tukey's multiple comparisons test). (**C**) Realtime-quantitative PCR analysis showing increased basal expression of oxidative stress response genes. 16 S expression was used as a control. The dotted line indicates a 1.5-fold-change (cut-off value). (**D**) *Mtb*, Tn-Library (Tn Lib), and $E_{MSH}$-oxidized Tn-mutants ($E_{MSH}$-OXD Tn) were treated with a non-bactericidal concentration of CHP (12.5 µM), and the ratiometric sensor response was measured over time. Data representative of two independent experiments done in duplicate. Statistical comparisons are with respect to the WT *Mtb*. (p<0.0001: ****, unpaired two-tailed Student's t-test). (**E**) Log-phase cultures were exposed to oxidants with indicated concentration for 8 hr and plated. CFU was enumerated after 6 weeks. The data (**C and E**) are means ± SEM of two independent experiments. (p>0.05: ns, p<0.01: **, p<0.0001: ****, two-way ANOVA with Sidak's multiple comparisons test).

The online version of this article includes the following source data for figure 2:

**Source data 1.** Numerical values used to plot the graphs in *Figure 2A–E*.

such as *ahpC, ahpD, trxB1,* DNA repair protein *alkA*, a regulator of iron-sulfur cluster biogenesis *sufR*, and redox-stress responsive sigma factor *sigH* were ≈two- to eight-fold upregulated in $E_{MSH}$-oxidized Tn-mutants (*Figure 2C*).

We next assessed the dynamic response of $E_{MSH}$-oxidized Tn-mutants to an exogenous oxidant. We exposed WT *Mtb* expressing Mrx1-roGFP2, the Tn-Library, and $E_{MSH}$-oxidized Tn-mutants to a non-bactericidal concentration of CHP (12.5 µM; *Anand et al., 2021*). Exposure to CHP resulted in a rapid, but transient (90 min) increase in the 405/488 ratio (from 0.4 to 0.6) in WT *Mtb* and the Tn-Library, suggesting efficient mobilization of antioxidant machinery (*Figure 2D*). In contrast, $E_{MSH}$-oxidized Tn-mutants showed a significantly greater and long-lasting increase in the 405/488 ratio (from 0.65 to 0.9; *Figure 2D*). The slower recovery in biosensor ratio indicated that $E_{MSH}$-oxidized Tn-mutants have an impaired ability to orchestrate an efficient and dynamic antioxidant response. The breakdown of intracellular redox homeostasis enhances susceptibility toward exogenous stress (*Brynildsen et al., 2013*). Consistent with this, the $E_{MSH}$-oxidized Tn-mutant pool was nearly 10-fold more susceptible to diverse oxidants, including CHP, diamide (a thiol-specific oxidant), hydrogen peroxide ($H_2O_2$), and menadione (an $O_2^{\cdot-}$ generator; 1X MIC = 12.5 µg/mL) after 8 hr of treatment (*Figure 2E*). Altogether, data indicate that $E_{MSH}$-oxidized Tn-mutants collectively display features of oxidative stress.

## TraDIS analysis revealed genetic determinants of redox homeostasis under aerobic growth conditions in *Mtb*

To map the site of transposon insertions in redox-altered $E_{MSH}$-oxidized Tn-mutants, the TraDIS protocol (*Barquist et al., 2016*) was re-employed. Further, we used a differential gene representation analysis pipeline – DESeq2 (differential gene expression analysis based on the negative binomial distribution; *Love et al., 2014*) and edgeR (Empirical Analysis of Digital Gene Expression Data in R) (*Robinson et al., 2010*) to identify 368 genes containing significantly greater Tn insertions in $E_{MSH}$-oxidized Tn-mutants relative to the input Tn-Library ($log_2$ fold change >1, FDR or *p*. adj. <0.05; *Figure 3—figure supplement 1*, *Supplementary file 2*). We called this set of genetic factors '*Mtb* redoxosome'. Several genes known to be associated with redox homeostasis were part of the '*Mtb* redoxosome' (*Figure 3A*), validating our experimental approach for genome-scale identification of redox pathways. For example, genes involved in NAD metabolism (*pntAa, pntAb, pntB, sthA, pncB1*), redox-sensor (*whiB6*), cysteine metabolism (*cysK2, cysM, cds1 Kunota et al., 2021*), thiol buffering (*mshB, ino1, thiX, egtB, doxX*), antioxidant enzymes (*mymT, rv2633c, rv3177*), Fe homeostasis (*mbtL, dppA*), *sufR*, redox-regulated chaperone (*rv0991c Becker et al., 2020*), and respiration (*rv0247c, rv0248c, rv0249c, sdhA, sdhB, ctaC, ctaD, fixB*), were overrepresented in the $E_{MSH}$-oxidized Tn-mutant pool (*Figure 3A*). Additionally, we noted that several genes contributing to housekeeping functions such as cofactor biogenesis (*moaD2, moaX, moeA2, ribA1, lipA, panB, cobB*), oxidized DNA and lipid repair (*ephF, mutY*), membrane integrity, RNA processing, and amino acid metabolism were also part of *Mtb* redoxosome (*Figure 3A*, *Supplementary file 3*). Overall, we noted that ≈1/3$^{rd}$ of genes in the redoxosome have been previously, directly or indirectly associated with redox processes in *Mtb* (see details; *Supplementary file 3*).

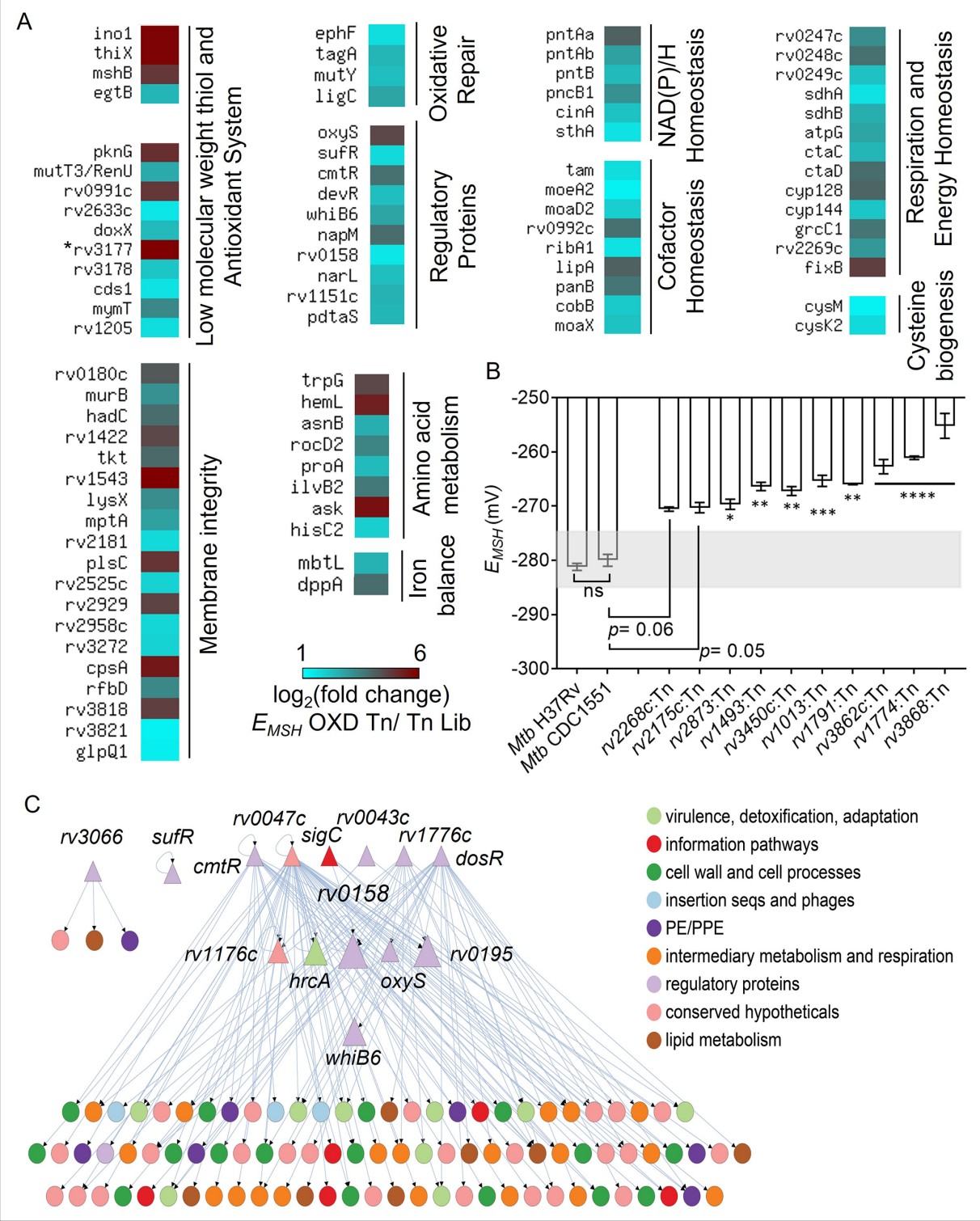

**Figure 3.** TraDIS analysis reveals determinants of redox homeostasis in *Mtb* under standard aerobic culture conditions. (**A**) Heat map showing genetic factors regulating $E_{MSH}$ of *Mtb* based on TraDIS analysis. These factors are functionally classified based on the MycoBrowser database and literature studies. The color gradient of the box indicates the $\log_2$ fold enrichment of respective genes in the $E_{MSH}$-oxidized Tn pool compared to the Tn Lib. The color code for the fold change is at the bottom of the second column. Genes are grouped according to function. (The gradient of cyan to red colour indicates increasing $\log_2$ fold change values). * *rv3177* is 10.02 $\log_2$ fold-change enriched. (**B**) The $E_{MSH}$ of *Mtb* mutants harboring transposon in the genes identified as part of *Mtb* redoxosome. The mutants were grown to log phase (OD $_{600\,nm} \approx 0.6$) under standard aerobic growth conditions, and $E_{MSH}$ was measured. The light grey area represents basal $E_{MSH}$. The data are means ± SEM of two independent experiments (n=4). (p>0.05: ns, p<0.05: *, p<0.01:

*Figure 3 continued on next page*

*Figure 3 continued*

\*\*, p<0.001: \*\*\*, p<0.0001: \*\*\*\*, one-way ANOVA with Dunnett's multiple comparisons test). (**C**) Hierarchical visualization of the redox-homeostasis network of 107 genes (14 TFs, 93 target genes) and 143 edges. Here, the TFs are shown as triangles, and the target genes are shown as circles. The size of the TFs (triangles) is proportional to their betweenness centrality. The figure was generated using Cytoscape 3.6. The TF Rv0158 in the middle layer has the highest betweenness centrality and, thus, a bottleneck in the redox-homeostasis network.

The online version of this article includes the following source data and figure supplement(s) for figure 3:

**Source data 1.** $E_{MSH}$ numerical values of *Mtb* mutants harboring transposon in the genes identified as part of *Mtb* redoxosome.

**Figure supplement 1.** Strategy to identify genes critical for preserving redox balance in *Mtb* using integrated transposon mutagenesis, FACS, deep Sequencing, and redox biosensor technology.

**Figure supplement 2.** Transposon insertion in the genes of redoxosome resulted in elevated ROS levels and DNA damage in the individual Tn mutants.

**Figure supplement 2—source data 1.** Median Fluorescence Intensity values of CellROX Deep Red dye (quantification of endogenous ROS inside cell) and TMR red- labelled nucleotides (quantification of DNA breaks) in individual *Mtb* Tn mutants.

While experimentally ascribing redox function to 368 genes of the redoxosome requires independent investigation, we begin validating our data by randomly selecting 10 genes that are part of the redoxosome and analysed $E_{MSH}$ of *Mtb* strains harboring disruption in these genes. These 10 mutants were generated by transposon mutagenesis in the *Mtb* CDC1551 strain (collected from BEI resources). Both *Mtb* CDC1551 and *Mtb* H37Rv displayed identical basal $E_{MSH}$ (−280 to −285 mV; *Figure 3B*). As expected, the $E_{MSH}$ of *Mtb* mutants displayed an oxidative shift of +10 mV to +30 mV and exhibited higher eROS and DNA damage as compared to WT *Mtb* (*Figure 3B*, *Figure 3—figure supplement 2*). Therefore, we propose that the set of 368 genes represents a valuable resource to understand the contribution of *Mtb* factors in maintaining redox balance in *Mtb*.

## Network analysis identified *rv0158* as an important regulator of redox balance in *Mtb*

To the best of our knowledge, the identity of regulator(s) that helps *Mtb* in maintaining redox balance during aerobic growth is not known. Although, similar studies have been performed on non-pathogenic *E. coli* (*Brynildsen et al., 2013*; *Lopatkin et al., 2021*) and *S. cerevisiae* (*Ayer et al., 2012*; *Radzinski et al., 2018*). We, therefore, adopted a system-scale approach to analyze the redoxosome and to discover a network of regulators responsible for maintaining basal $E_{MSH}$ in aerobically growing *Mtb*. To this end, we generated a redox-homeostasis network by integrating 368 redoxosome genes with the *Mtb* transcriptional regulatory network (TRN) (*Minch et al., 2015*; *Figure 3C*). A similar approach has been used to identify transcriptional regulators coordinating drug tolerance (*Peterson et al., 2016*), adaptation to hypoxia or nutrient starvation (*Balázsi et al., 2008*) and to host-stress (*Peterson et al., 2019*) in *Mtb*. In our redox-homeostasis regulatory network of 107 nodes and 143 edges, a node represents a TF (transcription factor) or a target gene, and a directed edge indicates the transcriptional activity of a target gene by a TF. Further, the largest component of 102 genes and 139 edges in the redox-homeostasis network can be organized into three hierarchical levels with 6 TFs in the top, 6 TFs in the middle, and 90 target genes in the bottom (*Figure 3C*).

Among the 6 TFs in the top layer are known regulators- *dosR* responsive to $O_2$/NO/CO/$H_2$S (*Kumar et al., 2011*; *Park et al., 2003*; *Sevalkar et al., 2022*), *cmtR* ($Pb^{2+}$/$Cd^{2+}$Sensing; $H_2O_2$-dependent ESX3 regulator *Cavet et al., 2003*; *Chauhan et al., 2009*; *Li et al., 2020*), *sigC* (Cu acquisition *Grosse-Siestrup et al., 2021*), and recently elucidated *rv0047c* (Nucleoid-associated protein that act as DnaA- antagonist to regulate *Mtb* replication *Liu et al., 2019*; *Liu et al., 2016*). In terms of the number of target genes or out-degree, *rv0047c* has the highest degree, followed by *dosR* in the redox-homeostasis network (*Figure 3C*). We also measured betweenness centrality, which measures the fraction of paths in the network that pass-through a given node. A high betweenness centrality indicates that the node is influential in maintaining the network's connectivity. We found that the TFs in the top layer including *dosR* and *rv0047c* have negligible betweenness centrality compared to those in the middle layer. In particular, we noted that *rv0158* has the highest betweenness centrality in the complete network and the highest out-degree among TFs in the middle layer (*Figure 3C*). These features indicate that *rv0158* is both a bottleneck and a hub in the redox-homeostasis network. Furthermore, *rv0158* is a part of 2 out of the 3 feed-forward loops (FFLs *Mangan and Alon, 2003*) in

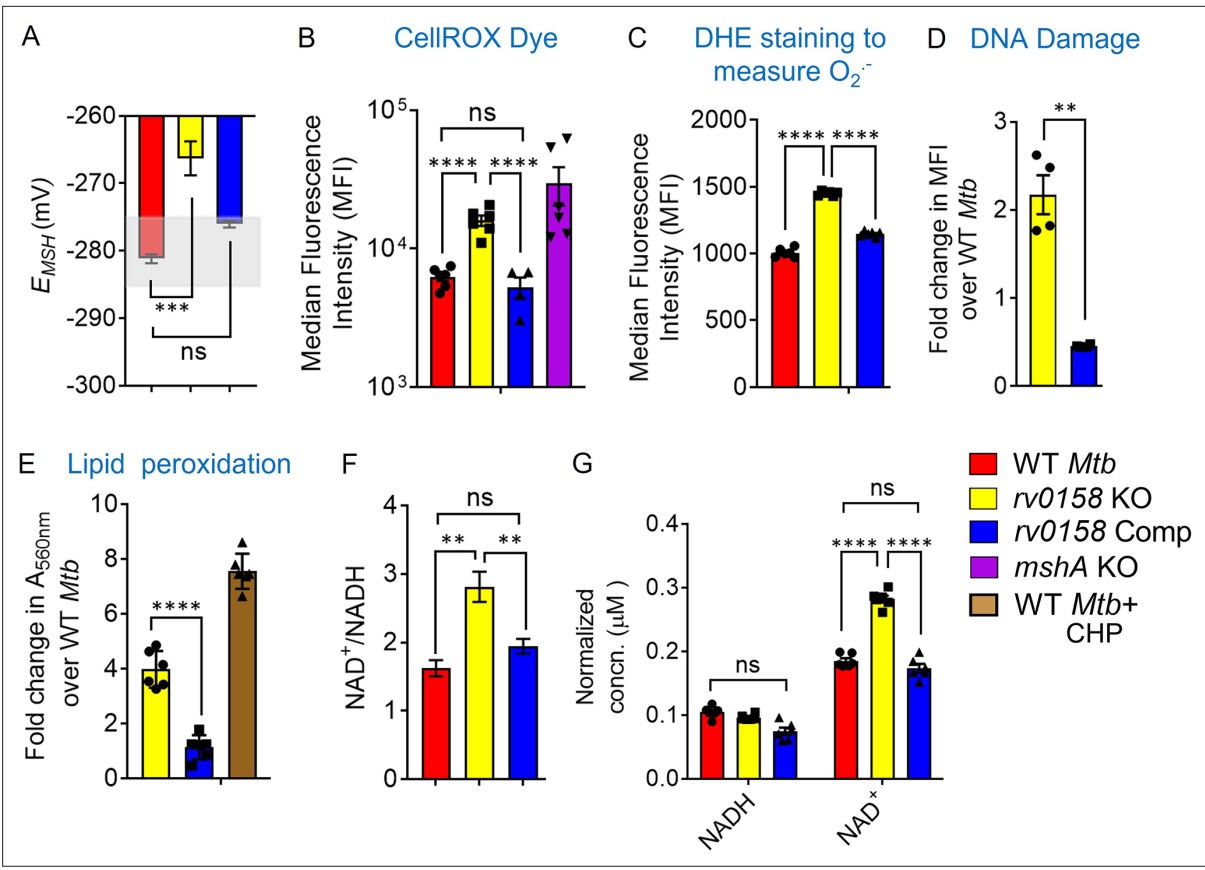

**Figure 4.** *rv0158* KO experiences enhanced eROS, oxidative damage to biomolecules and altered NAD$^+$/NADH during normal aerobic growth. Bacterial cultures were grown to log phase (OD$_{600\,nm}$ ≈ 0.6) (**A**) The $E_{MSH}$ of WT *Mtb* (WT *Mtb*), *rv0158* KO, and *rv0158* Comp. The light grey area represents basal $E_{MSH}$. The data are means ± SEM of two independent experiments (n=4). (p>0.05: ns, p<0.001: ***, one-way ANOVA with Dunnett's multiple comparisons test). (**B, C**) eROS and superoxide levels were quantified using FACS by measuring the fluorescence of cell-permeant CellROX Deep Red dye with absorption/emission maxima of ~644/665 nm and Dihydroethidium (Ex/Em: 488 nm/ 560 nm) dye, respectively. Median Fluorescence Intensity (MFI) is plotted. A total of 10,000 events per sample were analyzed. (**D**) Double-stranded DNA breaks were quantified by TUNEL Assay using Flow-cytometry. Fold change in MFI is plotted. The data are means ± SEM of two independent experiments (n=4). (p<0.05: *, unpaired two-tailed Student's t-test with Welch's correction). (**E**) Cellular lipid-hydroperoxides levels were measured by FOX2 assay, and absorbance at 560 nm is normalized to that of WT *Mtb* H37Rv. (**F**) NAD$^+$/NADH ratio (**G**) pyridine nucleotide levels as estimated by an enzymatic assay using alcohol dehydrogenase and ethanol as enzyme and substrate, respectively. The concentration was normalized to the cell mass. The data are means ± SEM of two independent experiments (n≥4). (For panel other than E, statistical measurement: p>0.05: ns, p<0.05: *, p<0.01: **, p<0.0001: ****, one-way ANOVA with Tukey's multiple comparisons test).

The online version of this article includes the following source data and figure supplement(s) for figure 4:

**Source data 1.** Numerical values used to plot the graphs in *Figure 4A–G* showing perturbed redox state in *rv0158* KO *Mtb*.

**Figure supplement 1.** Graphic illustration of generation of *rv0158* knockout *Mtb* strain.

**Figure supplement 1—source data 1.** Raw data plots of qRT- PCR as shown in *Figure 4—figure supplement 1b*.

the redox-homeostasis network. In sum, *rv0158* owing to its position in the middle of the hierarchy, along with the highest betweenness centrality, could function to regulate redox signalling in *Mtb*.

## RNA-Sequencing indicates that *rv0158* regulates lipid metabolism and oxidative stress response in *Mtb*

Based on our in silico analysis, we aimed to understand the mechanism of how *rv0158* regulates redox homeostasis in *Mtb*. We first deleted the *rv0158* gene in *Mtb* by an allelic-exchange strategy and confirmed the knockout (KO) by qRT-PCR (*Figure 4—figure supplement 1*). Similar to findings with the Tn-mutant pool, $E_{MSH}$ of *rv0158* KO displayed oxidative shift (–266.3±5 mV; *Figure 4A*), accumulated eROS (*Figure 4B*), and superoxide radical (*Figure 4C*), exhibited increased DNA double-strand

breaks (*Figure 4D*), and lipid hydroperoxides compared to WT *Mtb* and *rv0158* Comp (*Figure 4E*). The mutant also showed a ≈ two-fold increase in NAD$^+$/NADH ratio relative to WT *Mtb* and *rv0158* Comp (*Figure 4F and G*).

We next performed whole-genome RNA sequencing (RNA-Seq) to identify *rv0158*-dependent gene expression in *Mtb*. Bacterial strains were grown to log phase (OD$_{600\,nm}$ = 0.6), RNA was harvested and subjected to RNA-seq. We identified 232 genes that were differentially expressed (DE genes; fold change >1.5; FDR <0.05) in the absence of *rv0158* (*Figure 5A*, *Supplementary file 4*). We validated the RNA-seq data by examining the expression of 18 randomly selected genes by qRT-PCR under identical experimental conditions (*Figure 5—figure supplement 1*). Consistent with elevated eROS and DNA damage in *rv0158* KO, genes involved in the antioxidant response, such as thioredoxin (*trxC*), catalase-peroxidase (*katG*), chaperones (*groEL1, groEL2, groES*, and *hsp*), epoxide hydrolase (*ephA*), and DNA-repair genes (*ung, dnaE2*) were upregulated (*Figure 5B*). Supporting these observations, the *rv0158* transcriptome showed strong overlap with the transcriptomes of oxidative stress (H$_2$O$_2$; odds ratio = 3.4, p=2*10$^{-7}$) and nitrosative stress (NO; odds ratio = 3, p=2*10$^{-6}$). In contrast, transcriptional changes associated with acidic pH, known to induce reductive $E_{MSH}$ in *Mtb* (*Mehta et al., 2016*) and hypoxia did not significantly overlap with the *rv0158* transcriptome (*Figure 5—figure supplement 2*). Collectively, the data are consistent with our hypothesis that *rv0158* maintains redox balance and eROS levels.

Functional classification of DE genes based on the MycoBrowser database (*Kapopoulou et al., 2011*) confirmed a ≈three fold overrepresentation of genes belonging to 'Lipid Metabolism' (37 genes) in the mutant (*Figure 5C*, *Supplementary file 4*). Genes involved in β-oxidation of fatty acids catabolism (*fadA, fadB, echA7, fadE12, fadE13, fadE20, fadD7, accA2, accD2, fadE23, fadE24, fadD9, fadD22*), propionate metabolism (*prpC, prpD, icl1*) (*Eoh and Rhee, 2014*) were significantly induced, while mycolic acid biosynthesis and modification (*fabD, fabG4, htdX, desA1, desA2, desA3*) were downregulated in the KO strain (*Figure 5B*, *Supplementary file 4*). Agreeing to transcriptional changes, radiolabelling of mycolic acids with $^{14}$C acetate showed a ≈ 40% reduction in mycolic acid methyl esters (MAMEs; α, methoxy, and keto) and a simultaneous accumulation of fatty acid methyl esters (FAMES; precursors of mycolic acids) in *rv0158* KO as compared to WT *Mtb* and *rv0158* Comp (*Figure 6A-C*). A decrease in mycolic acid content increases *Mtb*'s outer membrane permeability (*Gao et al., 2003*; *Liu et al., 1996*). Consistent with this, we observed a higher intracellular accumulation of $^{14}$C-isoniazid (*Figure 6—figure supplement 1*) and ethidium bromide (EtBr) in *rv0158* KO than WT *Mtb* and *rv0158* Comp (*Figure 6D*). Altogether, loss of *rv0158* induces widespread transcriptional changes, significantly influencing membrane integrity and fatty acid metabolism in *Mtb*.

## *rv0158* regulates growth, mycothiol-redox potential ($E_{MSH}$), and bioenergetics in response to carbon (C)-source

Our data suggest that genes associated with fatty acid catabolism were induced in *rv0158* KO. To examine the role of *rv0158* in utilizing fatty acids as nutrients by *Mtb*, we measured the growth of WT *Mtb*, *rv0158* KO, and *rv0158* Comp in 7H9 medium supplemented with either 0.3% glucose or host-relevant fatty acids (0.3% sodium acetate or 0.02% cholesterol or 200 µM palmitate) as a sole carbon (C) source (*Figure 7A–D*, *Figure 7—figure supplement 1*). Both absorbance (OD 600 nm) and CFU estimation showed a slow growth phenotype with an extended lag phase on glucose (*Figure 7A*, *Figure 7—figure supplement 1*). The defective growth phenotype of *rv0158* KO was absent if glucose was replaced by acetate/cholesterol/palmitate (*Figure 7B–D*). The methylcitrate cycle (MCC) genes (*prpC, prpD, icl1*), which alleviate propionate toxicity (*Eoh and Rhee, 2014*), were significantly upregulated in *rv0158* KO. Therefore, we tested the phenotype of *rv0158* KO on propionate. We indeed observed a gradual increase in the growth of *rv0158* KO on 0.3% sodium propionate, a concentration bacteriostatic to both WT *Mtb* and *rv0158* Comp (*Figure 7E*).

Next, we examined the influence of C source on $E_{MSH}$. The *rv0158* KO was significantly more oxidized ($E_{MSH}$ = –266.3 ± 5) as compared to WT *Mtb* ($E_{MSH}$ = –279.4 ± 3.5) and *rv0158* Comp ($E_{MSH}$ = –277.4 ± 2) on glucose (*Figure 7F*). We further observed that WT *Mtb* cells growing on acetate or propionate or cholesterol uniformly exhibit a reductive shift in $E_{MSH}$ (–285 to –305 mv) compared to glucose (–279 mV), agreeing with fatty acids being a more reduced C substrate than glucose (*Borah et al., 2021*; *Singh et al., 2009*). Interestingly, oxidative-$E_{MSH}$ of *rv0158* KO showed a reductive shift and recovered to WT *Mtb* levels (–285 to –305 mv) on acetate/propionate/cholesterol (*Figure 7F*).

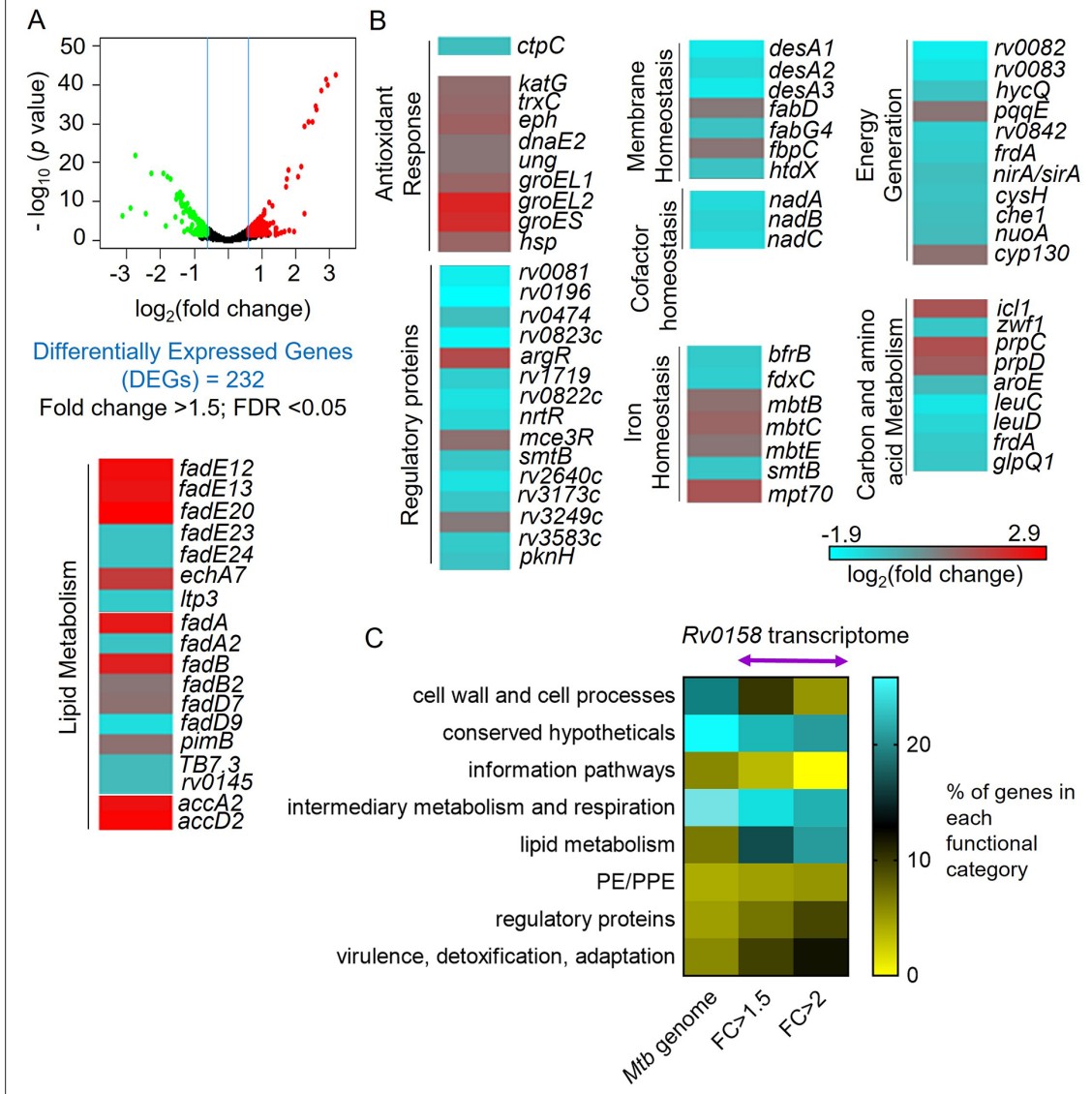

**Figure 5.** RNA Sequencing analysis reveals that *rv0158* presumably regulates lipid metabolism as a mechanism of redox homeostasis. (**A**) Volcano plot of differentially expressed genes (absolute fold change >1.5; FDR <0.05) between the WT *Mtb* and *rv0158* KO grown *in vitro*. (**B**) Heat-map comparing expression profile in WT and *rv0158* KO. The colour code for the fold change is at the bottom of the third column (red: upregulated genes; cyan: down-regulated genes). Genes are grouped according to function. (**C**) Functional categorization of *rv0158*-dependent genes in *Mtb*. The % of genes in each functional category (according to the MycoBrowser database) was determined for the complete *Mtb* genome and for *rv0158* transcriptome with different cut-offs (FC ≥1.5: absolute fold change ≥1.5, FDR <0.05, differentially expressed genes = 232 and FC ≥2: absolute fold change ≥2; FDR <0.05, differentially expressed genes = 77).

The online version of this article includes the following source data and figure supplement(s) for figure 5:

**Source data 1.** Numerical values of the % of genes in each functional category (according to the MycoBrowser database) of the complete *Mtb* genome and for *rv0158* transcriptome.

**Figure supplement 1.** Validation of RNA Sequencing results by measuring the expression of a set of genes in wild-type *Mtb*, *rv0158* KO and complemented strain by quantitative real-time PCR (qRT-PCR) analysis.

**Figure supplement 1—source data 1.** Numerical values indicating fold changes in *rv0158*- dependent gene expression.

**Figure supplement 2.** Heatmap showing differential gene overlap between *rv0158* transcriptome and various stress conditions.

**Figure supplement 2—source data 1.** Raw values of differential gene overlap between *rv0158* transcriptome and various stress conditions.

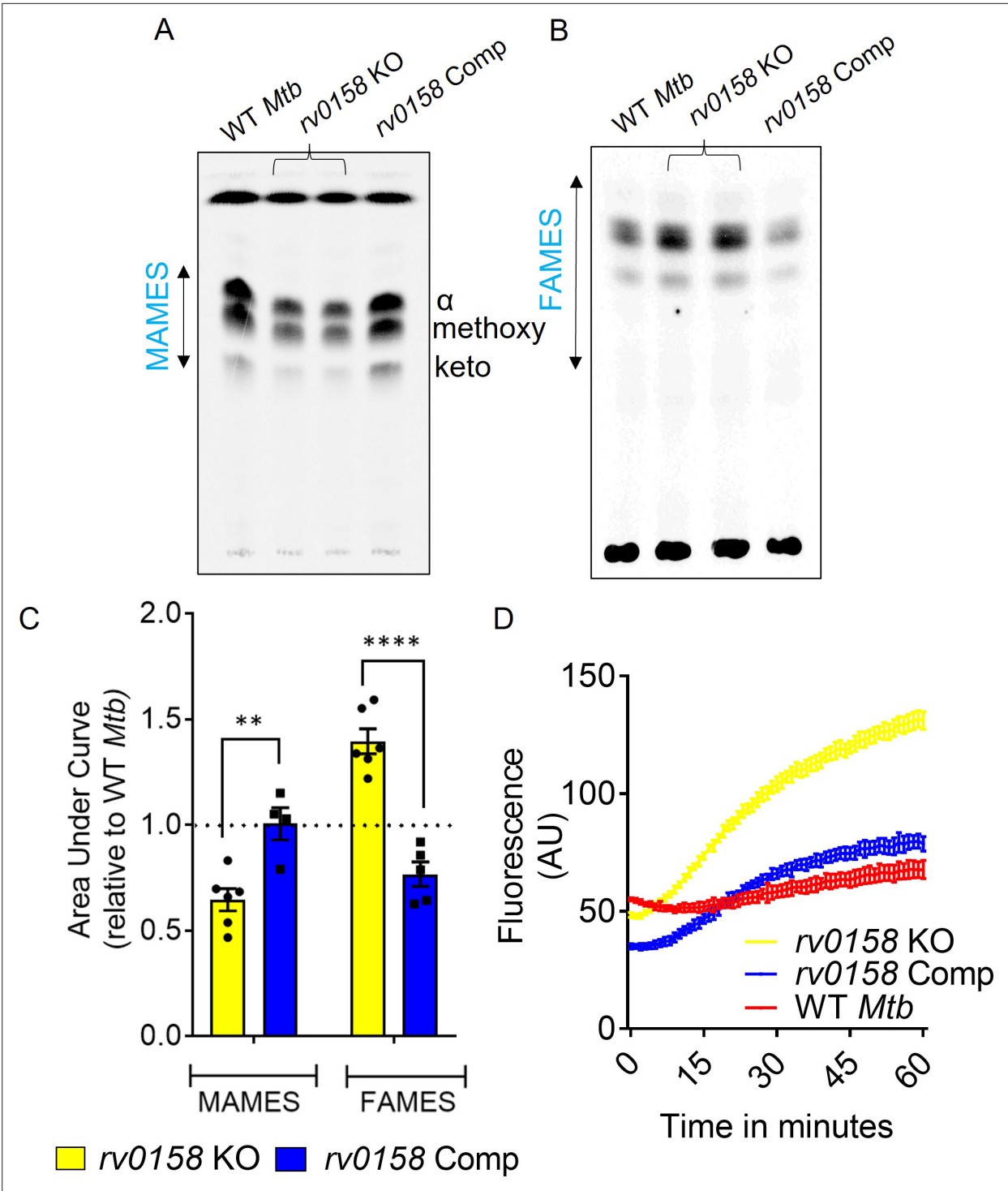

**Figure 6.** *rv0158* deletion impaired mycolic acid biogenesis and increased membrane permeability in *Mtb*. WT *Mtb*, *rv0158* KO, and *rv0158* Comp were cultured in 7H9+ADS to log phase (OD $_{600\,nm}$ ≈ 0.8), and total lipids were labeled using [1, 2-$^{14}$C] -sodium-acetate for 24 hr. TLC-autoradiograph analysis of mycolic acids (MAMEs) and fatty acids (FAMEs)- fraction from total lipids was done by loading equal counts (50000 CPM). (**A**) Normal-phase TLC of MAMEs from bacterial strains was developed using hexane/ethyl acetate (19:1, v/v) solvent system. See Source Data: *Figure 6—source data 1* used to generate this figure. ImageJ software (1.53e version) was used to edit the images. (**B**) Reverse-phase TLCs using acetonitrile/dioxane solvent (1:1, v/v) showing the fatty acid (FAMEs) distribution. See Source Data: *Figure 7—source data 1* used to generate this figure. ImageJ software (1.53e version) was used to edit the images. (**C**) Densitometric analysis by ImageJ software (1.53e version) to quantify changes in MAMEs and FAMES relative to WT levels. The data are means ± SEM of two independent experiments (n≥4). Statistical comparisons are with respect to the WT *Mtb*. (p<0.01: **, p<0.0001:

*Figure 6 continued on next page*

*Figure 6 continued*

****, two-way ANOVA with Sidak's multiple comparisons test). (**D**) Ethidium bromide accumulation within *Mtb* strains grown on glucose as a function of time. Results are representative of three independent experiments performed in quadruplicate.

The online version of this article includes the following source data and figure supplement(s) for figure 6:

Source data 1. Raw files containing TLC images of MAMEs and FAMEs.

Source data 2. Numerical values used to plot graphs in *Figure 6C* and *Figure 6D*.

Figure supplement 1. Increased membrane permeability of *rv0158* KO resulted in higher intracellular accumulation of [14]C- isoniazid.

Figure supplement 1—source data 1. Numerical values used to plot bar graph in *Figure 6—figure supplement 1*.

Variable growth on fatty acids versus carbohydrates directly influences the bioenergetics of *Mtb* (*Lamprecht et al., 2016*). We tested the bioenergetic state of *Mtb* strains utilizing either glucose, acetate, or propionate as a C source using the non-invasive Seahorse XFp analyzer. Bacterial strains were starved overnight, and ECAR (extracellular acidification rate) and OCR (oxygen-consumption rate) between WT *Mtb* and *rv0158* KO were measured by sequentially injecting a C source and membrane potential uncoupler (CCCP) in the XF microchamber. CCCP depolarizes the cell membrane and stimulates maximum respiration in *Mtb*. Fold changes were measured by comparing the OCR or ECAR readings after glucose or CCCP addition. When glucose at 0.3% was injected, the %OCR and %ECAR were ≈ two fold less in *rv0158* KO than in WT *Mtb* and *rv0158* Comp (*Figure 8A and D*). As expected, the addition of CCCP stimulated OCR and ECAR in all strains; however, the increase was ≈two fold lower in *rv0158* KO (*Figure 8A and D*). Importantly, impaired OCR and ECAR of *rv0158* KO fully recovered to WT *Mtb* levels when 0.3% acetate or 0.3% propionate was provided as the sole C source (*Figure 8B, C, E and F*). Furthermore, adding CCCP stimulates OCR and ECAR in *rv0158* KO more than WT *Mtb* and *rv0158* Comp in the propionate-containing medium (*Figure 8C and F*). Since an equal number of viable cells ($2 \times 10^6$ cells/well) were used for these assays, the changes in OCR and ECAR were likely due to altered bioenergetics of the *rv0158* KO strain.

## Metabolic profiling of *rv0158* KO defines propionate adaptability

To elucidate metabolic changes associated with the better adaptability of *rv0158* KO to propionate as a C source, we performed liquid chromatography-mass spectrometry (LC-MS) metabolomics following exposure of WT *Mtb* and *rv0158* KO to 0.3% glucose or 0.3% propionate. The consumption of propionate requires overlapping reactions encompassing the TCA cycle, methyl citrate cycle (MCC), and glyoxylate cycle (*Eoh and Rhee, 2014*). Therefore, we focussed on the steady-state metabolite levels of the intermediates associated with these pathways. The metabolites were extracted, derivatized with O-benzylhydroxylamine (OBHA), and detected based on the LC-MS profile of the TCA cycle and glyoxylate cycle metabolite standards, using methods described earlier (*Tan et al., 2014*; *Walvekar et al., 2018*). In particular, since some TCA cycle metabolites such as oxaloacetate are unstable in aqueous environments, the OBHA derivatization after extraction allows their reliable detection and quantitation (*Figure 9—figure supplements 1–2*) (also see *Walvekar et al., 2018* for details). We observed a ≈100-fold accumulation of 2-methyl(iso)citrate (2 M(I)C) in WT *Mtb* and *rv0158* KO under propionate but not glucose (*Figure 9*), confirming the activation of propionate breakdown by MCC. Furthermore, downstream metabolites of propionate catabolism such as pyruvate, succinate, fumarate, malate, oxaloacetate, and glyoxylate were significantly accumulated (two- to six fold) in propionate-grown WT *Mtb* as compared to glucose (*Figure 9*). In contrast, these metabolites were either not accumulated (*e.g.* succinate, malate, and oxaloacetate) or marginally accumulated (*e.g.* fumarate and glyoxylate) in propionate-grown *rv0158* KO (*Figure 9*), presumably due to efficient catabolism of propionate via MCC in the mutant relative to WT *Mtb*. Similarly, propionate-grown WT *Mtb* displayed an increased pool of pyruvate and lactate, but these metabolites were relatively lesser in the *rv0158* KO (*Figure 9*). Since the levels of pyruvate and lactate depend on glyoxylate shunt, MCC, and gluconeogenesis (*Serafini et al., 2019*), these findings agree with the better metabolic ability of *rv0158* KO to consume propionate. Lastly, although propionate had similar metabolic consequences for both WT *Mtb* and *rv0158* Comp, the decrease in the pool sizes of metabolic intermediates in *rv0158* KO was more evident compared with *rv0158* Comp (*Figure 9*). Altogether, data suggest that *rv0158* KO mitigates propionate toxicity by efficient metabolization via MCC and TCA cycle.

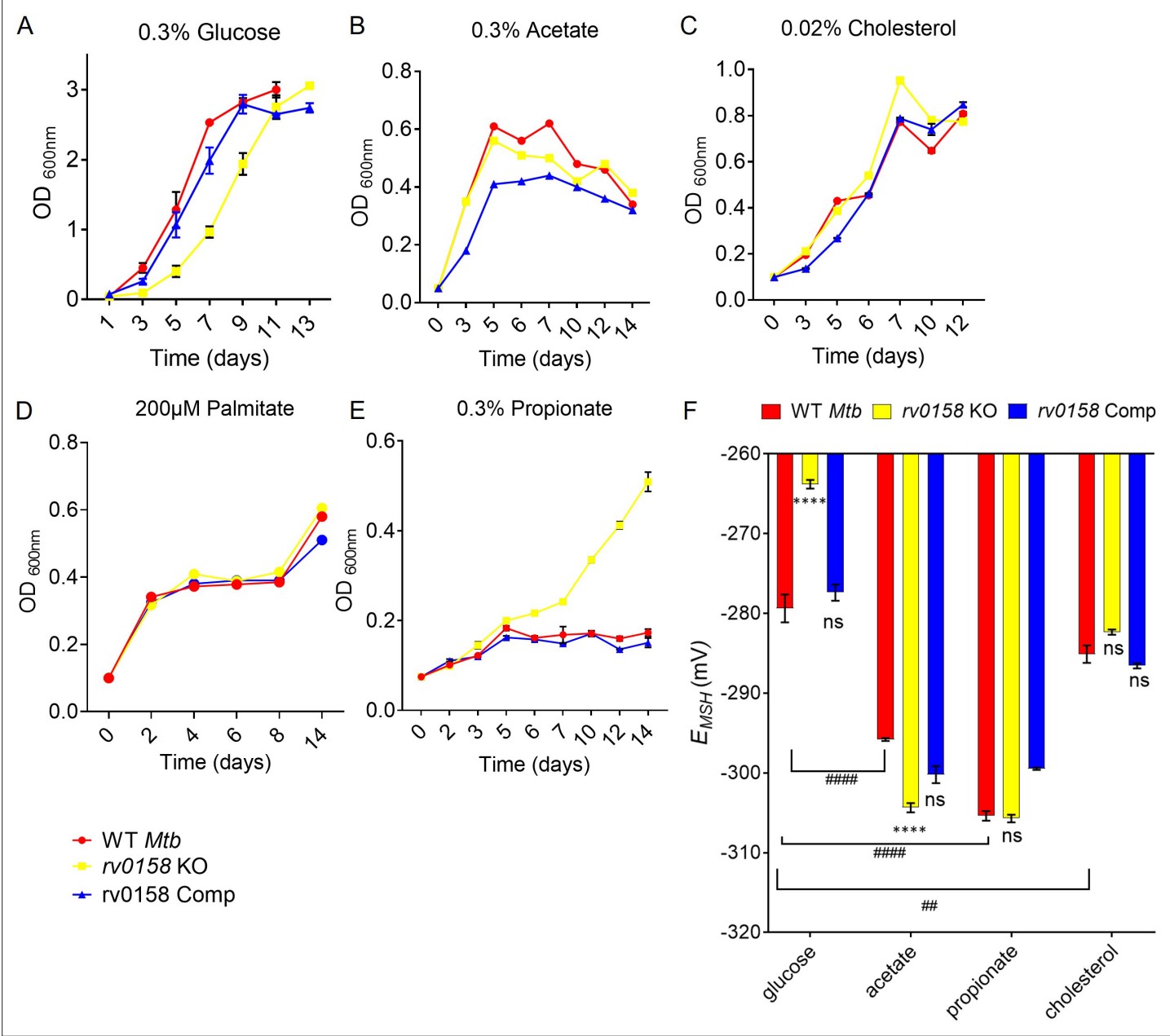

**Figure 7.** *rv0158* deletion improved *Mtb* growth and intracellular redox state with fatty acids as carbon source. (**A– E**) Log-phase cultures of *Mtb* strains (WT *Mtb*, *rv0158* KO, and *rv0158* Comp) were synchronized to $OD_{600}$=0.1 and grown in 7H9 broth (no glycerol) containing (**A**) 0.3% D-glucose, (**B**) 0.3% sodium acetate, (**C**) 0.02% cholesterol (**D**) 200 μM palmitate, and (**E**) 0.3% sodium propionate. Culture turbidity ($OD_{600\ nm}$) was measured at the indicated time points for determining growth kinetics. Results are representative of two independent experiments. (**F**) C source-dependent changes in the redox state of *Mtb* is mediated by *rv0158*. *Mtb* strains (WT *Mtb*, *rv0158* KO, and *rv0158* Comp) were synchronized to $OD_{600\ nm}$ = 0.1 and grown in 7H9 broth (no glycerol) containing 0.3% D-glucose or 0.3% sodium acetate or 0.3% sodium propionate or 0.02% cholesterol to log-phase ($OD_{600\ nm}$ = 0.5–0.6) and Mrx1-roGFP2 response was measured. The data are means ± SEM of two independent experiments (n=4). p was determined by two-way ANOVA with Tukey's multiple comparisons test (**** p<0.001, ns indicates not significant: Comparison is performed between KO or Comp strain and WT *Mtb*; ####: p<0.001: ##: p<0.01, the comparison is performed between WT *Mtb* grown in different C sources).

The online version of this article includes the following source data and figure supplement(s) for figure 7:

**Source data 1.** Numerical values used to plot graphs in *Figure 7A–F*.

**Figure supplement 1.** *rv0158* deletion leads to extended lag phase in *Mtb*.

**Figure supplement 1—source data 1.** Numerical values used to plot bar graph in *Figure 7—figure supplement 1*.

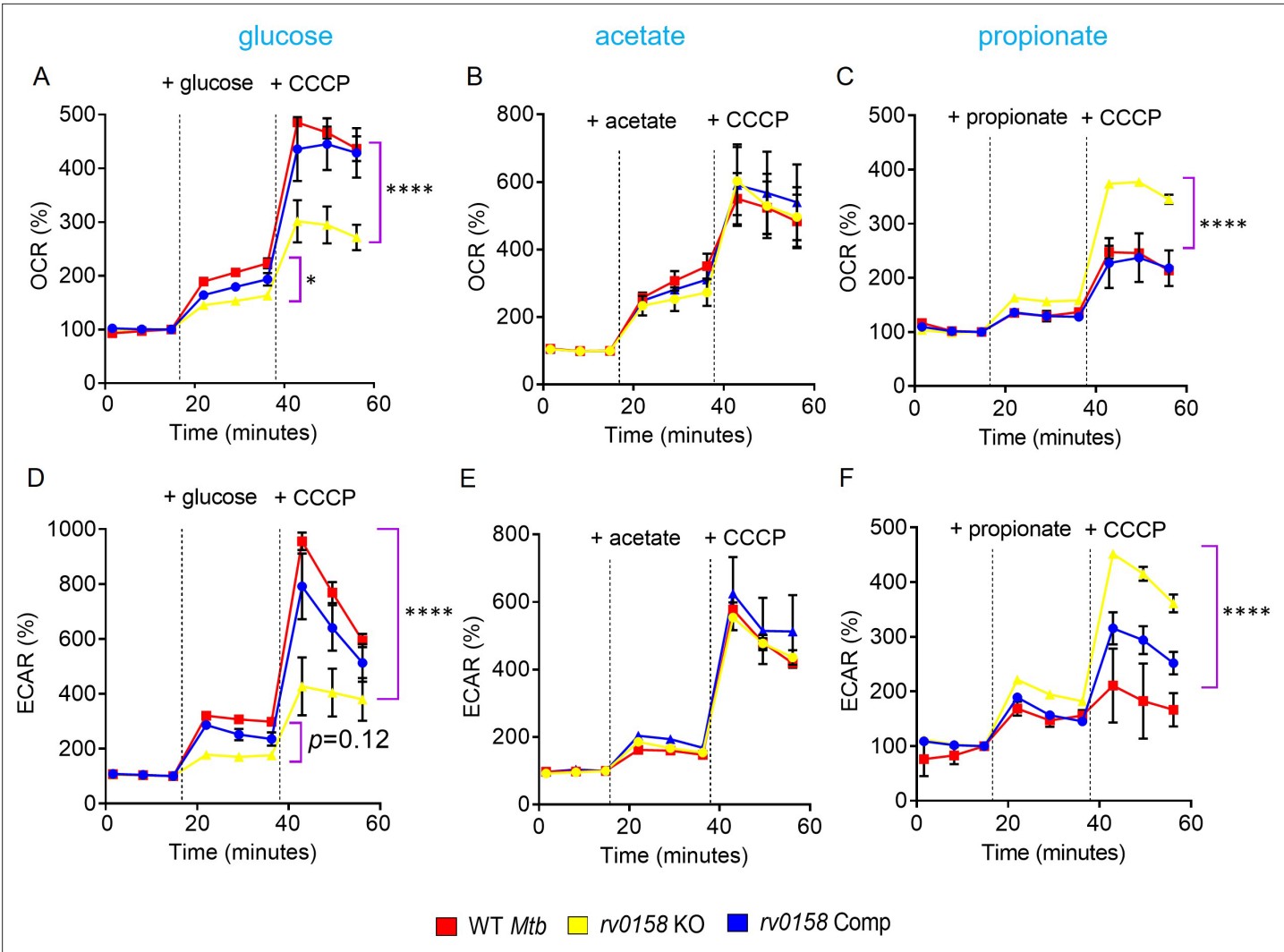

**Figure 8.** Fatty acid metabolites are better substrates for respiration in *rv0158* KO than glucose. ECAR and OCR were measured using the Agilent Seahorse XFp analyzer. ECAR (mpH/min) is an indicator of H⁺ production or extracellular acidification due to glycolytic flux and TCA, and OCR (pmol/min) indicates the rate of oxygen consumption, suggesting the rate of electron flux through the Electron Transport Chain during oxidative phosphorylation. %OCR and %ECAR are displayed as a percentage of third baseline OCR and ECAR values, respectively. Log-phase bacterial cultures were starved overnight, $2 \times 10^6$ cells were seeded on each well, and (**A– C**) %OCR or (**D- F**) %ECAR profiles of WT *Mtb*: red, *rv0158* KO: yellow and *rv0158* Comp: blue were plotted over time. At the indicated times, either 0.3% (final concentration) glucose or 0.3% sodium acetate, or 0.3% sodium propionate (dashed line) was added as the only carbon source, followed by uncoupler- 10 mM CCCP. Results are representative of two independent experiments. The data are means ± SEM. (p-value was calculated with the third data point across groups after the addition of a C source. Another statistical significance was calculated between the first data point obtained across groups after the addition of CCCP; $p < 0.05$: *, $p < 0.0001$: ****, one-way ANOVA with Dunnett's multiple comparisons test).

The online version of this article includes the following source data for figure 8:

**Source data 1.** %OCR and %ECAR numerical values of WT *Mtb*, *rv0158* KO, and *rv0158* Comp.

To support our metabolomics data, we performed qRT-PCR of MCC genes (*icl1*, *prpR*, *prpC*, *prpD*) in WT *Mtb*, *rv0158* KO, and *rv0158* Comp growing on glucose or propionate by qRT-PCR. Consistent with the RNA-seq data, we observed increased expression of MCC genes in *rv0158* KO compared to WT *Mtb* on glucose (**Figure 10A**). In line with the metabolomics data, propionate exposure further increased MCC gene expression in *rv0158* KO compared to WT *Mtb* (**Figure 10B**). In addition to the MCC pathway, *Mtb* mitigates propionate toxicity via the biosynthesis of PDIM (phthiocerol dimycocerosates) (**Lee et al., 2013**; **Figure 10C**). We measured the gene- expression of the PDIM biosynthesis cluster in *rv0158* KO. Expression of PDIM genes was largely unchanged in *rv0158* KO on glucose compared to WT *Mtb* (**Figure 10D**). However, similar to MCC genes, the PDIM cluster was induced

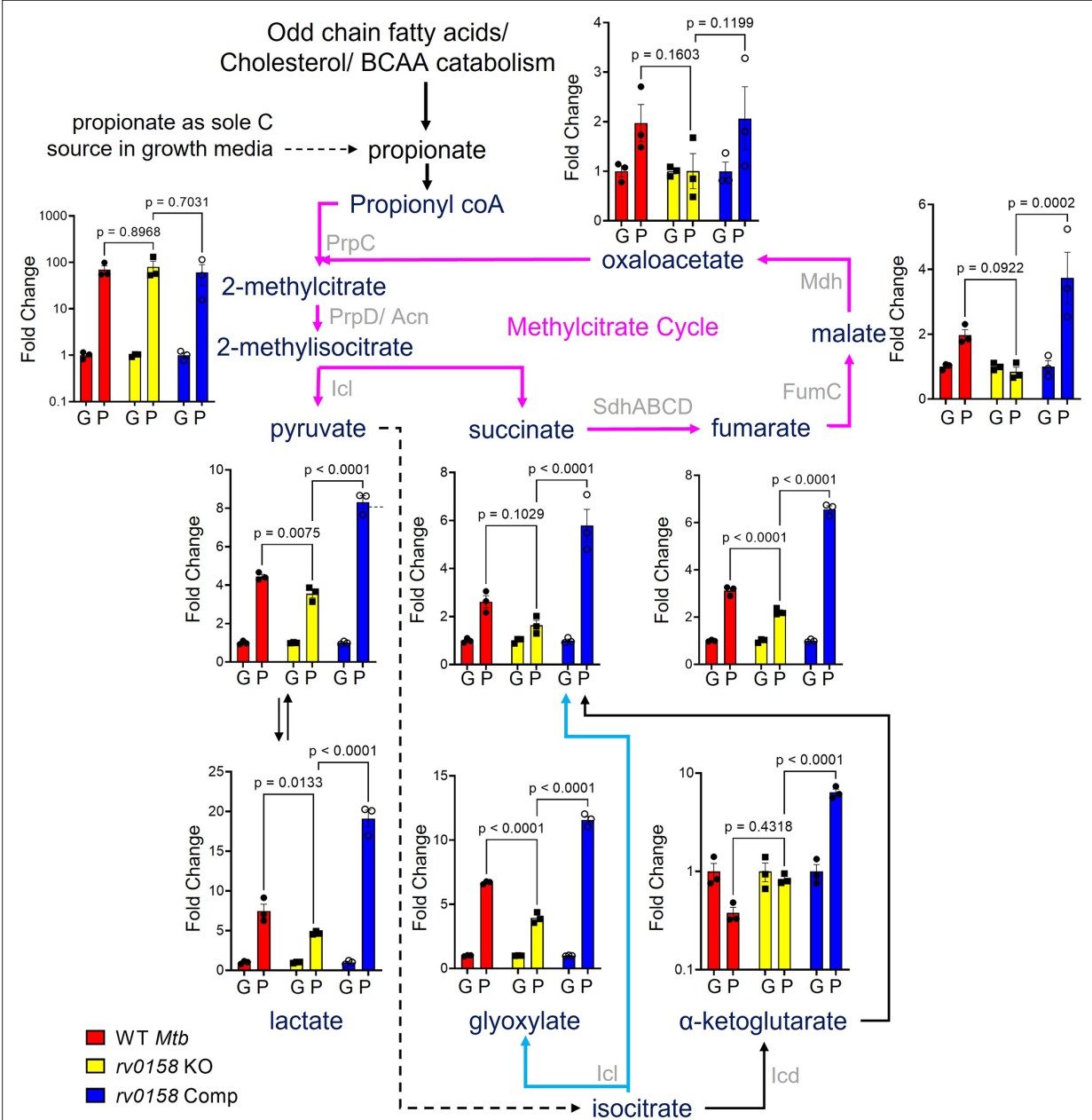

**Figure 9.** Measurement of steady-state metabolite levels in *Mtb* strains growing on either glucose or propionate as C source. Log-phase cultures of *Mtb* strains (Wild-type *Mtb* (WT *Mtb*), *rv0158* KO, and *rv0158* Comp) were synchronized to $OD_{600}$=0.1 and grown in 7H9 broth (no glycerol) containing either 0.3% glucose (**G**) or 0.3% propionate (**P**) as sole C source. Quantitative LC-MS/MS analysis of metabolic intermediates of Methylcitrate cycle (magenta arrows), glyoxylate shunt (Cyan), and TCA cycle. Data are represented as fold change in metabolite levels under propionate conditions relative to that of growing on glucose as C source. (Abbreviations: PrpC- Methylcitrate synthase PrpC, PrpD/ Acn- Methylcitrate dehydratase PrpD/ Methylaconitase, Icl- Isocitrate lyase, Sdh- Succinate dehydrogenase, FumC- Fumarase C, Mdh- Malate dehydrogenase, Icd- Isocitrate dehydrogenase). The data are means ± SEM of two independent experiments. (Statistical significance was calculated by one-way ANOVA with Tukey's multiple comparisons test).

The online version of this article includes the following source data and figure supplement(s) for figure 9:

**Source data 1.** Raw data files used to measure steady-state metabolite levels in *Mtb* strains.

**Figure supplement 1.** Chromatogram of metabolite standards.

**Figure supplement 2.** Chromatogram of oxaloacetate standard.

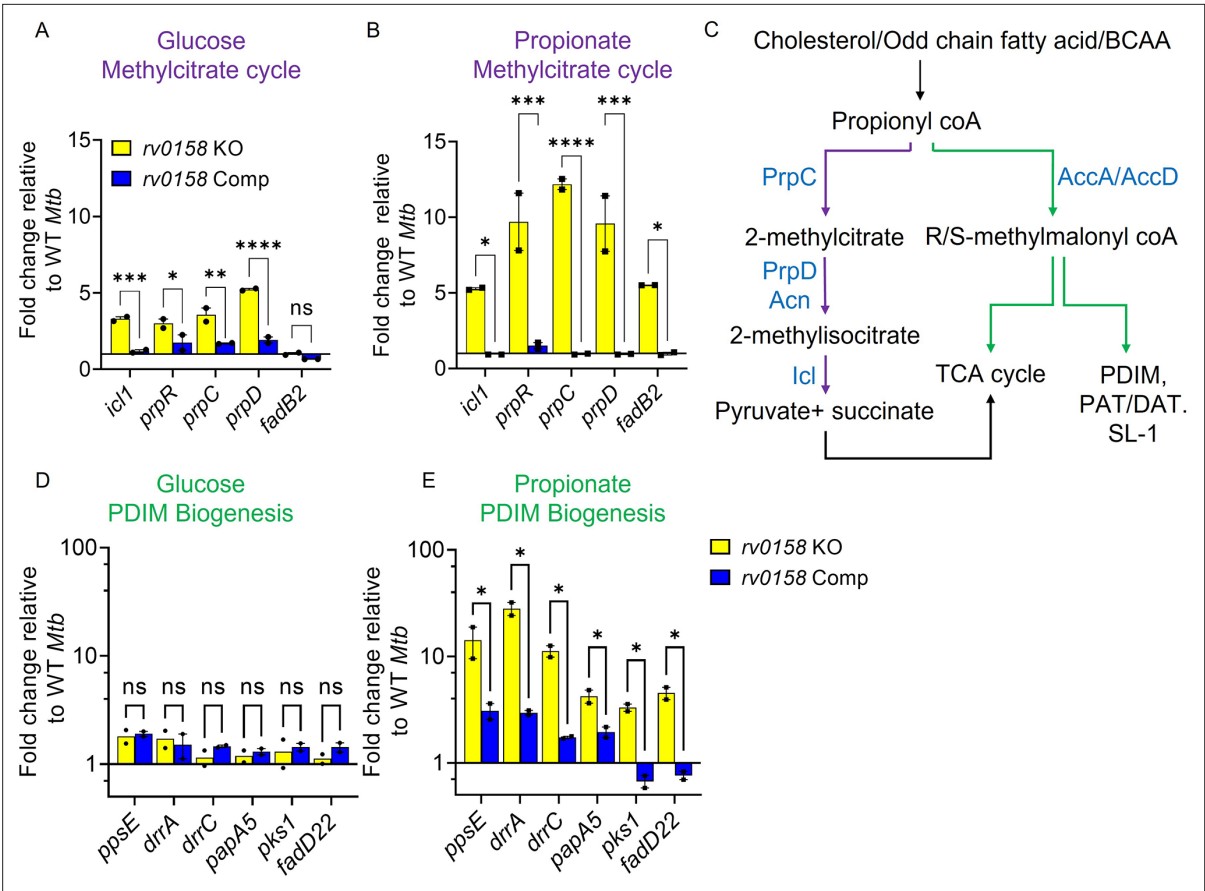

**Figure 10.** Rv0158 Transcriptional regulator is a negative regulator of genes involved in the methylcitrate cycle and biosynthesis of methyl-branched fatty acids. *Mtb* strains (WT *Mtb*, *rv0158* KO, and *rv0158* Comp) were synchronized to $OD_{600\,nm} = 0.1$ and grown in 7H9 broth (no glycerol) containing 0.3% D-glucose or 0.3% sodium propionate to log-phase ($OD_{600\,nm} = 0.5–0.6$) and RNA was isolated to perform qRT-PCR. (**A, B**) C source-dependent change in the RNA levels of genes involved in the methylcitrate cycle and *fadB2* of *Mtb* is mediated by Rv0158, as determined by qRT-PCR. (**C**) Fates of propionyl coA derived from the catabolism of cholesterol or fatty acids. (Abbreviations: PDIM- Phthiocerol dimycocerosates, SL- sulfolipids, PAT/DAT-polyacyltrehaloses/ diacyltrehaloses, PrpC- Methylcitrate synthase PrpC, PrpD/ Acn- Methylcitrate dehydratase PrpD/ Methylaconitase, Icl- Isocitrate lyase, AccA/ AccD-acetyl-/propionyl-CoA carboxylase alpha chain/ acetyl-/propionyl-CoA carboxylase (beta subunit)). (**D, E**) C source-dependent change in the RNA levels of genes involved in the biosynthesis of methyl-branched fatty acids of *Mtb*, is mediated by Rv0158, as determined by qRT-PCR. The data are means ± SEM of two independent experiments. (p>0.05: ns, p<0.05: *, p<0.001: ***, two-way ANOVA with Sidak's multiple comparisons test).

The online version of this article includes the following source data and figure supplement(s) for figure 10:

**Source data 1.** Numerical values indicating fold changes in *rv0158*- dependent expression of genes involved in the methylcitrate cycle and biosynthesis of methyl-branched fatty acids.

**Figure supplement 1.** Biophysical characteristics of Rv0158 protein from *Mtb* H37Rv.

**Figure supplement 1—source data 1.** Raw data files containing SDS-PAGE gel images as shown in ***Figure 10—figure supplement 1b***.

**Figure supplement 1—source data 2.** Numerical values used to plot graph in ***Figure 10—figure supplement 1b***.

**Figure supplement 2.** Rv0158 Transcriptional regulator is a negative regulator of genes involved in Methylcitrate cycle and biosynthesis of methyl-branched fatty acids.

**Figure supplement 2—source data 1.** Numerical values indicating fold changes in *rv0158*-dependent expression of genes involved in the methylcitrate cycle and biosynthesis of methyl-branched fatty acids.

≈10-fold in *rv0158* KO compared to WT *Mtb* on propionate (***Figure 10E***). The expression profile was restored to WT *Mtb* levels in *rv0158* Comp (***Figure 10A–B and D–E***). Overall, the metabolic and expression data suggest that Rv0158 deficiency accelerates propionyl-CoA utilization and reduces the pool of toxic metabolites such as glyoxylate, thereby positively influencing the growth and bioenergetics of *rv0158* KO utilizing propionate as a sole carbon source.

**Table 2.** Binding affinities of small molecule lipid metabolites with Rv0158 protein, as determined by microscale thermophoresis (MST).

| SL. No | Name of molecule | $K_d$ |
|---|---|---|
| 1. | CoA | n.b. |
| 2. | Acetyl-CoA | n.b. |
| 3. | Propionyl-CoA | n.b. |
| 4. | Malonyl-CoA | 2.4±1 µM |
| 5. | Methylmalonyl-CoA | 0.9±0.3 µM |
| 6. | Decanoyl-CoA | 6.4±1.2 µM |
| 7. | C12-CoA | 1.1±0.6 µM |
| 8. | C16-CoA | 1.8±0.5 µM |
| 9. | C18-CoA | 2.8±0.8 µM |

n.b.- No Binding.

The online version of this article includes the following source data for table 2:

**Source data 1.** Raw data files and Microscale Thermophoresis profiles indicating binding of different small molecules with His- tagged Rv0158 protein.

## Rv0158 binds methylmalonyl-CoA and associates with the genomic regions involved in propionate metabolism

To understand the regulatory function of Rv0158, we biochemically characterized the protein. Based on the nucleotide sequence, *the rv0158* gene has been classified into the TetR family of transcriptional regulators (TFTRs) (*Balhana et al., 2015*). In silico investigation of Rv0158 protein revealed significant similarity to the CoA-thioester (3aα-*H*-4α(3-propanoate)–7aβ-methylhexahydro-1,5-indanedione- coA (HIP-CoA)) responsive, TetR-family transcription factor KstR2 (Rv3557c; blastp: identity = 25.52%; Query cover = 88%; E value = 2*10⁻¹³) (*Figure 10—figure supplement 1*; *Ho et al., 2016*). Histidine-tagged Rv0158 elutes as a dimer, and circular dichroism shows that Rv0158 is mainly α-helical with characteristic minima at 209 nm and 222 nm (*Figure 10—figure supplement 1b–e*). Since *rv0158* KO showed differential expression of genes involved in fatty acid metabolism, we employed microscale thermophoresis (MST) to identify ligands for Rv0158. We tested various metabolites generated during the catabolism of long-chain fatty acids, propionate, and cholesterol (*Table 2*). We found that Rv0158 does not bind with CoA, acetyl-CoA, and propionyl-CoA (*Table 2*). However, Rv0158 showed binding with malonyl-CoA, decanoyl -CoA, C12/C16/C18-CoA, with the highest affinity for methylmalonyl-CoA (affinity constant [Kd] of 0.9±0.3 µM) (*Table 2*).

Since methylmalonyl-CoA is a crucial intermediary metabolite regulating propionate flux into MCC, methylmalonyl pathway, and methyl branched lipids (*Savvi et al., 2008*), we proposed that its physical association with Rv0158 likely facilitates regulation of genes associated with propionate metabolism in *Mtb*. To examine this, we performed ChIP-Sequencing (ChIP-Seq) using an anhydrotetracycline (ATc)-inducible flag-tagged Rv0158 (pexCF-*rv0158*; *Minch et al., 2015*). We harvested chromatin samples for ChIP-Seq using anti-FLAG antibody conjugated to magnetic beads and sequenced the crosslinked DNA by Illumina Genome Analyzer. As a negative control, we sequenced the input chromatin before immunoprecipitation using the anti-FLAG antibody. When glucose was used as a sole C-source, Rv0158 binds to the promoter regions (–500 bp to +30 bp) of 70 *Mtb* genes (*Figure 11*, *Supplementary file 5*; Fold change over input control >1.5; *q value* <0.0001). Further classification of these genes indicates that Rv0158 binds to several genes involved in methyl-branched lipid biosynthesis (*pks2, fadD23, rv2958c, rv2959c, rv2961*), lipid transporter (*rv0987-rv0988, mmpL1-mmpS1, mmpL4-mmpS4*), biotin synthesis (*bioF2*), and surface-associated proteins (*esp* operon, *lpqG*, esxQ, *rv1501-rv1507A*).

Under propionate condition, Rv0158 binds to 32 genes (Fold change over input control >1.5; *q value* <0.0001). Out of 32 genes, 24 genes overlap with glucose condition. The promoter regions of the remaining eight genes Rv0158 exclusively bound in propionate- grown *Mtb* were part of the glyoxylate cycle and MCC (*icl1, prpR, and prp* operon) and fadB2 (*Figure 11B and C*, *Supplementary file 5*). As shown earlier, expression of MCC genes was specifically induced in WT *Mtb* growing on propionate (*Figure 10—figure supplement 2*). In contrast to WT *Mtb*, MCC genes were uniformly induced in *rv0158* KO independent of the C source (*Figure 10A and B*). However, propionate resulted in a far greater induction of MCC genes in *rv0158* KO than glucose (*Figure 10A and B*, *Figure 10—figure supplement 2*). Rv0158 did not bind to any PDIM genes, indicating that their higher expression in *rv0158* KO *relative* to WT *Mtb*, specifically on propionate, was an indirect effect. These findings

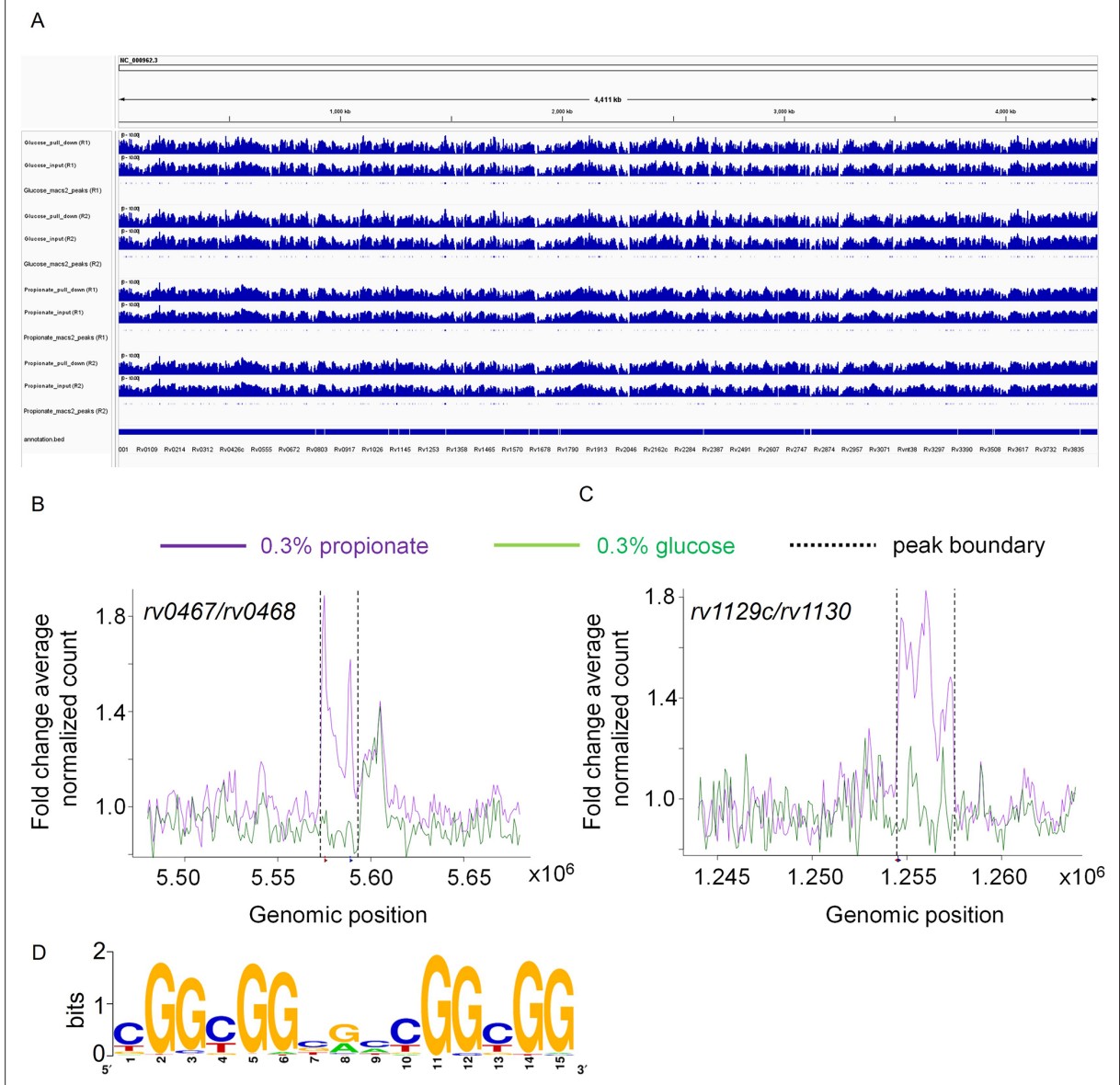

**Figure 11.** Rv0158 Transcriptional regulator binds to the promoter region of genes involved in methylcitrate cycle. 0.3 OD cultures of *rv0158* KO-pexCF-*rv0158 Mtb,* either growing in 0.3% glucose or 0.3% propionate were induced with 12.5 ng/mL of anhydrotetracycline for 24 hr, and DNA-protein binding interactions were determined as mentioned in *Material and Methods*. (**A**) A snapshot from integrative genomics viewer (IGV) showing histogram of log-scaled normalised reads (RPGC normalisation) with *Mtb* H37Rv genome (Accession no.- NC_000962.3) as reference. Horizontal axis shows genomic positions on the reference genome with the bottommost horizontal panel representing genes with positions obtained from the.gff file. Each horizontal panel represents histogram of mapped reads (bin size of 100 base pairs) in pull down sample which is followed by the input sample (without the pull down) and is followed by the representation of peak regions predicted by the macs2. (**B and C**) Line plot representing the distribution of fold change of average normalised reads (CPM) with respect to the genomic position of (**B**) *rv0467/rv0468* peak and (**C**) *rv1129c/rv1130* peak on the X-axis. The purple and green lines represent the propionate and glucose sample conditions. The normalised data (CPM) from both the replicates were averaged and a ratio of averaged normalised data of the pull-down to that of the input samples was calculated for the plot. The *q* values associated with the peaks are 9.9 *10$^{-64}$ for *rv0467/rv0468* peak and 3.4* 10$^{-108}$ for *rv1129c/rv1130* peak, respectively. (**D**) Consensus sequence of Rv0158 binding motif as obtained from MEME-ChIP (e value = 2.6*10$^{-109}$).

The online version of this article includes the following source data for figure 11:

**Source data 1.** Snapshots from IGV showing histogram of log-scaled normalised reads (RPGC normalisation) with NC_000962.3 (*Mtb* H37Rv) as reference genome.

indicate that Rv0158 can bind to methylmalonyl-CoA to adjust the expression of pathways involved in propionate metabolism. However, the direct or indirect nature of this regulation needs future experimentation.

Based on all the sequences enriched in the ChIP analysis under glucose and propionate conditions, the conserved binding motif for Rv0158 was found to be YGGCGGBGMCGGCGG (where Y represents pyrimidines (C/T), B represents G/T/C, and M represents A/C, enrichment e value = $2.6 \times 10^{-109}$; *Figure 11D*). Interestingly, the enriched motif was present close to the binding motifs of other transcription factors involved in regulating MCC (*e.g.* PrpR, GlnR, PhoP, and RamB; *Supplementary file 6*), suggesting overlapping or competing mechanisms to regulate propionate metabolism in *Mtb*.

## *rv0158* KO is susceptible to exogenous oxidative stress and anti-TB drugs *in vitro*, inside macrophages, and *in vivo*

Altered eROS and carbon metabolism influence the lethality caused by exogenous oxidants and antibiotics in *E. coli* (*Brynildsen et al., 2013*). On this basis, we investigated the sensitivity of *rv0158* KO to redox stressors and anti-TB drugs. We found that *rv0158* KO showed greater killing by 10–100-fold to $H_2O_2$, CHP, menadione, and ≈ four fold to the nitric-oxide donor DETA-NO than WT *Mtb* and *rv0158* Comp (*Figure 12A*). Interestingly, the sensitivity of *rv0158* KO towards $H_2O_2$ stress was alleviated when glucose was substituted with fatty acids (0.03% acetate or 0.3% propionate or 0.02% cholesterol) as C source (*Figure 12—figure supplement 1*). This is in line with the restoration of MSH redox state (relative to WT *Mtb*) in the knockout strain when growing with fatty acids as C source (*Figure 7F*).

Given that the mutant showed increased permeability and higher accumulation of isoniazid, we examined if the disruption of Rv0158 affected the intrinsic resistance of *Mtb* to antibiotics. We measured MIC of *rv0158* KO against first-line (isoniazid and rifampicin) and second-line anti-TB drugs (moxifloxacin and bedaquiline) by monitoring the resazurin reduction to resorufin using resazurin microtiter assay (REMA). In the absence of antibiotics, we did not see any difference in the extent of resazurin reduction to resorufin among the untreated *Mtb* strains (*Figure 12—figure supplement 2a*). The minimal inhibitory concentrations (MICs) of anti-TB drugs in the case of *rv0158* KO showed a modest decrease by two fold as compared to WT *Mtb* and *rv0158* Comp (*Table 3*, *Figure 12—figure supplement 2b–d*). We further examined if the mutant is more susceptible to killing by anti-TB drugs. Treatment with 1 X MIC of isoniazid (MIC = 0.06 μg/mL), moxifloxacin (MIC = 0.125 μg/mL), and bedaquiline (MIC = 0.03 μg/mL) for 5 days decreased the survival of *rv0158* KO by >10-fold relative to WT *Mtb* (*Figure 12B*).

Since *rv0158* KO is sensitive to oxidative and nitrosative stress, we examined if Rv0158 is required to maintain intramycobacterial $E_{MSH}$ and survival in immune-activated macrophages. Immune-activation of macrophages elevates phagosomal oxidative and nitrosative stress to induce redox stress in *Mtb* (*Bhaskar et al., 2014*). We infected lipopolysaccharide (LPS), and IFN-γ activated RAW 264.7 macrophages with *Mtb* strains expressing Mrx1-roGFP2 at an MOI of 10, and monitored $E_{MSH}$ and survival of internalized bacteria by FACS and CFU analysis, respectively. We have previously shown that *Mtb* cells inside naïve macrophages diversify into $E_{MSH}$-oxidized, $E_{MSH}$-reduced, and $E_{MSH}$-basal fractions (*Bhaskar et al., 2014*), where immune-activation of macrophages increases the fraction of $E_{MSH}$-oxidized *Mtb* cells (*Bhaskar et al., 2014*). Interestingly, we detected a greater proportion of *rv0158* KO cells in an $E_{MSH}$-oxidized state at 48 hr post-infection (p.i.) than WT *Mtb* and *rv0158* Comp (*Figure 12C*). The increased oxidative shift in the $E_{MSH}$ of *rv0158* KO correlated with bacteriostasis at 2 days p.i. and ≈50% killing at 4 days p.i. (*Figure 12D*). Therefore, our results suggest a critical role of *rv0158* in maintaining redox balance under the stressful environment of activated macrophages.

Based on the results described above, we hypothesized that *rv0158* might be necessary for the pathogen's survival *in vivo*. To test this possibility, we aerosol-infected C57BL/6 J mice with ≈100 CFU of WT *Mtb*, *rv0158* KO, and *rv0158* complemented strains (*Figure 12E*). At 4- and 8 weeks post-infection, the bacterial burden in the lungs of mice infected with *rv0158* KO was ≈2 and ≈10-fold lower than those infected with WT *Mtb*, respectively (*Figure 12F*). Similarly, at 4- and 8 weeks post-infection, bacterial numbers in the spleen decreased by about 3-fold and 28-fold, respectively (*Figure 12G*). The phenotype exhibited by *rv0158* KO was mostly restored in mice infected with the complemented strain (*Figure 12F and G*). The data indicate that *rv0158* is required for the persistence of *Mtb*.

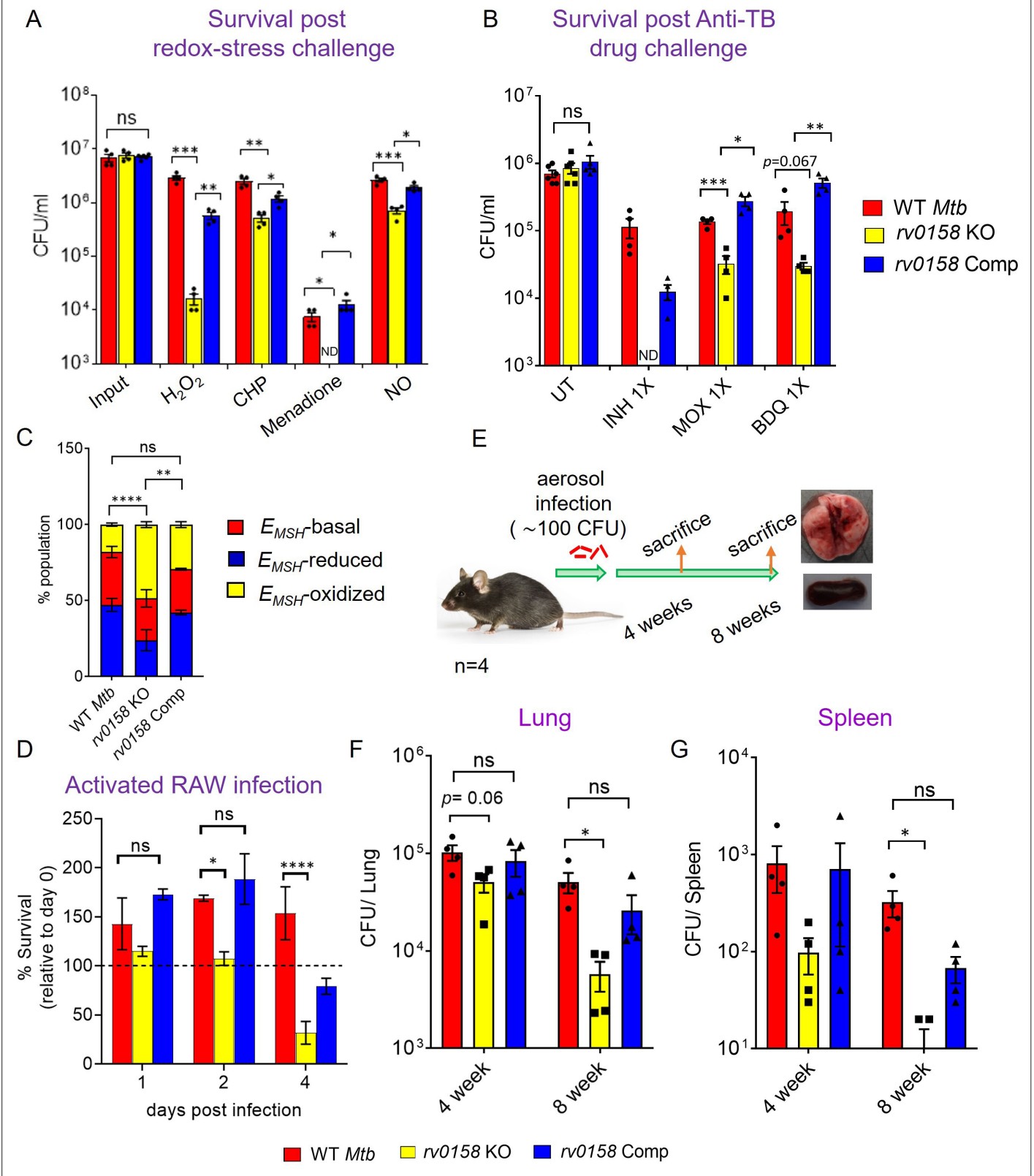

**Figure 12.** *Mtb* requires *rv0158* to counteract exogenous redox stress, anti- TB drugs, and to survive inside macrophages and mice. (**A**) Log-phase *Mtb* cultures were exposed to diverse oxidants ($H_2O_2$=5 mM; CHP = 100 µM; menadione = 0.5 mM; DETA-NO=1 mM) for 8 hours and plated. CFU was determined after 5–6 weeks. (**B**) Viability of bacterial strains was plotted after treatment with isoniazid (INH; 1 X MIC = 0.06 µg/ mL), moxifloxacin (MOXI; 1 X MIC = 0.125 µg/ mL), and bedaquiline (BDQ; 1 X MIC = 0.6 µg/ mL) for 5 days. Bar-graph represents survival (CFUs) of WT *Mtb*, *rv0158* KO, and

*Figure 12 continued on next page*

*Figure 12 continued*

complemented strain. (ND- bacterial colonies not detected at the dilutions plated). (**C, D**) *rv0158* KO *Mtb* is attenuated inside activated macrophages. IFN-γ- and LPS-treated (activated) RAW 264.7 macrophages were infected with Mrx1-roGFP2 expressing WT *Mtb*, *rv0158* KO, and *rv0158* Comp at an MOI of 10. At 48 hr post-infection, (**C**) 10,000 infected macrophages were analyzed by flow-cytometry, intramycobacterial $E_{MSH}$ was measured, and the percentage of bacilli in each redox subpopulation $E_{MSH}$-oxidized (yellow), $E_{MSH}$-basal (red), and $E_{MSH}$-reduced (blue) was determined and plotted as a stacked bar graph as described in Materials and methods. Statistical comparison was performed between $E_{MSH}$-oxidized fractions of *Mtb* strains. (**D**) Intramacrophage survival was monitored by enumerating CFU at 24-, 48-, and 96- hr post-infection. The data are means ± SEM of two independent experiments (n=4). p was determined by two-way ANOVA with Tukey's multiple comparisons test (p>0.05: ns, p<0.05: *, p<0.0001: ****). (**E– G**) *rv0158* is required for *Mtb* survival and persistence in mice. (**E**) Experimental design. C57BL/6 J mice were given a low-dose aerosol challenge (≈100 bacilli) with WT *Mtb*, *rv0158* KO, and *rv0158* Comp and assessed for survival in (**F**) lungs and (**G**) spleen at 4- and 8 weeks post-infection. Data shown are mean ± SEM from 4 mice per group. (p>0.05: ns, p<0.05: *, unpaired two-tailed Student's t-test with Welch's correction).

The online version of this article includes the following source data and figure supplement(s) for figure 12:

**Source data 1.** Numerical values used to plot graphs in *Figure 12A–G*.

**Figure supplement 1.** Fatty acids as C source improves tolerance of *rv0158* KO *Mtb* against exogenous redox stress.

**Figure supplement 1—source data 1.** Numerical values used to plot graph in *Figure 12—figure supplement 1*.

**Figure supplement 2.** Dose response curves of isoniazid, rifampicin, moxifloxacin, and bedaquiline against *rv0158* KO *Mtb*, as determined by REMA.

**Figure supplement 2—source data 1.** Numerical values used to plot graph in *Figure 12—figure supplement 2*.

# Discussion

While *Mtb* is known to tolerate host-induced oxidative stress, it is susceptible to endogenous ROS or RNI levels. In this context, PA-824, an endogenous NO donor to *Mtb*, has been approved for treating drug-resistant TB (as part of the B**Pa**L regimen) (*Conradie et al., 2020*). These findings underscore the importance of identifying genetic factors responsible for keeping eROS levels low in *Mtb*. We integrated high-throughput technologies: Tn mutagenesis, FACS, and Illumina-Sequencing platform with a redox-biosensor (Mrx1-roGFP2) and identified genetic factors required for maintaining basal redox balance in *Mtb*. We noted that during enrichment cycles, few unfit Tn-mutants exhibiting over-whelming oxidative $E_{MSH}$ (e.g. *katG, ahpC, sodC, trxA, trxB1*) might have been outcompeted by rela-tively fit Tn-mutants in the *Mtb* redoxosome. Additionally, certain antioxidant enzymes (*sodA, trxB2*) are essential (*DeJesus et al., 2017*) and, therefore, unlikely to be detected by our method. Mutation in genes responsible for oxidative stress response, turnover of thiol buffers, cell-envelope homeo-stasis, and bioenergetics, which are expected to impart higher eROS accumulation, were part of the redoxosome (*Figure 3*). Interestingly, we also identified factors not directly involved in redox balance including genes associated with RNA processing, transporters and secretion, and amino acid metab-olism. (*Supplementary file 3*). Future experiments are needed to understand how these pathways contribute to redox balance in *Mtb*.

We discovered and characterized a conserved hypothetical gene, *rv0158*, as a critical node with the highest betweenness centrality in the hierarchical redox-homeostasis network. Interestingly, *rv0158* is one of the key transcriptional regulators that genetically interact with *ctpC* (*sodA* hypomorph) in a network to impart oxidative stress resistance during infection in mice (*Nambi et al., 2015*). Addition-ally, the *rv0158* gene was also shown to be upregulated in THP1 macrophages, 0.5% SDS (*Fontán et al., 2008*), and in C57BL/6 mice (*Dubnau et al., 2005*). These studies and our data clearly demanded an in-depth understanding of the *rv0158* function. Our data indicate that *rv0158* deletion resulted in

**Table 3.** Rv0158 contributes to the intrinsic resistance of *Mtb* to anti-TB drugs.
Resazurin Microtiter assay to determine the minimum inhibitory concentration (MIC values) of first-line (Isoniazid and Rifampicin) and second-line (Bedaquiline and Moxifloxacin) anti-TB drugs.

| Strain | $MIC_{90/Day5}$ in µg/mL | | | |
| --- | --- | --- | --- | --- |
| | Moxifloxacin | Bedaquiline | Isoniazid* | Rifampicin |
| WT *Mtb* | 0.12 | 0.03–0.06 | 0.06 | 0.0156 |
| *rv0158* KO | 0.06 | 0.0156 | 0.03 | 0.0078 |
| *rv0158* Comp | 0.12 | 0.03 | 0.06 | 0.0313 |

*MIC determined visually by observing the change of colour from blue to pink.

higher eROS accumulation, rendering the strain more susceptible to exogenous redox stressors and reduced intrinsic resistance to anti-TB drugs. Previous studies with Tn-mutant libraries have shown that *rv0158* confers optimal bacterial fitness in mice model of infection, and the phenotype varies with different mouse genotypes (*Bellerose et al., 2020*; *Smith et al., 2022*; *Zhang et al., 2013*). While this requires further experimentation, our findings largely agree with the requirement of *rv0158* for the persistence of *Mtb* in mice.

*Mtb* isolated from the lungs chiefly catabolizes fatty acids (*Bloch and Segal, 1956*). β-oxidation of fatty acids generates NADH and acetyl-CoA or propionyl-CoA to generate ATP. On the other hand, these lipid intermediates also act as building blocks for cell wall lipids synthesis. Remarkably, despite its importance, the overload of propionyl-CoA and its metabolites adversely affect the survival of *Mtb* (*Eoh and Rhee, 2014*; *Lee et al., 2018*). Therefore, there is a trade-off between the beneficial effects of fatty acid catabolism (its skeleton is a valuable carbon source) and toxicity. To deal with the overload of toxic metabolites, *Mtb* must have a sophisticated way to compartmentalize or calibrate the flux of fatty acid metabolites toward either catabolism or anabolism. The absence of such a control can have two consequences: (1) increased catabolism which then leads to the accumulation of chemically reactive and biologically intoxicating metabolites, such as propionate, glyoxylate, or fumarate (*Dong et al., 2021*), and (2) decreased lipid anabolism that can perturb membrane structure, function, and virulence. Using multiple techniques, our data suggest that *rv0158* regulates *Mtb*'s growth and redox balance on fatty acids such as propionate. Based on our data, we propose that Rv0158 maintains redox balance by regulating the utilization of fatty acids in catabolic (MCC, TCA, and glyoxylate) and anabolic (PDIM, mycolic acids) pathways of *Mtb*.

While *rv0158* KO grew slow in response to glucose, the growth of the mutant was either comparable on fatty acids (palmitate, acetate, and cholesterol) or better (propionate) than WT *Mtb*. Since propionate stunts the growth of WT *Mtb* but not of *rv0158* KO, we begin understanding the role of rv0158 in propionate metabolism. It is known that propionate (or one of its catabolic intermediates, 2-methylcitrate or methylmalonyl-CoA) is lethal to *Mtb* if allowed to accumulate (*Eoh and Rhee, 2014*). Similarly, the accumulation of glyoxylate and disruption of intrabacterial pH homeostasis, redox balance, and membrane potential in the glyoxylate shunt and MCC pathway mutants, resulted in lethality on odd and even chain fatty acids (*Eoh and Rhee, 2014*; *Puckett et al., 2017*). The relatively lesser accumulation of glyoxylate and TCA cycle intermediates in *rv0158* KO, and increased expression of MCC genes (*prpC*, *prpD*, and *icl*) suggest that the mutant is better equipped to prevent accumulation of toxic intermediates when metabolizing propionate. Another consequence of growth on propionate is the undesirable accumulation of NAD(P)H pool, resulting in intrabacterial reductive stress (*Singh et al., 2009*). A reductive shift in the $E_{MSH}$ of *Mtb* on propionate suggests that *Mtb* utilizes mycothiol as a sink to dissipate NAD(P)H-reductive stress. Reductive stress is also mitigated by NADPH-dependent incorporation of propionyl moieties into cell wall polyketide lipids (SL-1, PAT/DAT, and PDIM) and triacylglycerol (TAG) (*Jain et al., 2007*; *Singh et al., 2009*). Propionate cultured *rv0158* KO displayed higher expression of PDIM biosynthesis genes-cluster (*Figure 10*), a more significant reductive shift in $E_{MSH}$ (*Figure 7F*), and efficient respiration (*Figure 8C and F*). All these observations are consistent with the greater adaptability of the mutant on propionate.

While metabolism through MCC and glyoxylate shunt pathway are active on carbohydrates and induced on fatty acid substrates in *Mtb* (*Muñoz-Elías et al., 2006*), the relatively higher expression of *prpR*, *prpC*, *prpD*, *icl1*, and PDIM genes in *rv0158* KO compared to WT *Mtb* on glucose could have resulted in directing a greater flux of glucose carbon away from energy generating TCA cycle/ oxidative phosphorylation to MCC/glyoxylate cycle and energy-requiring polyketide lipid anabolism. The expression of the MCC pathway is activated by a transcription factor PrpR in *Mtb*, whose activity depends on an oxidation-resistant 4Fe-4S cluster (*Tang et al., 2019*). Furthermore, PrpD retained activity under high ROS conditions in other bacteria (*Grimek and Escalante-Semerena, 2004*). Therefore, it is likely that MCC enzymes remained functionally active in *rv0158* KO on glucose despite higher eROS. These metabolic changes are possible reasons behind altered redox balance, respiration slow down, and impaired growth of *rv0158* KO on glucose. The mutant also showed slow growth in media containing a combination of nutrients (glucose, glycerol, and tween-80), impaired resistance to redox stress, antibiotics, and persistence defect in mice. The *in vivo* phenotype of *rv0158* KO is consistent with the fact that *Mtb* persistence relies upon efficient metabolization of glucose, co-catabolism of

multiple nutrients, and resistance to oxidative/nitrosative stress in mice (*Adams et al., 1997*; *Anand et al., 2021*; *de Carvalho et al., 2010*; *Marrero et al., 2013*; *Noy et al., 2016*).

We found that Rv0158 protein strongly binds to methylmalonyl-CoA, an essential intermediate of propionyl-CoA metabolism, coupled with the synthesis of methyl branched lipids (SL-1, PAT/DAT, and PDIM) in *Mtb*. Moreover, our ChIP-seq data revealed that Rv0158 specifically binds to the genomic regions encompassing the *prp* operon, *prpR*, and *icl1* when *Mtb* metabolizes propionate. These findings, along with our gene expression and metabolomics data, indicate that Rv0158 binding to methylmalonyl-CoA might calibrate the flux of propionate into MCC and methyl-branched lipids biosynthesis. Interestingly, probable propionyl-coenzyme A carboxylase genes (*accA2 and accD2*) required for the biotin-dependent conversion of propionyl-CoA to methylmalonyl-CoA are highly induced in *rv0158* KO (six- to sevenfold; *Figure 5B*) and Rv0158 directly binds to gene encoding biotin synthase (*bioF2*). These findings suggest a possible regulatory loop between biotin synthesis, AccA2/D2-dependent methylmalonyl-CoA formation, and propionate metabolism, where Rv0158 may act as an important link between these pathways. The vitamin B12 (Vit-B12) dependent methylmalonyl-CoA pathway is another means by which propionate toxicity can be alleviated in *Mtb* (*Savvi et al., 2008*). While we did not experiment with Vit-B12, the direct and robust binding of methylmalonyl-CoA with Rv0158 protein indicates that Rv0158 could regulate the flux of propionate into methylmalonyl-CoA pathway in *Mtb* during periods of Vit-B12 availability *in vivo*.

The transcription of the *prp* operon is carefully controlled by PrpR. Under propionate-rich conditions, PrpR binds to propionyl-CoA and induces the expression of itself, *prp* operon, and *icl1* to catabolize propionate by MCC (*Tang et al., 2019*). However, PrpR remains in an inactive conformation during growth on carbohydrates resulting in the basal expression of the *prp* operon and *icl1* (*Tang et al., 2019*). The expression of *prpR*, *prp* operon, and *icl1* was uniformly derepressed in *rv0158* KO on glucose and showed hyperinduction on propionate, suggesting that Rv0158 likely functions as a repressor of PrpR. These findings are supported by ChIP-Seq data showing the physical binding of Rv0158 to the *prpR* promoter region under propionate-grown conditions. Since PrpR is an autoinducer and activates the *prp* operon, a repressor of *prpR* that monitors the metabolic flux of propionate by sensing methylmalonyl-CoA might be essential to avoid unnecessary hyperinduction of *prpR* and providing greater control on propionate metabolism in *Mtb*. In addition to propionate catabolism, *Mtb* sequesters propionate via protein propionylation to regulate free propionyl-CoA (*Nambi et al., 2013*; *Singhal et al., 2015*). Therefore, a metabolic intermediate of propionate catabolism such as methylmalonyl-CoA that Rv0158 could sense might be a valuable indicator of propionyl-CoA flux into MCC, methylmalonyl CoA pathway, and methyl branched lipids biosynthesis in *Mtb*. Moreover, propionyl-CoA binding by PrpR and methylmalonyl-CoA binding by Rv0158 could jointly facilitate a more stringent control on the overall consumption of propionate by non-metabolic (propionylation) and metabolic (MCC, polyketide lipids) pathways in *Mtb*.

Although our data suggest that Rv0158 likely senses methylmalonyl-CoA and regulates the expression of pathways associated with propionate metabolism, several issues remain to be addressed. It remains unknown if methylmalonyl-CoA is perceived as a metabolic signal by Rv0158 in *Mtb*, and how this association affects direct interaction with DNA and gene expression. Also, the lack of Rv0158 may induce a physiological state in *Mtb* that is more tolerant to toxicity caused by fatty acids such as propionate. For example, cAMP homeostasis is associated with modulating redox balance, respiration, fatty acid uptake, and fatty acid toxicity in *mycobacteria* (*Ko and Oh, 2020*; *Nambi et al., 2013*; *Nazarova et al., 2019*; *Wong et al., 2023*). Moreover, My*cobacterium bovis* BCG elevates cAMP levels in response to propionate, and phosphodiesterases such as Rv0805 regulate cAMP homeostasis to modulate growth on propionate (*McDowell et al., 2023*). Other evidences suggest that reduced cAMP levels result in overactive fatty acid catabolism (*Nambi et al., 2013*; *Nazarova et al., 2019*) as we observed in the case of *rv0158* KO growing on propionate. While overactive fatty acid catabolism is generally detrimental due to the accumulation of toxic intermediates, mycobacteria have evolved sophisticated regulatory mechanisms (e.g., FdmR) to direct the flux of excess fatty acids away from catabolism to lipid anabolism (*Dong et al., 2021*). These alterations on the mycobacterial surface might be important to counteract toxicity caused by fatty acid-dependent disruption of *the Mtb* envelope (*Kengmo Tchoupa et al., 2022*). We observed a higher expression of the PDIM biosynthesis cluster, altered mycolic acids content, and changes in membrane permeability of *rv0158* KO, suggesting overall changes in the cell envelope of the mutant. Therefore, the physiological effects of

*rv0158* disruption in *Mtb* could result from multiple effectors such as redox, respiration, and cAMP. Also, propionate metabolism is regulated by multiple transcription factors (*e.g.* GlnR, PrpR, RamB, PhoP, WhiB3) in *Mtb* (*Lee et al., 2013*; *Micklinghoff et al., 2009*; *Qi et al., 2021*; *Singh et al., 2009*; *Tang et al., 2019*). Therefore, future experiments are needed to understand the breadth of Rv0158-mediated gene regulation in collaboration with additional regulators and metabolites (*e.g.,* acyl-CoA, malonyl-CoA, and cAMP) in coordinating fatty acid metabolism.

# Materials and methods

## Key resources table

| Reagent type (species) or resource | Designation | Source or reference | Identifiers | Additional information |
|---|---|---|---|---|
| Strain, strain background (*Mycobacterium smegmatis*) | *Mycobacterium smegmatis* mc$^2$155 (*Msm*) | Dr. Kanury V.S. Rao (International Centre for Genetic Engineering and Biotechnology, New Delhi, India). | | |
| Strain, strain background (*Mycobacterium tuberculosis*) | *Mycobacterium tuberculosis* H37Rv (*Mtb*) | Dr. Kanury V.S. Rao (International Centre for Genetic Engineering and Biotechnology, New Delhi, India). | | |
| Recombinant DNA reagent | pMV762-Mrx1-roGFP2 (plasmid) | This paper; *Bhaskar et al., 2014* | | This plasmid can be obtained by sending a request to Dr. Amit Singh. |
| Strain, strain background (mycobacteriophage) | *pHAE180* mycobacterio-phage | Dr. Vinay Nandicoori (National Institute of Immunology, New Delhi, India) | | |
| Strain, strain background (*Mycobacterium tuberculosis*) | *Mtb* H37Rv *mshA* KO | Prof. William R. Jacobs Jr (Albert Einstein College of Medicine, New York, USA). | | |
| Strain, strain background (*Mycobacterium tuberculosis*) | *Mtb* CDC 1551 | BEI resources | | |
| Strain, strain background (*Mycobacterium tuberculosis*) | *Mtb* CDC1551 transposon mutants | BEI resources | | |
| Commercial assay or kit | DNA/RNA oxidative damage ELISA kit | Cayman Chemical | cat. no.- 589320 | |
| Other | CellROX Deep Red | Invitrogen | cat. no. C10422 | Final con-centration- 5 µM. See materials and methods (ROS quantification by CellROX Deep Red dye) |
| Other | [1, 2–14 C] sodium acetate | https://britatom.gov.in/ | product code- LCC34 | Final con-centration- 50 µCi. See materials and methods (Quantification of fatty acids and mycolic acids in *Mtb* strains) |
| Software, algorithm | GraphPad Prism | GraphPad Software (https://www.graphpad.com) | RRID:SCR_002798 | |
| Software, algorithm | ImageJ | ImageJ (http://imagej.nih.gov/ij/) | RRID:SCR_003070 | |
| Software, algorithm | Wave Desktop | Agilent Technologies | RRID:SCR_014526 | |

## Bacterial strains and growth conditions

*Mycobacterium smegmatis* mc$^2$155 (*Msm*) and *Mycobacterium tuberculosis* H37Rv (*Mtb*) were kind gifts from Dr. Kanury V.S. Rao (International Centre for Genetic Engineering and Biotechnology, New Delhi, India). Strain *Mtb*-Mrx1 roGFP2 was generated by transforming *Mtb* H37Rv with an *E. coli*-mycobacterial shuttle vector, pMV762 (*Bhaskar et al., 2014*). Plasmid pMV762 contains the Mrx1-roGFP2 biosensor construct under the control of the *Mtb* hsp60 promoter and a hygromycin-resistance gene as a selection marker. pHAE180 mycobacteriophage was a kind gift from Dr. Vinay Nandicoori (National Institute of Immunology, New Delhi, India). *Mtb* H37Rv *mshA* KO was a kind gift from Prof. William R. Jacobs Jr (Albert Einstein College of Medicine, New York, USA). Wild-type *Mtb* CDC 1551

and the *Mtb* CDC1551 transposon mutants were collected from BEI resources (*Supplementary file 7*; https://www.beiresources.org/). Plasmid pexCF-*rv0158* was a kind gift from Prof. David Sherman (Seattle Biomedical Research Institute, Washington, USA).

All strains were grown in 7H9 broth (BD DIFCO, cat. no.- 271310) supplemented with 0.2% glycerol, 0.1% Tween-80, and ADS (0.5% albumin VWR amresco Life sciences, cat. no. –0332–500 G), 0.2% dextrose, and 0.085% NaCl) with shaking at 180 RPM in a rotary shaker incubator (Lab Therm LT-X; Kuhner, Basel, Switzerland) or on 7H11 agar (BD DIFCO, cat. no.- 212203) supplemented with ADS or OADC (ADS plus 0.05% oleic acid (Sigma Aldrich, cat. no.- O1383). and 0.004% catalase (Sigma Aldrich, cat. no.- C9322) at 37 °C. Bacterial strains expressing Mrx1-roGFP2 were grown in media containing hygromycin. Hygromycin (MP Biomedical, cat. no.- 0219417091) and kanamycin (SRL, cat. no.- 25389-94-0) were used at a final concentration of 50 µg/mL and 25 µg/mL, respectively. To measure bacteria's response to oxidative stress, assays were performed in the 7H9 medium supplemented with a catalase-negative supplement (10% albumin-dextrose-sodium chloride (ADS)).

Protocol for carbon-utilization experiments was followed as described in previous studies (*Jain et al., 2007*; *Muñoz-Elías et al., 2006*). Briefly, *Mtb* strains were precultured to OD$_{600 nm}$ of ≈1.0, washed with 1 X PBS, and resuspended in fresh medium media containing 0.47% 7H9, 0.5% albumin, 0.085% NaCl, and 0.05% tyloxapol (MP Biomedical, cat. no.-157162) supplemented with 0.3% (w/v) D-glucose (Thermo Fisher scientific, cat. no.- Q24415) or 0.3% (w/v) sodium acetate (SRL, cat. no.-12709–3), 0.3% (w/v) or sodium propionate (Sigma Aldrich, cat. no.- P1880) or 0.02% (w/v) cholesterol (Sigma Aldrich, cat. no.- C3045) or 200 µM sodium palmitate (Sigma Aldrich, cat. no.- P9767-10G), followed by incubation at 180 RPM in a rotary shaker incubator.

## Whole-genome transposon mutagenesis

*Mtb* H37Rv-Mrx1 roGFP2 strain was grown till OD$_{600 nm}$ ~1. The cells were washed twice with wash medium (7H9 broth +5% glycerol) and resuspended in mycobacteriophage buffer (50 mM Tris-HCl pH 7.5, 150 mM NaCl, 10 mM MgSO$_4$, and 2 mM CaCl$_2$). These phage-competent cells were infected with *phAE180* ($10^{10}$–$10^{11}$ PFU/mL) at MOI of 1:10 for 3 hours at 37 °C followed by the selection of phage-transduced *Mtb* cells on 7H11+OADC plates containing hygromycin and kanamycin and incubated at 37 °C for 4–5 weeks.>$10^5$ colonies were scrapped, pooled together, and stored at –80 °C as frozen glycerol stocks.

## Transposon-directed Insertion site Sequencing (TRADIS) and analysis

Genomic DNA for TraDIS (TnSeq) protocol was isolated as previously described (*Majumdar et al., 2017*). The quality of isolated genomic DNA was checked by agarose gel electrophoresis. Phage DNA contamination in isolated genomic DNA was checked by PCR using primers to amplify the scaffold region of TM4 phage: TM4-ScaF: 5′ATGGCAGAACAAACTGAG3′ and TM4-ScaR: 5′GAATTGGT GTTGCCGTT G3′. TraDIS (TnSeq) protocol was utilized to identify the site of mutation (insertion by transposon) in the library of *Mtb*-transposon mutants (*Barquist et al., 2016*). Briefly, genomic DNA was sheared to produce ≈ 550 double-stranded DNA fragments using Covaris S220 and confirmed by Agilent Bioanalyzer (2100 High sensitivity). Using TruSeq Nano DNA LT Library Preparation Kit – Set A (12 Set A index tubes, 24 samples)-FC-121–4001 and MiSeq Reagent Kit v2 (50 cycle)-MS-102–2001, we performed Tn sequencing by Illumina Miseq sequencing platform as described by the manufacturer and in reference (*Barquist et al., 2016*). PCR-enrichment of DNA fragments containing transposon-chromosome junctions was done with a 3′ transposon sequence-specific primer 5′AATGATACGGCG ACCACCGAGATCTACACTCTGGGGTACGCGTAATACGACTCACTATAGGGTCTAGAGA3′ and an Illumina indexed-adapter specific primer. The following sequencing primer was used: 5′TAATACGACTCA CTATAGGGTCTAGAGACCGGGGACTTATCAGC3′. Raw reads (single end; read length = 50 bp; read quality ≥ Q20) were obtained for two biological replicates, each for the Tn-Library and $E_{MSH}$-oxidized Tn mutant pool of *Mtb* as.fastq files. The reference genome sequence (.fna) and annotation (.gff) files for the same strain were downloaded from the NCBI ftp website (https://ftp.ncbi.nlm.nih.gov/). The reads were mapped to the *Mtb* H37Rv genome (accession number: NC_018143.2) using Burrows-Wheeler Aligner (BWA; *Li and Durbin, 2009*). The raw read quality was checked using the FastQC software (version v0.11.5). BWA (version 0.7.12-r1039) was used to index the reference genome and align the reads. SAMTOOLS (version 1.2) was used to filter out the multiply mapped reads. BEDTOOLS (version 2.25.0) was used to calculate the reads count per gene using the annotation file (.bed). The format of

the annotation file (.gff) was changed to.bed using an in-house python script. The normalization and transposon enrichment analysis for the two conditions was carried out using the edgeR (*Robinson et al., 2010*) and DESeq2 (*Love et al., 2014*) pipeline. The p-value of differential gene representation analysis was obtained through DESeq2 and edgeR after correction for multiple hypothesis testing using false discovery rate (p.adj. or FDR at a cut-off <0.05). The genes (total number) that showed differential insertions (log$_2$ fold change $\geq$ 1 and FDR or p.adj. <0.05) in either of the pipelines were considered enriched for insertions in $E_{MSH}$- oxidized Tn mutant pool relative to the input Tn Lib. The union of over-represented genes derived from DESeq2 and edgeR is considered '*Mtb* redoxosome'. The TraDIS data has been submitted to NCBI's BioProject (Submission ID: SUB11081305; BioProject ID: PRJNA807454).

## Fluorescence-activated cell sorting (FACS) analysis of bacterial population

Bacteria were grown to OD$_{600\ nm}$ = 0.6–0.8 in OADC-supplemented 7H9 media at 37 °C, 180 rpm. Prior to experiments, cells were harvested and resuspended in 7H9 media containing ADS. Flow-cytometry using a BD FACS Aria flow cytometer equipped with the 405 nm and 488 nm lasers was performed to determine the ratio of Mrx1-roGFP2 fluorescence within *Mtb* cells at 510 nm emission. Ratiometric changes (Ex: 405/488 nm) upon 10 mM cumene hydroperoxide (maximum biosensor oxidation) and 20 mM dithiothreitol (maximum biosensor reduction) treatment to WT *Mtb* were utilized to gate ambiently $E_{MSH}$-oxidized or $E_{MSH}$-reduced *Mtb* transposon mutants. 10,000 events per sample were analyzed. For enriching this population, cells were sorted using BD FACS Aria with a four-way purity sort setting. The oxidized $E_{MSH}$ of sorted cells was confirmed by post-sort analysis and plated on 7H11 supplemented with OADC and allowed to grow at 37 °C for 4–5 weeks. The intracellular $E_{MSH}$ measurements were done as described previously (*Bhaskar et al., 2014*).

## Quantification of 8-OH deoxyguanosine (DNA damage marker)

Bacterial cultures were grown to the mid-log phase (OD$_{600\ nm}$ = 0.6). Genomic DNA was isolated from each sample. 8-OH deoxyguanosine species were quantified using ELISA according to the manufacturer's instructions (Cayman chemical-DNA/RNA oxidative damage ELISA kit, cat. no.- 589320).

## Lipid peroxidation assay

Bacterial cultures were grown to the mid-log phase (OD$_{600\ nm}$ = 0.6). Lipid hydroperoxides were quantified from cell pellets using FOX2 reagent (*Griffiths et al., 2000*; *Nambi et al., 2015*). Briefly, the pellet was resuspended in 1:2 chloroform/methanol and mixed by vortexing. Next, chloroform and water were added to it at 1:1. The samples were then centrifuged to separate phases. The organic phase was collected and washed twice with water. A total of 200 µl of the organic phase was incubated with 1 mL of FOX2 reagent in the dark for 1 hr at 22 °C. Lipid hydroperoxides were measured spectrophotometrically at 560 nm and normalized with cell mass or OD$_{600\ nm}$.

## Determination of bacterial survival

Bacterial cultures were grown to the mid-log phase (OD$_{600\ nm}$ = 0.6). Bacterial clumps were removed by centrifugation at 200 g for 5 minutes. The single-cell suspension was then set at 0.1 OD$_{600\ nm}$. Oxidants (Hydrogen peroxide Merck, cat. no.- 1.93408.0521), cumene hydroperoxide (Sigma Aldrich, cat. no.- 247502), diamide (Sigma Aldrich, cat. no.- D3648), menadione (Sigma Aldrich, cat. no.- M5625) and Diethylenetriamine/nitric oxide adduct (DETA-NO; Sigma Aldrich, cat. no.- D185) at indicated concentration were added to the culture, and CFU was determined after 8 or 24 hr of treatment. *In vitro* growth kinetics of bacterial strains were determined by removing 200 µL aliquots at various intervals, serially diluted, and plated on 7H11 agar for CFU enumeration. The killing of bacterial strains in the presence of anti-TB drugs (Isoniazid (Sigma Aldrich, cat. no.- 13377), Rifampicin (Sigma Aldrich, cat. no.- R3501), moxifloxacin (Sigma Aldrich, cat. no.- PHR1542) and bedaquiline (Cayman chemical, cat. no.- 20247)) was quantified by treating bacteria with antibiotics as described in Resazurin Microtiter Assay (REMA) protocol (*Padiadpu et al., 2016*). At day 5 post-treatment with drugs, aliquots were taken from bacteria-containing wells, serially diluted, and plated on drug-free agar for CFU enumeration. Plates were incubated for 4–5 weeks at 37 °C, and CFU was counted.

### Determination of minimal inhibitory concentration (MIC)

MIC was determined by a resazurin microtiter assay (REMA) using 96-well flat-bottom plates (*Padiadpu et al., 2016*). *Mtb* strains were cultured in 7H9+ADS medium and grown to exponential phase (OD$_{600 nm}$ = 0.4 to 0.8). Approximately 1×10$^5$ bacteria per well were added in a total volume of 200 µL of 7H9+ADS medium. Wells lacking *Mtb* served as a control. Additional controls consisted of wells containing cells (growth control). After 5 days of incubation at 37 °C in the presence of anti-TB drugs, 30 µL of 0.02% resazurin (Sigma-Aldrich, cat. no.- R7017) was added, and plates were incubated for an additional 24 hr. Fluorescence intensity was measured using a SpectraMax M3 plate reader (Molecular Devices) in the bottom-reading mode with excitation at 530 nm and emission at 590 nm. Percent inhibition was calculated from the relative fluorescence units compared with an untreated control culture; MIC was taken as the lowest drug concentration that resulted in at least a 90% reduction in fluorescence compared to untreated growth control.

### *In vitro* oxidative burst assay

Bacteria were grown in OADC-supplemented 7H9 media at 37 °C, 180 rpm. Prior to experiments, cells were harvested and resuspended in 7H9 supplemented with ADS. These cells were treated with 12.5 µM cumene hydroperoxide, and the Mrx1-roGFP2 biosensor ratio was measured by flow-cytometry to investigate the redox response of cells as a function of time. 10,000 events per sample were analyzed.

### Isolation of RNA and quantitative RT-PCR analysis

Bacterial strains were grown to the mid-log phase (OD$_{600 nm}$ ≈ 0.6). 25 mL GTC buffer (5 M guanidium thiocyanate (VWR Life sciences, cat. no. -VWRC0380), 0.5% sarcosyl, 0.5% Tween-80, 1% β mercaptoethanol (freshly added)) was added to 10 mL of bacterial culture, mixed, and cells were immediately harvested. Total RNA was extracted as described previously using MP Biomedicals FastRNA Pro Blue Kit (MP Biomedical, cat. no.- 116025050) and QIAGEN Rneasy Kit (cat. no.- 74136; *Chawla et al., 2012*). DNA contamination was removed from the isolated RNA using a TURBO DNA-free Kit (Invitrogen, cat. no. -AM1907), and 600 ng of RNA was used for cDNA synthesis using the iScript cDNA Synthesis Kit (BioRad, cat. no.- 170–8891). Quantitative RT-PCR is done with CFX96 RTPCR System (BioRad) using gene-specific primers (*Supplementary file 8*) and iQ SYBR Green Supermix (Bio-Rad, cat. no.- 1708886). Expression of genes was normalized with C$_t$ value for 16 s rRNA (internal housekeeping control).

### Generation of redox-homeostasis network

We used the *Mtb* Transcription Regulatory Network (TRN) consisting of 156 Transcription factors (TFs), 3679 target genes, and 15,980 protein-DNA interactions based on Chromatin Immunoprecipitation followed by Next-generation Sequencing (ChIP-Seq) data (*Minch et al., 2015*) to generate a 'redox-homeostasis network' of 107 genes (including 14 TFs) with 143 interactions (*Figure 3C*). This redox-homeostasis network comprises of the subset of 143 edges in the *Mtb* TRN wherein both the TF and the target gene of an interaction are part of the redoxosome of 368 genes which are overrepresented in the E$_{MSH}$- oxidized Tn mutant pool. The redox-homeostasis network has three connected components with a dominant largest component comprising of 102 genes (including 12 TFs) and 139 edges (*Figure 3C*). Moreover, the TFs and target genes in the largest component of the redox-homeostasis network were organized into three hierarchical levels as follows. The top layer comprises of 6 TFs that are not regulated by any other TF in the network (i.e. have in-degree 0 after omitting self-loops), the middle layer comprises of 6 TFs that are regulated by at least one other TF in the network, and the bottom layer comprises of the remaining 90 target genes. Thereafter, we computed the betweenness centrality, out-degree, and degree of collaboration (*Bhardwaj et al., 2010*) for the TFs in the redox-homeostasis network. In the 'redox-homeostasis network' shown in *Figure 3C*, we differentiated the genes based on their functional category obtained from the MycoBrowser database (*Kapopoulou et al., 2011*).

## Generation of a knockout strain of *rv0158* in *Mtb* H37Rv and genetically complemented strains

Knockout strains were generated by the Allelic-exchange strategy (*Chawla et al., 2018*). Approximately 1 kb up and down flanking DNA sequences of the genomic locus of *rv0158*-ORF were amplified and cloned upstream and downstream, respectively, of the *loxP-hyg-gfp-loxP* (*hyg*: hygromycin resistance gene, *gfp*: Green fluorescent protein-expressing gene) cassette in the mycobacterial suicidal vector pML523 (a kind gift from Dr. Michael Niederweis, University of Alabama, Birmingham). The complete construct of 'flanking sequences along with the *hyg-gfp* DNA sequence' was cloned into the pRSF-duet vector (Clontech). UV-treated pRSF-*rv0158* was electroporated into WT *Mtb* H37Rv for allelic exchange as described previously (*Chawla et al., 2018*). In a double crossover recombination event, the entire ORF of interest in the *Mtb* genome was replaced by up and down fragments flanking the *rv0158* gene along with the *loxP-hyg-gfp-loxP* cassette. Colonies were screened for knock-out strains (only Hyg-resistant) by antibiotic selection and quantitative RT-PCR (qRT-PCR). The knockout strains were unmarked using the pCre-Zeo-SacB plasmid (a kind gift from Dr. Amit K. Pandey, Translational Health Science and Technology Institute, Haryana, India). For generating complemented strains, *rv0158* ORF and upstream 400 bp promoter sequence were cloned into pCV125 (*Alland et al., 2000*), an integrative *E. coli* -Mycobacterial shuttle vector. This construct was then electroporated into the unmarked knockout strain; colonies were selected on 7H11+kanamycin plates, and the complemented strain was validated with qRT-PCR.

## ROS quantification by CellROX Deep Red dye

Bacterial strains were grown to the mid-log phase ($OD_{600\,nm} \approx 0.6$). Cells were harvested by centrifugation (4200 *g* for 5 min) and resuspended in 100 µL of growth medium. As per the manufacturer's instructions, CellROX Deep Red (Invitrogen, cat. no.- C10422) was added to a final concentration of 5 µM, and cells were agitated on a rocker (Biobee Tech) for 30 min at 37 °C. After incubation, cells were washed to remove residual dye by centrifugation (4200 *g* for 5 min). Cells were resuspended in 300 µL phosphate-buffered saline (pH 7.4) and then fixed by adding 4% paraformaldehyde (PFA) for 1 h at room temperature. Fluorescence of 10,000 events per sample was measured at a fixed emission (670 nm) after excitation with a red laser (640 nm) using a BD FACSVerse Flow cytometer (BD Biosciences). 10,000 events per sample were analyzed.

## Quantification of DNA strand breaks by TUNEL assay

DNA damage was measured by using *In Situ* Cell Death Detection Kit, TMR red (Roche Molecular Biochemicals, Indianapolis, IN, Cat. No.- 12156792910), which is based on TUNEL (TdT-mediated dUTP-X nick end labeling) assay (*Shee et al., 2022*). An equal number of cells (based on $OD_{600}$) were taken, washed once by centrifugation, and fixed in 2% paraformaldehyde (PFA). PFA was removed by washing cells, followed by resuspension in 2% sodium dodecyl sulfate (SDS) and a second wash by centrifugation. DNA double-strand breaks were labeled in 100 µl of TUNEL reaction mix for 3–4 hr. Cells incubated with label solution only (no terminal transferase) were used as negative controls. Fluorescence was measured at a fixed emission (585 nm) after excitation with green-yellow laser (561 nm) using a BD FACSAria Flow cytometer (BD Biosciences, San Jose, CA). A total of 10,000 events were acquired per sample.

## Preparation of RNA for RNA sequencing by Illumina HiSeq sequencing platform

Bacterial strains were grown to the mid-log phase ($OD_{600\,nm} \approx 0.6$). Total RNA was extracted as described above. Ribosomal RNA (16 s and 23 s rRNA) was depleted from RNA samples using MICROBExpress Bacterial mRNA Enrichment Kit (Life Technologies, cat. no.- AM1905). RNA was quantified using Qubit 2.0 Fluorometer (Invitrogen). rRNA depleted RNA ($\approx$ 15 ng per sample) was fragmented and then used to prepare the library using NEBNext Ultra II Directional RNA Library Prep Kit for Illumina (New England Biolabs, cat. no.-E7765S) and NEBNext Multiplex oligos for Illumina (New England Biolabs, cat. no.-E7335S), according to the manufacturer's protocol. The fragment-size distribution, uniformity, and quality were examined using a high-sensitivity DNA Chip in Agilent Bioanalyzer (2100 High sensitivity). Two nM of cDNA libraries were pooled together and sequenced using the HiSeq2500 platform (Illumina).

## Analysis for differential gene expression with RNA Sequencing data

Raw reads (single end; read length = 50 bp; read quality ≥ Q20) were obtained for two biological replicates as.fastq files. The reference genome sequence (.fna) and annotation (.gff) files for the same strain- *Mtb* H37Rv were downloaded from the NCBI ftp website. The reads were mapped to the *Mtb* H37Rv genome (accession number: NC_000962.3) using Burrows-Wheeler Aligner (BWA) (*Li and Durbin, 2009*). The raw read quality was checked using the FastQC software (version v0.11.5). BWA (version 0.7.12-r1039) was used to index the reference genome and align the reads. SAMTOOLS (version 1.2) filtered out the multiply mapped reads. BEDTOOLS (version 2.25.0) was used to calculate the reads count per gene using the annotation file (.bed). The format of the annotation file (.gff) was changed to.bed using an in-house python script. The normalization and differential analysis for the two conditions were carried out using the edgeR pipeline. The *p*-value of differential expression analysis was obtained through edgeR after correction for multiple hypothesis testing using a false discovery rate (FDR <0.05). Genes with absolute fold change ≥1.5 or≤−1.5 and FDR <0.05 were considered differentially expressed in comparative groups. Raw data has been submitted to NCBI's Gene Expression Omnibus (GEO, accession number- GSE196844).

## Measurement of cellular bioenergetics by Seahorse XF-flux analyzer

To measure oxygen consumption rate (OCR) and extracellular acidification rate (ECAR), log-phase *Mtb* cultures were briefly (one day) incubated in 7H9 medium containing the non-metabolizable detergent tyloxapol and lacking ADS or a C-source. These cultures were then passed 10 times through a 26-gauge syringe needle followed by centrifugation at 100 *g* for 1–2 min to remove clumps of bacterial cells. The resulting single-cell suspensions of bacteria at $2\times10^6$ cells/well were placed in the bottom of wells of a Cell-Tak (Corning, cat. no.- 354240)-coated XF culture plate (Agilent/Seahorse Biosciences). To confirm that an equal number of cells were seeded on the plate, *Mtb* strains were plated on 7H11 agar, and CFU was enumerated after 4 weeks. Measurements were performed using a Seahorse XFp analyzer (Agilent/Seahorse Biosciences) with cells in an unbuffered 7H9 growth medium (pH 7.45 lacking monopotassium phosphate and disodium phosphate). OCR and ECAR measurements were recorded for ~21 min (3 initial baseline readings) before the addition of either glucose (0.3% w/v final concentration) or sodium acetate (0.3% w/v) or sodium propionate (0.3% w/v) as a C-source, which was delivered automatically through the drug ports of the sensor cartridge (Wave Software, Agilent Technologies). Carbonyl cyanide 3-chlorophenylhydrazone (CCCP; Sigma-Aldrich, cat. no.- C2759) at 10 µM final concentration was similarly added through drug ports as indicated in figures. Changes in OCR (% OCR) and ECAR (% ECAR) readings triggered by C-source or CCCP were calculated as a percentage of the third baseline reading for OCR and ECAR taken before C-source injection.

## Quantification of fatty acids and mycolic acids in *Mtb* strains

Metabolic radiolabeling of mycolic acids and fatty acids of *Mtb* was performed as previously described (*Singh et al., 2009*; *Wilson et al., 2013*). Briefly, *Mtb* was cultured to $OD_{600\,nm}$ = 0.8 in 50 mL of 7H9 medium, followed by the addition of 50 µCi of [1, 2-$^{14}$C] sodium acetate (specific activity = 56 mCi/mmole; product code- LCC34, https://britatom.gov.in/) and incubating for another 24 hr. Bacteria were harvested after washing with 1 X PBS and sterilized by autoclaving. To the cell pellets, 2 mL of tetra-n-butylammonium hydroxide (TBAH; Spectrochem Pvt. Ltd., cat. no.-0120224) was added and incubated overnight at 100 °C (silicone oil bath) to hydrolyze all lipids. Fatty acids were esterified by adding 4 mL dichloromethane (SDFCL, cat. no.- 20340), 300 µL iodomethane (SDFCL, cat. no.-38585), and 2 mL Milli-Q water and mixing at room temperature on a rocker for 1 h. After centrifugation at 4200 *g* for 10 min, the lower organic phase was washed twice with Milli-Q water, dried in fumehood overnight, and resuspended in 3 mL of diethyl ether (SDFCL, cat. no.- 20094). This was sonicated in a water-bath sonicator for 30 min for uniform resuspension of lipids. FAMES/MAMES were harvested after centrifugation and transfer of supernatant into a fresh glass vial, dried, and lipids were resuspended in 200 µL dichloromethane. Equal counts (50,000 CPM) were loaded on a silica gel 60 F254 thin-layer chromatography (TLC) plate (Sigma Aldrich, cat. no.- 1055540007) and resolved using hexane/ethyl acetate (19:1, v/v). To determine the distribution of FAMEs, the same sample was analyzed on a C18 reverse-phase TLC (Sigma Aldrich, cat. no.- 1055590001) and developed with acetonitrile/dioxane (1:1, v/v). The TLCs were visualized using a phosphorimager (GE Typhoon phosphorimager).

### $^{14}$C-Inh accumulation assay

Log-phase *Mtb* cultures (OD$_{600 \text{ nm}}$ ≈ 0.6) were washed twice with ice-cold PBS and resuspended in PBS. An equal number of cells ($10^9$) for each strain were treated with [$^{14}$C]-Isoniazid (ViTrax Inc) at a concentration of 0.5 μCi/mL. After 1 hr of treatment, bacteria were pelleted and extensively washed to remove residual [$^{14}$C]-isoniazid. The intracellular level of isoniazid was determined by bacterial lysis using bead-beating. Lysates were added to 1,4-Bis(5-phenyl-2-oxazolyl)benzene solution in toluene (or POPOP luminophore, Sigma), and radioactivity was measured by a liquid scintillation counter (LKB WALLAC 1209 RACKBETA liquid scintillation counter).

### Ethidium bromide accumulation assay

Cell-membrane permeability of the *Mtb* cells was determined as previously described (*Dong et al., 2021*), with minor modifications. Briefly, log-phase bacterial cultures were washed twice with PBS and resuspended in PBST (PBS + Tween 80). An equal number of cells ($10^8$) for each strain were added into the 96-well plate. Ethidium bromide (1 μg/ mL, Sigma Aldrich, cat. no.- 1.11608) was added to the cells, and fluorescence intensity was measured for 1 hr using a SpectraMax M3 plate reader (Molecular Devices) in the bottom-reading mode with excitation at 530 nm and emission at 590 nm.

### Quantification of cellular NAD$^+$ and NADH

*Mtb* H37Rv strains were grown to log phase (OD$_{600 \text{ nm}}$ = 0.6), and pyridine nucleotides (NAD$^+$ and NADH) levels were determined by a redox-cycling assay (*Singh et al., 2009*; *Vilchèze et al., 2013*). Briefly, 10 mL of each culture was harvested, washed once with 1 X PBS, and treated with 0.2 M HCl (1 mL, NAD$^+$ extraction) or 0.2 M NaOH (1 mL, NADH extraction). The samples were heated at 55 °C for 20 min, followed by cooling at 0 °C for 5 min. Samples were then neutralized with 0.1 M NaOH (1 mL, NAD$^+$ extraction) or 0.1 M HCl (1 mL, NADH extraction). After centrifugation, the supernatant containing pyridine nucleotides was passed through a 0.2 μm filter. The redox cycling assay was performed using a reagent mixture consisting of equal volumes of 1 M bicine buffer (pH8.0) (bicine: Sigma Aldrich, cat. no.- B3876), absolute ethanol as substrate, 40 mM EDTA (pH8.0), 4.2 mM MTT (3- [4,5-dimethyl thiazol-2-yl]–2,5-diphenyl tetrazolium bromide; thiazolyl blue), and twice the volume for 16.6 mM PES (phenazine ethosulfate; Sigma Aldrich, cat. no.- P4544-1G), previously incubated at 30 °C. Of the reagent mixture, 100 μL was incubated with 90 μL of cell extract followed by the addition of 10 μL alcohol dehydrogenase in 0.1 M bicine buffer, ≈ 6 U for NAD(H) estimation (Sigma Aldrich, cat. no.- A3263-30KU). The absorbance at 565 nm was recorded every minute for 10 min at 30 °C. Using 0.01–0.1 mM standard solutions of NADH (NADH-Sigma Aldrich, cat. no.- N8129) and NAD$^+$ (NAD- Sigma Aldrich, cat. no.- N1636), a standard curve (absorbance versus time plot) was generated whose slope (ΔAbsorbance/ min) of the linear region was correlated to the concentration of coenzyme (mM) by a linear fit equation. This equation was used to determine the concentration(mM) of NAD/H in the samples. NADH and NAD$^+$ concentrations were normalized to OD$_{600 \text{ nm}}$.

### Homology modelling of Rv0158 protein

Rv0158 protein sequence was taken from the Mycobrowser repository (https://mycobrowser.epfl.ch/). To find similar protein sequences and structures, the BLAST program (blastp) of Basic Local Alignment Search Tool (BLAST) (https://blast.ncbi.nlm.nih.gov/Blast.cgi) was performed using the query sequence of Rv0158 protein against the protein data bank (PDB). Based on the BLAST result, we selected the top hit: KstR2 protein crystal structure (PDB ID: 4W97) of TetR-family, form *Mtb* (*Crowe et al., 2015*). This protein structure (PDB ID: 4W97) was further used as a reference structure to model the Rv0158 protein using the homology modeling tool MODELLER version 9.19 (https://salilab.org/modeller/).

### Expression and purification of His-tag *Mtb* Rv0158 protein in *E. coli*

The nucleotide sequence of *rv0158* ORF was PCR-amplified from *Mtb* H37Rv genome and cloned into pet28a plasmid. The recombinant plasmid was transformed into *E. coli* BL21 (DE3) cells. At OD$_{600 \text{ nm}}$ of ~0.6, 0.5 mM isopropyl-β-D thiogalactopyranoside (IPTG; VWR Chemicals, cat. no.- 0487–10 g) were added to induce expression of His-tagged Rv0158 for 24 hr at 18 °C. The culture medium containing the bacteria was centrifuged at 3000 *g* for 10 min, and the cell pellet was suspended in lysis buffer 25 mM Tris-Cl pH 8.0, 500 mM NaCl, 5% glycerol, 5 mM βmercaptoethanol, 5 mM imidazole, and

2 mM phenylmethylsulfonyl fluoride (PMSF; Sigma Aldrich, cat. no.- P9625-14). The cells were sonicated for 30 min on ice, followed by centrifugation at 3000 $g$ for 45 min. His-tagged Rv0158 protein in the supernatant was purified on a Ni-NTA (Qiagen, cat. no.- 30210) resin chromatography column and eluted with elution buffer 25 mM Tris-Cl pH 8.0, 500 mM NaCl, 5% glycerol, 5 mM βmercaptoethanol, 250 mM imidazole and 2 mM phenylmethylsulfonyl fluoride (PMSF), followed by dialysis to remove imidazole. The protein purity was evaluated by SDS-PAGE and quantified by BCA assay (Thermo Scientific, cat. no. 23225, Rockford, IL).

## Circular Dichroism spectroscopy and gel filtration chromatography of *Mtb* H37Rv Rv0158 protein

For the CD spectroscopy study, the concentration of the protein sample (Rv0158) was set at 20 µM in 20 mM Tris (pH 7.5), 100 mM NaCl Buffer. Scans and melting curve experiments of the Rv0158 protein sample were performed on a JASCO J-715 spectropolarimeter. CD spectra of Rv0158 were recorded at 25 °C over the wavelength range from 260 nm to 190 nm. Prior to sample spectra, baseline spectra were obtained using 20 mM Tris (pH 7.5), 100 mM NaCl. CD melting curve was recorded at 209 nm over the temperature ranges from 10 °C to 90 °C.

For gel filtration chromatography of protein sample of His-tagged Rv0158 (Ni-NTA eluted), Buffer-20 mM Tris, pH 7.5, 100 mM NaCl was used in Superdex 75 column using GE AKTA prime plus FPLC system. To find protein Rv0158 in the fractions, we performed SDS PAGE to check for bands of the expected size.

## Microscale Thermophoresis to detect binding interactions between small molecules and Protein

10 µM of Rv0158 protein (in 50 mM Tris pH 7.5, 100 mM NaCl) was labelled using RED-NHS MonolithTM Protein Labeling Kit (NanoTemper Technologies, cat. no.- MO-L011) according to manufacturer's instructions. Each of the small molecule ligands (coA (Sigma Aldrich, cat. no.- C4282), acetyl-CoA (Sigma Aldrich, cat. no.- A2056), propionyl-CoA (Sigma Aldrich, cat. no.- P5397), malonyl-CoA (Sigma Aldrich, cat. no.- M4263), methylmalonyl- CoA (Sigma Aldrich, cat. no.- M1762), C10-CoA (Sigma Aldrich, cat. no.- D5269), C12-CoA (Sigma Aldrich, cat. no.- L2659), C16-CoA (Sigma Aldrich, cat. no.- P9716), C18-CoA (Sigma Aldrich, cat. no.- S0802)) were titrated against the labeled protein in 1:1 dilution series starting with the highest ligand concentration of of 75 µM. A total of 16 twofold serial dilutions of the target ligand were prepared. The labeled protein was added such that the final concentration of the protein is 5 nM. Experiments were carried out using MonolithTM NT.115 MST Standard Capillaries (NanoTemper Technologies, cat. no.- MO-K022) and measured using a Monolith NT.115 instrument with MO.Control software. Curves were fitted with a single-site binding model using PALMIST 1.5.6 software (*Scheuermann et al., 2016*) and figures were generated using GUSSI software v1.1.0 (*Brautigam, 2015*).

## Chromatin Immunoprecipitation- Sequencing and Analysis

*rv0158* KO *Mtb* was transformed with ATc-inducible expression vector (pexCF-*rv0158*) and transformants were grown in the presence of 50 µg/mL− hygromycin B to maintain the plasmid. Log phase culture at an OD$_{600}$ of 0.35 was washed twice with 1X PBS, and induced for 24 hr using a low ATc concentration 12.5 ng/mL in presence of either 0.3% (w/v) D-glucose or 0.3% (w/v) sodium propionate as sole C source. Samples for ChIP-Seq were prepared as described previously (*Chawla et al., 2018*; *Minch et al., 2015*). Briefly, DNA–protein interactions were established by cross-linking 50 ml of the culture of *Mtb* strains with 1% formaldehyde while agitating cultures at 37 °C/ 180 RPM for 30 min. Crosslinking was quenched by the addition of glycine to a final concentration of 250 mM. Cells were pelleted, washed in 1 X PBS + 1 X protease inhibitor cocktail (Sigma Aldrich, cat. no.- 11873580001), and resuspended in IP Buffer (20 mM K-HEPES pH 7.9, 50 mM KCl, 0.5 mM dithiothreitol, 10% glycerol and 1 X protease inhibitor cocktail). Cells were lysed and the supernatant was subjected to Covaris S2 ultrasonicator at settings: Duty factor = 10%, Peak power = 174 W, cycles per burst = 200, for 20 min (4 times 5 min with 1 min gap) to shear chromatin to a uniform size centred around 200 bp. Following shearing, the sample was adjusted to buffer IPP150 (10 mM Tris-HCl—pH 8.0, 150 mM NaCl and 0.1% NP40) and shearing of DNA to an average size of 100–400 bp was confirmed by running a de-crosslinked aliquot on Agilent Bioanalyzer. Immunoprecipitation of FLAG-tagged proteins was

initiated by incubating samples overnight rotating at 4 °C with Anti-FLAG M2 Magnetic Beads (Sigma Aldrich cat. no.- M8823), following manufacturer's instructions. Samples without undergoing immuno-precipitation served as Input control for each condition. Beads were washed twice with IP buffer and once with Tris-EDTA buffer pH 7.5. Elution was performed in 50 mM Tris–HCl pH 7.5, 10 mM EDTA, 1% SDS for 40 min at 65 °C. All Samples were finally treated with RNAse A (1 mg/ ml) for 1 hr at 37 °C, and cross-links were reversed by incubation for 2 hr at 50 °C and for 9 hr at 65 °C in elution buffer with Protease K (10 mg/ ml). Input and Immunoprecipitated DNA were subsequently purified using QIAquick PCR purification Kit and the concentration of the DNA was measured using the Qubit HS DNA kit. NEBNExt UltraTM II DNA library Prep with sample purification beads (NEB, cat. no.- E7103L) was used for the library preparation. Samples were sequenced on the Illumina HiSeq 2500 Sequencing System.

The single-end reads with raw read quality of Phred score ≥ 20 were mapped using bwa (version 0.7.12-r1039) to the reference genome (*Mycobacterium tuberculosis* H37Rv, NC_000962.3) obtained from the NCBI database. Samtools (version 0.1.19-96b5f2294a) were used to map the reads with a mapping quality of at least 1 resulting in.sam file. After the conversion of.sam file to.bam, the bam files were sorted using samtools sort command. The sorted bam files of each pulldown-input pair were used to call peaks using Macs2 (version 2.2.6) callpeak function with the following parameters; `--mfold` 2 80 s 50 `--nomodel --keep-dup` all. The peaks obtained were at least 1.5- fold enriched with ≤ 0.01 Q value (FDR). To obtain the genes having overlap with the peaks, gene positions were extracted using the.gff file of the reference file (NC_000962.3), and a custom.bed format file was made. Based on the peak positions and overlap with the genes/regulatory region (−500,+30 from the start codon), genes were identified using in-house python script. A peak was associated with a gene only if both the biological replicates have at least a peak for that gene.

For visualisation of the peaks, the sorted bam files were converted to bigwig format with the following parameters; `--binsize` 100 `--effectiveGenomeSize` 4411532 `--normalizeUsing` RPGC (reads per genomic content), which were further uploaded on integrative genomics viewer (IGV) software. To prepare for the line plots for the pull-down regions, bedtools coverage (version 2.30.0) was used to make coverage using a bed file made from the sorted bam file (bamtobed option) and a custom bed file of chromosome positions with 100 base pairs intervals. From the coverage file, normalised values corresponding to every bin were obtained by dividing the total number of mapped reads for every sample (CPM normalisation). Averages of both the replicates for every sample (both pull-down and input) were calculated which were further used to calculate the Fold change using the formula: (Average CPM of Pull-down samples)/(Average CPM of Input samples). Raw data has been submitted to NCBI's Gene Expression Omnibus (GEO, accession number- GSE242285).

## Identification of consensus motifs of Rv0158 protein

To identify consensus motifs of Rv0158, we extracted 500 nucleotide length sequences corresponding to the peaks obtained from macs2 output. The peak summit was centred in this sequence with 250 base pairs on either side. These sequences were used as input for the MEME-ChIP program *Machanick and Bailey, 2011* of the MEME suite (*Bailey et al., 2015*). Different markov order for the background model was used in the classic mode of MEME-ChIP with maximum number of motifs as 30 and default motif length range from 6 to 15. The motif with lowest enrichment e-value was considered as the best consensus motif. The consensus motif was consistently the topper in all models used.

To identify the positions of the best enriched motif and known motifs (PhoP, PrpR, RamB, GlnR) we used FIMO *Grant et al., 2011* from the MEME suite (*Bailey et al., 2015*) with the regulatory region (−500 to +30) of the genes (the genes with significant peaks) as input. The output files were processed using in-house python and shell-based scripts to obtain the distance between the enriched motifs and known motifs.

## Metabolite extraction from *Mtb* strains and LC-MS/MS-based analysis

Metabolite isolation and Targeted analysis of Methylcitrate and TCA cycle metabolites were done as described previously (*Bandyopadhyay et al., 2022*; *Tan et al., 2014*; *Tripathi et al., 2022*; *Walvekar et al., 2018*). Briefly, *Mtb* strains were precultured to OD $_{600 nm}$ of ≈1.0, washed with 1 X PBS, and resuspended in 7H9 supplemented with either 0.3% (w/v) D-glucose or 0.3% (w/v) sodium propionate, followed by incubation at 180 RPM in a rotary shaker incubator. Log phase cells (OD = 0.8–1.2)

were quenched for 5 min in 4 volumes of 60% methanol (maintained at –45 °C in dry ice–methanol bath) and then centrifuged at 4800 $g$ (–10 °C). The pellet was re-suspended in 700 µL of 60% methanol (maintained at –45 °C in dry ice–methanol bath) and then centrifuged at 4800 $g$ (–10 °C). The pellet was re-suspended in 1 mL of 75% ethanol and incubated at 80 °C for 3 min with intermittent vortexing at 1.5 min intervals, immediately followed by incubation on ice for 5 min and centrifugation at 17,000 $g$ for 15 min. The supernatant was then lyophilized on a vacuum concentrator for 3–4 h and then stored at –80 °C till further analysis.

For measurement of the TCA cycle and glyoxylate cycle intermediates, metabolites were derivatized with O-benzylhydroxylamine (OBHA) as mentioned previously (*Tan et al., 2014*; *Walvekar et al., 2018*). Oxaloacetate is unstable in aqueous solutions, hence was stabilized by OBHA derivatization to prevent degradation. The metabolites were separated using Synergi 4 µm Fusion-RP 80 Å LC column (150x4.6 612 mm, Phenomenex) on Shimadzu Nexera UHPLC system. Solvents used for the separation of TCA intermediates are the following: 0.1% formic acid in water (solvent A) and 0.1% formic acid in methanol (solvent B). The metabolites were detected using ABSciex QTRAP 5500 mass spectrometer. The detailed parameters used are described in *Walvekar et al., 2018*. The data was acquired using Analyst 1.6.2 software (Sciex) and analyzed using MultiQuant version 3.0.1 (Sciex).

## Maintenance, culturing, and infection of the macrophage cell line- RAW 264.7 with *Mtb*

The murine macrophage cell line- RAW 264.7 were acquired from ATCC (Manassas, VA) and tested negative for mycoplasma contamination by DE-MycoX Mycoplasma PCR Detection Kit (CELLclone cat. no.- GX-E-250). RAW 264.7 was grown in DMEM medium (Cell Clone) supplemented with 10% heat-inactivated (55 °C) fetal bovine serum (MP Biomedical, cat. no.- 092916754) in an incubator containing 5% $CO_2$ at 37 °C. A total of $3 \times 10^5$ cells/well was seeded into a 24-well cell-culture plate. RAW macrophages were activated with IFN-γ (100 U/mL) (Peprotech; cat. No. 315–05) and *E. coli* lipopolysaccharide (50 ng/mL; Sigma Aldrich, cat. no.- L2630) for 16 hr. The resulting activated macrophages were infected with *Mtb* expressing Mrx1-roGFP2 at a multiplicity of infection (MOI) of 10 and incubated for 4 hr at 37 °C in 5% $CO_2$. After infection, extracellular bacteria were removed by washing three times with phosphate-buffered saline (PBS; 137 mM NaCl, 2.7 mM KCl, 10 mM $Na_2HPO_4$, and 1.4 mM $KH_2PO_4$, pH 7.4). For CFU determination, infected cells were lysed in 7H9 medium containing 0.06% sodium dodecyl sulfate (SDS); dilutions were prepared using 7H9 medium, and aliquots were plated on 7H11+OADC agar plates. Plates were incubated at 37 °C for 3–4 weeks before counting colonies.

For intra-mycobacterial $E_{MSH}$ determination, infected macrophages were treated with 10 mM N-ethylmaleimide (NEM; Sigma-Aldrich, cat. no.-E3876) for 5 min at room temperature, followed by fixation with 4% paraformaldehyde (PFA; Himedia, cat. no.- GRM3660) for 1 hr at room temperature. Infected macrophages were analyzed using a FACS Verse Flow cytometer (BD Biosciences). Intramycobacterial $E_{MSH}$ was determined using the Nernst Equation as described previously (*Bhaskar et al., 2014*). 10,000 events per sample were analyzed.

## Animal handling, maintenance, and aerosol infection in C57BL/6J mice

All animal studies were executed as per guidelines prescribed by the Committee for the Purpose of Control and Supervision of Experiments on Animals, Government of India, with approval from the Institutional Animal Ethical Committee and Biosafety Level-3 Committee. 6- to 8-week-old female pathogen-free C57BL/6 J mice were infected via a low-dose aerosol exposure to the *Mtb* strains using a Madison chamber aerosol generation instrument. At one day post-infection, three mice were sacrificed to verify the implantation of ~100 CFU of bacteria per mouse. Feed and water were given *ad libitum*. 4 mice/ group were sacrificed at 4 weeks and 8 weeks post-infection, and the lungs and spleen were harvested for measurement of bacterial burden. CFUs were determined by plating appropriate serial dilutions on 7H11 (supplemented with OADC) agar plates and counting visible colonies after 3–4 weeks of incubation at 37 °C.

## Statistical analysis

Statistical analysis was performed using GraphPad Prism version 8.4.3 software. The mean and standard error of mean values was plotted as indicated in figure legends. A p-value of less than 0.05 was

considered significant. Statistical Significance was determined by unpaired two-tailed student's t-test; one-way ANOVA was performed where a comparison of multiple groups was made.

## Resource availability

### Lead contact
Further information and requests for resources and reagents should be directed to the Lead Contact: Dr. Amit Singh (asingh@iisc.ac.in).

### Materials availability
This study did not generate new unique reagents. Constructs and other reagents generated in this study will be made available from the lead contact for academic/non-commercial research purposes on request without restriction under a Material Transfer Agreement.

## Acknowledgements

We thank Prof. Karl Drlica for his critical comments on the manuscript.

## Additional information

### Funding

| Funder | Grant reference number | Author |
|---|---|---|
| Wellcome Trust - DBT India Alliance | IA/S/16/2/502700 | Amit Singh |
| Indian Institute of Science | Graduate Student Fellowship | Somnath Shee |
| Department of Atomic Energy, Government of India | project number 12-R&D-TFR-5.04-0800 | Aswin N Seshasayee |
| Department of Biotechnology, Ministry of Science and Technology, India | BT/PR29098/Med/29/1324/2018 | Amit Singh |
| Department of Biotechnology, Ministry of Science and Technology, India | BT/HRD/NBA/39/07/2018-19 | Amit Singh |
| Infosys Foundation | | Amit Singh |
| Department of Science & Technology, Ministry of Science and Technology, India | SERB/CRG/2022/002009/ | Amit Singh |
| Ignite Life Science Foundation | SP/ILSF-22-0001 | Amit Singh |
| Department of Science and Technology, Ministry of Science and Technology, India | DST (FIST) | Amit Singh |
| Department of Science and Technology, Ministry of Science and Technology, India | UGC (special assistance) | Amit Singh |

The funders had no role in study design, data collection and interpretation, or the decision to submit the work for publication.

## Author contributions
Somnath Shee, Conceptualization, Data curation, Software, Formal analysis, Validation, Investigation, Visualization, Methodology, Writing – original draft, Writing – review and editing; Reshma T Veetil, Karthikeyan Mohanraj, Mayashree Das, Nitish Malhotra, Devleena Bandopadhyay, Vikrant Kumar Sinha, Chandrani Thakur, Nagasuma Chandra, Data curation, Formal analysis, Investigation, Methodology; Hussain Beig, Shalini Birua, Data curation, Investigation, Methodology; Shreyas Niphadkar, Data curation, Investigation, Methodology, Writing – review and editing; Sathya Narayanan Nagarajan, Conceptualization, Investigation, Methodology; Raju S Rajmani, Investigation, Methodology; Sunil Laxman, Data curation, Formal analysis, Investigation, Methodology, Writing – review and editing; Mahavir Singh, Areejit Samal, Aswin N Seshasayee, Data curation, Formal analysis, Investigation, Methodology, Writing – original draft; Amit Singh, Conceptualization, Resources, Data curation, Formal analysis, Supervision, Funding acquisition, Validation, Investigation, Visualization, Methodology, Writing – original draft, Project administration, Writing – review and editing

## Author ORCIDs
Somnath Shee ⓘ http://orcid.org/0000-0001-9313-8559
Mayashree Das ⓘ http://orcid.org/0000-0003-1776-3648
Nitish Malhotra ⓘ http://orcid.org/0000-0001-8580-9623
Sathya Narayanan Nagarajan ⓘ http://orcid.org/0000-0002-3902-3678
Nagasuma Chandra ⓘ http://orcid.org/0000-0002-9939-8439
Sunil Laxman ⓘ http://orcid.org/0000-0002-0861-5080
Amit Singh ⓘ http://orcid.org/0000-0001-6761-1664

## Ethics
All animal studies were executed as per guidelines prescribed by the Committee for the Purpose of Control and Supervision of Experiments on Animals, Government of India, with approval from the Institutional Animal Ethical Committee (CAF/Ethics/544/2017- Institute animal ethical clearance number) and Biosafety Level-3 Committee.

## Decision letter and Author response
Decision letter https://doi.org/10.7554/eLife.80218.sa1
Author response https://doi.org/10.7554/eLife.80218.sa2

---

# Additional files

### Supplementary files
- Supplementary file 1. TnSeq summary table.
- Supplementary file 2. List of genes that constitutes *Mtb* redoxosome.
- Supplementary file 3. List of previous studies that discuss about potential function of *Mtb* redoxosome- genes.
- Supplementary file 4. List of differentially expressed genes in untreated *rv0158* KO *Mtb* and *rv0158* Comp relative to untreated wild-type *Mtb.*
- Supplementary file 5. Binding sites of Rv0158 protein in *Mtb* genome.
- Supplementary file 6. Binding motifs for other TFs (PrpR, PhoP, RamB, and GlnR) known to regulate propionate metabolism in the regulatory regions of genes bound by Rv0158. The distance between the Rv0158 motif and motifs of other TFs is also indicated.
- Supplementary file 7. Transposon mutants of *Mtb* CDC 1551 were collected from BEI Resources used in this study.
- Supplementary file 8. DNA oligonucleotides (Primers) used in this study.
- MDAR checklist

### Data availability
All data generated or analysed during this study are included in the manuscript (in the materials and methods section). The TraDIS data has been submitted to NCBI's BioProject (BioProject ID: PRJNA807454). RNA Sequencing data have been submitted to NCBI's Gene Expression Omnibus (GEO, accession number GSE196844). ChIP-Sequencing data have been submitted to NCBI's Gene

Expression Omnibus (GEO, accession number GSE242285). Source Data have been provided for all Figures. Source Data contain the numerical data used to generate the figures.

The following datasets were generated:

| Author(s) | Year | Dataset title | Dataset URL | Database and Identifier |
|---|---|---|---|---|
| Shee S, Veetil RT, Malhotra N, Narain Seshashyee AS, Singh A | 2022 | Transposon directed insertion site sequencing to identify site of transposon-insertion in Mtb genome | http://www.ncbi.nlm.nih.gov/bioproject/?term=PRJNA807454 | NCBI BioProject, PRJNA807454 |
| Shee S, Veetil RT, Malhotra N, Narain Seshashyee AS, Singh A | 2022 | Transcriptomic analysis of untreated log-phase Mtb H37Rv, Mtbrv0158 Knockout strain and Mtbrv0158 complemented strain | https://www.ncbi.nlm.nih.gov/geo/query/acc.cgi?acc=GSE196844 | NCBI Gene Expression Omnibus, GSE196844 |
| Shee S, Malhotra N, Narain Seshasayee AS, Singh A | 2023 | Biosensor-integrated transposon mutagenesis reveals rv0158 as a coordinator of redox homeostasis in Mycobacterium tuberculosis (ChIP-Seq) | https://www.ncbi.nlm.nih.gov/geo/query/acc.cgi?acc=GSE242285 | NCBI Gene Expression Omnibus, GSE242285 |

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
