## [Editor Report]

This manuscript is important as it describes the powerful combination of TnSeq approaches with a reporter for mycothiol redox potential to identify genes in *Mycobacterium tuberculosis* that are required to maintain the reducing environment of the cell. Through this study, the authors provide compelling data that Rv0158 functions as a critical regulator of bacterial responses to endogenous reactive oxygen species. This study will be important to investigators interested in bacterial physiology and how this can impact pathogenesis.

---

## [Decision Letter]

**Decision letter after peer review:**

Thank you for submitting your article "Biosensor-integrated transposon mutagenesis reveals rv0158 as a coordinator of metabolism-linked redox homeostasis in *Mycobacterium tuberculosis*" for consideration by *eLife*. Your article has been reviewed by 3 peer reviewers, and the evaluation has been overseen by a Reviewing Editor and Wendy Garrett as the Senior Editor. The reviewers have opted to remain anonymous.

Essential revisions:

1) A more thorough analysis and dissection of Rv0158 is necessary to provide insight into its role in mycobacterial metabolism and redox homeostasis.

2) The authors should be more conservative with their data interpretations and open to alternative models of Rv0158 function.

*Reviewer #1 (Recommendations for the authors):*

In this manuscript, the authors combined TnSeq approaches with a reporter for mycothiol redox potential to identify genes in Mtb that are required to maintain the reducing environment of the cell. The authors then examine the pooled library of Tn mutants that were more oxidized for evidence of DNA damage, lipid peroxidation, upregulation of oxidative stress gene expression, and inability to recover from oxidative stress. All of this validated that they had selected a pool of mutants that were on average exhibiting a more oxidized state. The authors then focus the rest of the manuscript on the transcription factor Rv0158, by generating a deletion mutant and complemented version for Rv0158 and confirming that the mutant exhibits a more oxidized condition. This is an interesting and important characterization of Rv0158 in Mtb. Based on their data, the authors propose that Rv0158 is required to calibrate the cytoplasmic redox potential in response to changes in carbon sources. However, there are multiple explanations for the phenotypes observed, and the function for Rv0158 in metabolism and redox homeostasis remains unclear.

– Lines 329-331- the authors mention that docking analysis highlighted cholesterol as a potential ligand for Rv0158, but is this just because they used the KstR2 structure as a model, which would fit the Rv0158 structure to KstR2 and may not capture important differences in the ligand binding site? What is the potential model for how cholesterol sensing would allow Rv0158 to serve as a signal for redox homeostasis?

– For the experiments in Figure 8, the authors state that they hypothesized that the role of Rv0158 would be carbon source dependent, but it is not immediately obvious why that is, and what the outcome for particular carbon sources would be expected. Therefore, in Lines 422-423, it is not clear that the data is in agreement with the hypothesis because it was not obvious what the hypothesis was. In particular, because the authors earlier state that Rv0158 may respond to cholesterol, why would one not expect the TF to function in the presence of cholesterol? Better flushing out of the model and mechanism would be helpful.

– Figure 8 suggests that the deletion mutant is better equipped for growth on lipid carbon sources, possibly affecting metabolism and carbon source use, which impacts redox. The authors state in lines 447-448 that their findings indicate a requirement for rv0158 to calibrate the cytoplasmic redox potential in response to changes in carbon sources. However, it seems equally possible that Rv0158 has some role in carbon metabolism or assimilation of certain carbon sources into central metabolism, and this is having an indirect effect on redox potential. This may be particularly relevant since the initial experiments were performed in media with multiple available carbon sources and there was still a phenotype. The seahorse experiments also suggest issues with carbon source use.

– Does the carbon source affect the sensitivity to the conditions used in 10A?

– The growth defect in macrophages could just reflect the same growth defect observed in culture in glucose-containing media. It does not appear to be specific to the macrophages and this should be taken into account when interpreting that data.

*Reviewer #2 (Recommendations for the authors):*

Shee and colleagues implemented a clever TnSeq screen to identify mutants with impaired intracellular redox homeostasis. They identified, what they call the redoxosome, which includes functionally diverse genes of which 1/3rd is related to oxidative stress/antioxidant response. Network analysis led to the identification of rv0158, which encodes a putative transcriptional regulator. The authors constructed a deletion mutant and characterized it by RNAseq, seahorse measurements, lipid analyses, and during growth in a macrophage cell line and in mice. The authors conducted many experiments, which revealed that the rv0158 mutant has many phenotypes, but unfortunately the direct regulon of Rv0158 remains unidentified. The mechanistic basis for the attenuation of the rv0158 deletion mutant in the mouse model is therefore unexplained.

The identified "redoxosome" includes genes that are functionally very diverse, and I am not convinced that the majority is required for redox homeostasis. Moreover, the authors failed to validate individual transposon mutants. The validation experiments done with the enriched mutant pool (Figure 2) are not informative as the data could be driven by just a few of the mutants. They analyzed EMSH response in a handful of transposon mutants obtained from BEI but did not carry out any further functional analysis that would provide confidence that the identified mutants are in fact required for redox homeostasis. The EMSH response is not sufficient to support this claim. It is not clear why these specific mutants were selected.

The authors focused on the putative transcriptional regulator Rv0158 and used RNAseq to identify its regulon. The approach cannot distinguish genes that are directly regulated by Rv0158 from those that are indirectly regulated. This is supported by the large number of genes identified. Accordingly, the follow-up work does not lead to significant insight into the function of Rv0158. The mutant accumulated fewer mycolic acids than wt was more permeable to EtBr, and contained higher NADPH levels than wt Mtb; the underlying mechanisms and specific dysregulated genes that are responsible for these phenotypes remain undetermined. Similarly, the growth defects are difficult to interpret and the conclusion that rv0158 deletion improved growth with fatty acids as a carbon source is not supported by the data; this is only observed with propionate.

The conclusion that rv0158 is required to survive in macrophages is incorrect. The mutant survives perfectly well (100% survival); it fails to grow like wt. The slight *in vivo* growth defect and more drastic persistence defect seems inconsistent with the data from the macrophage cell line.

The MIC measurements using resazurin are a concern, given that resazurin reports on respiration, it is reduced in aerobically growing cells. The altered intracellular redox state of the rv0158 mutant could affect the resazurin reduction and confound the data.

There are some inaccurate statements that need to be corrected. For example, the authors state "Mtb secretes antioxidant enzymes (catalase-peroxidase (KatG), superoxide dismutase (Fe-SodA), thioredoxin reductase (trxB2), Alkyl hydroperoxide reductase (AhpC) and redox buffers thioredoxin (trxC), and ergothioneine). Not all of these are secreted; in fact, the majority are not.

They wrote" Mtb displays a low MIC of 3 μM for the endogenous superoxide (O2.-) generator 2, 3-Dihydro-1,4, naphthoquinone and 0.4 μM for the intracellular NO donor PA-824. On what basis do the authors quantify the MICs as "low"?

The experiments shown in Figure 2 should be conducted with isolated transposon mutants.

The genes that are directly regulated by Rv0158 need to be identified to gain mechanistic insight into the many phenotypes of the mutant and potentially explain the *in vivo* attenuation.

The authors should test growth with additional fatty acids, especially those that are relevant to the *in vivo* environment to substantiate the claim that rv0158 is required for growth with fatty acids.

The conclusion that the key rv0158 function is to maintain the flux of lipid metabolism intermediates needs to be substantiated by carbon flux analyses.

MIC measurements using growth/OD600 readings should be conducted.

*Reviewer #3 (Recommendations for the authors):*

The authors of the study combined the Mrx-roGFP2 redox biosensor with transposon mutagenesis and FACS to identify the Mtb redoxome. From this library, the authors identify a hypothetical protein, RV0158 that modulates intracellular redox balance and lipid homeostasis. The corresponding Mtb knockout strain is implicated in anti-TB drug susceptibility and survival *in vivo*. I am concerned that the claims by the authors are not supported by the data.

The strengths of the manuscript are combining roGFP with FACS to identify a set of genes involved in redox balance, implicating a regulatory gene (rv0158) involved in this process, and demonstrating its role in anti-TB drug susceptibility and virulence.

The weaknesses of the manuscript include using pooled libraries of transposon mutants (the redoxome) in their Q-PCR and oxidative stress exposure experiments. Examining pooled libraries of mutant clones complicates the interpretation of data that is based on the average contribution of all the genes and makes it impossible to make reliable conclusions. Also, the redoxome was identified using a single oxidant, CHP, which is not a physiologically relevant oxidant. Hence, the redoxome is an artificial signature. An important control, an antioxidant to reverse the response, is lacking. The manuscript lacks a mechanism whereby RV0158 regulates redox homeostasis. Although the lipid metabolism, oxidative stress, and redox couple experiments provide circumstantial evidence, a clear sensing/regulatory mechanism is lacking. The authors favor that rv0158 regulates lipid metabolism as a mechanism for redox homeostasis, but unfortunately, no mechanism is provided. Lastly, I have concerns about the bioenergetic data and interpretation. Overall, numerous casually linked experiments were performed, and the manuscript appears unfocused.

There are numerous grammatical issues throughout the manuscript that warrant attention.

Lines 33-34; ambiguous sentences, rephrase.

Lines 56-61; add references where relevant and not at the end of the sentence.

Line 188; what do the authors mean by nontoxic? How was non-toxicity measured/assessed? This should be provided.

Line 298-301; some of the references are outdated.

Lines 313-333; this section is unnecessary and should be moved to the supplemental data.

Line 423; figure 8A show delayed growth; the growth rate appears to be similar to the wild-type and complemented clones. Also, in figure 8E, the more reducing environment when cultured on fatty acid precursors compared to glucose is expected. However, only in the case of acetate (and glucose) is there a difference between the knockout and corresponding wild-type strain; propionate and cholesterol show no difference. Wise is the case and how does it relate to the role of rv0158? The authors should explain this.

Lines 459-461; this data is not convincing as panels A and D are not baselined whereas all the other panels are. My suspicion is that if these data sets are baselined, there will be very little difference between the %OCR of the knockout and wild-type/complemented clones. Also, precisely how did the authors calculate the ~two-fold less/lower values?

Line 457; CCCP is solely an uncoupler and does not increase carbon catabolism and it does not reveal the metabolic flexibility of Mtb; it only does so in response to particular stressors.

Line 523-526; confusing sentence, rephrase.

Line 1729; correctly rearrange panels E, F, and G.

Line 1824; this heading is probably wrong. Respiration and not energy were measured in these experiments. Indicate that the Agilent Seahorse XFp was used to measure respiration. Also, in line 1826, extracellular acidification is not only due to glycolytic flux. This should be corrected.

Line 1835; CCCP does not allow the detection of spare respiratory capacity. The authors confuse spare respiratory capacity with mammalian experiments.

[Editors' note: further revisions were suggested prior to acceptance, as described below.]

Thank you for resubmitting your work entitled "Biosensor-integrated transposon mutagenesis reveals *rv0158* as a coordinator of redox homeostasis in *Mycobacterium tuberculosis*" for further consideration by *eLife*. Your revised article has been evaluated by Wendy Garrett (Senior Editor) and a Reviewing Editor.

The manuscript has been improved but there are some remaining issues that need to be addressed, as outlined below:

The authors have added a tremendous amount of work to the manuscript and strengthened their dataset. They have addressed many prior concerns to yield an impressive story, but there are some concerns with the newly added data that need to be addressed. In addition, the authors still seem to bias their interpretations towards their favored model and should revise the manuscript to consider alternatives.

The authors should address the following comments, which they should largely be able to do by not overstating their results, considering other models, and providing a more thorough and transparent analysis of the metabolite measurements.

1. The findings that the rv0158 deletion mutant can grow on propionate at a concentration that inhibits growth of the wt and complemented strains and that Rv0158 can bind methylmalonyl-CoA are interesting. However, the authors' interpretations are often based on phenotypic differences whose physiological relevance remains unclear. Their interpretation of the phenotypes seems biased towards the model they propose, but other explanations should be considered and included, at least in the discussion.

2. There are concerns regarding the metabolomics data. The authors do not seem to have used metabolite standards. Even the source data fail to show which masses they utilized for the measured metabolites. Oxaloacetate is very unstable and can typically not be measured directly. The statement "We found that α-ketoglutarate pools are marginally higher in rv0158 KO compared to WT Mtb under propionate (Figure 9; [Figure 9 in the revised manuscript]), indicating another reason for better mutant growth on propionate" is not justified, because in the complemented mutant α-ketoglutarate pools are even higher than that in the KO.

3. The authors added CFU data to assess susceptibility to antibiotics, but this is different than MIC measurements using growth as readout to assess drug resistance, which can be different than susceptibility to drug induced killing. This should be considered when describing and interpreting the data.

4. The authors' explanation for the different binding targets of Rv0158 during growth in glucose and in propionate is not clear. It seems that they suggest that methylmalony -CoA binding is required for Rv1058 to bind and repress the prpR promoter, but direct evidence for this is lacking. Overall, their model of Rv0158 activity is very complex and is sometimes based solely on associations. The authors would benefit by more rigorously evaluating their data and being more careful with their interpretations.

---

## [Author Response]

Essential revisions:1) A more thorough analysis and dissection of Rv0158 is necessary to provide insight into its role in mycobacterial metabolism and redox homeostasis.

To better understand Rv0158 function in mycobacterial metabolism and redox balance, we performed several key experiments described below.

One of the significant findings of our manuscript is the slow growth of *rv0158* KO on glucose and mixed nutrients (glucose, glycerol, and tween-80), whereas growth was better on fatty acids such as propionate (Figure 7 and Figure 7—figure supplement 1 in the revised manuscript). Therefore, to understand the better adaptability of *rv0158 KO* on propionate as a carbon (C) source, we performed liquid chromatography-mass spectrometry (LC-MS) metabolomics following exposure of WT *Mtb* and *rv0158 KO* to 0.3% glucose or 0.3% propionate (Figure 9 in the revised manuscript). We assessed the pool sizes of the intermediates associated with the methylcitrate cycle (MCC), TCA, and glyoxylate cycles. Our data suggest that Rv0158 deficiency accelerates propionyl-CoA utilization and reduces the pool of toxic metabolites such as glyoxylate (Figure 9 in the revised manuscript), thereby positively influencing the growth and redox potential of *rv0158 KO* utilizing propionate as a sole carbon source (Figure 7 in the revised manuscript).

We then extended our study to understand the potential model of propionate sensing by Rv0158. We employed microscale thermophoresis (MST) to identify ligands for Rv0158 and found that it binds with malonyl-CoA, decanoyl -CoA, C12/C16/C18-CoA, and methylmalonyl-CoA. The highest affinity was for methylmalonyl-CoA (affinity constant [Kd] of 0.9 ± 0.3 µM) (Table 1 in the revised manuscript). Methylmalonyl-CoA is a crucial intermediary metabolite regulating propionate flux into MCC, methylmalonyl pathway, and methyl branched lipids in *Mtb*. Therefore, we proposed that the physical association of methylmalonyl-CoA with Rv0158 likely facilitates the regulation of genes associated with propionate metabolism in *Mtb*. We performed ChIP-sequencing (ChIP-seq) of Rv0158 on glucose or propionate as a sole carbon source to examine this (Figure 11 and Supplementary file 5 in the revised manuscript). On glucose, Rv0158 binds to several genes involved in methyl-branched lipid biosynthesis (*pks2*, *fadD23*, *rv2958c*, *rv2959c*, *rv2961*), lipid transporter (*rv0987-rv0988*, *mmpL1-mmpS1*, *mmpL4*-*mmpS4*), biotin synthesis (*bioF2*), and surface-associated proteins (*esp* operon, *lpqG*, esxQ, *rv1501*-r*v1507A*). Under propionate condition, Rv0158 binds to the genomic region encompassing 32 genes, 24 genes overlap with glucose. The promoter regions of the remaining eight genes that Rv0158 exclusively bind in propionate- grown *Mtb* were part of the glyoxylate cycle and MCC (*icl1*, *fadB2*, *prpR*, *and prp* operon) (Figure 11 and Supplementary file 5 in the revised manuscript).

Lastly, to support ChIP-seq data, we performed qRT-PCR of MCC genes (*icl1*, *prpR*, *prpC*, *prpD*) in WT *Mtb*, *rv0158* KO, and *rv0158 Comp* growing on glucose or propionate by qRT-PCR. We observed increased expression of MCC genes in *rv0158* KO compared to WT *Mtb* on glucose (Figure 10A, 10B, and Figure 10—figure supplement 2 in the revised manuscript). Propionate exposure increased MCC gene expression in *rv0158* KO compared to WT *Mtb.* In addition to the MCC pathway, *Mtb* mitigates propionate toxicity via the biosynthesis of PDIM (phthiocerol dimycocerosates). We measured the expression of the PDIM biosynthesis cluster in *rv0158* KO. Expression of PDIM genes was broadly unchanged in *rv0158* KO on glucose compared to WT *Mtb* (Figure 10D and Figure 10—figure supplement 2 in the revised manuscript)*.* However, similar to MCC genes, the PDIM cluster was induced ≈10-fold in *rv0158* KO compared to WT *Mtb* on propionate (Figure 10E and Figure 10—figure supplement 2 in the revised manuscript). In sum, our ChIP-seq, RT-qPCR, and MST data indicate that Rv0158 prevents the hyperexpression of MCC genes and adjusts the partitioning of propionate between the catabolism and methyl branched lipids anabolism in *Mtb.*

We thank the Reviewers for these excellent recommendations.

2) The authors should be more conservative with their data interpretations and open to alternative models of Rv0158 function.

We have extensively revised the manuscript and provided alternative models of the Rv0158 function in *Mtb*.

Reviewer #1 (Recommendations for the authors):In this manuscript, the authors combined TnSeq approaches with a reporter for mycothiol redox potential to identify genes in Mtb that are required to maintain the reducing environment of the cell. The authors then examine the pooled library of Tn mutants that were more oxidized for evidence of DNA damage, lipid peroxidation, upregulation of oxidative stress gene expression, and inability to recover from oxidative stress. All of this validated that they had selected a pool of mutants that were on average exhibiting a more oxidized state. The authors then focus the rest of the manuscript on the transcription factor Rv0158, by generating a deletion mutant and complemented version for Rv0158 and confirming that the mutant exhibits a more oxidized condition. This is an interesting and important characterization of Rv0158 in Mtb. Based on their data, the authors propose that Rv0158 is required to calibrate the cytoplasmic redox potential in response to changes in carbon sources. However, there are multiple explanations for the phenotypes observed, and the function for Rv0158 in metabolism and redox homeostasis remains unclear.

We thank the Reviewer for responding positively to our manuscript. We have addressed all of the Reviewer’s concerns in the revised manuscript.

– Lines 329-331- the authors mention that docking analysis highlighted cholesterol as a potential ligand for Rv0158, but is this just because they used the KstR2 structure as a model, which would fit the Rv0158 structure to KstR2 and may not capture important differences in the ligand binding site? What is the potential model for how cholesterol sensing would allow Rv0158 to serve as a signal for redox homeostasis?

We agree with the Reviewer that *in silico* investigation may not provide a potential model for how Rv0158 senses fatty acids such as cholesterol and propionate for redox homeostasis. Therefore, we employed microscale thermophoresis (MST) to identify ligands for Rv0158. We tested various metabolites generated during the catabolism of long-chain fatty acids, propionate, and cholesterol (Lines: 457- 464 in the revised manuscript). We found that Rv0158 does not bind with CoA, acetyl-CoA, and propionyl-CoA. However, Rv0158 showed binding with malonyl-CoA, decanoyl -CoA, C12/C16/C18-CoA, with the highest affinity for methylmalonyl-CoA (affinity constant [Kd] of 0.9 ± 0.3 µM) (Table 2 and Table 2- Source Data 1 in the revised manuscript). Since methylmalonyl-CoA is a crucial intermediary metabolite regulating propionate flux into MCC, methylmalonyl pathway, and methyl branched lipids (Savvi et al., 2008), we proposed that its physical association with Rv0158 likely facilitates regulation of genes associated with propionate metabolism in *Mtb*. We supported this model by performing ChIP- Seq and expression analysis during the growth of *Mtb* on glucose or propionate as a sole carbon source.

– For the experiments in Figure 8, the authors state that they hypothesized that the role of Rv0158 would be carbon source dependent, but it is not immediately obvious why that is, and what the outcome for particular carbon sources would be expected. Therefore, in Lines 422-423, it is not clear that the data is in agreement with the hypothesis because it was not obvious what the hypothesis was. In particular, because the authors earlier state that Rv0158 may respond to cholesterol, why would one not expect the TF to function in the presence of cholesterol? Better flushing out of the model and mechanism would be helpful.

We thank the Reviewer for this comment. We have simplified this section (Lines: 355- 370 in the revised manuscript). Our RNA-Seq data on media containing mixed carbon sources (glucose, glycerol, and tween 80) indicated that genes associated with fatty acid catabolism were induced in *rv0158* KO. Furthermore, genes related to the methylcitrate cycle (MCC; *prpC, prpD,* and *icl1*), which alleviate propionate toxicity (Eoh and Rhee, 2014a), were significantly upregulated in *rv0158* KO. On this basis, we first examined the role of *rv0158* in utilizing fatty acids as nutrients by *Mtb*. Furthermore, fatty acids being a more reduced C substrate than glucose (Borah et al., 2021; Singh et al., 2009), we tested if differences in the growth of *rv0158* KO on fatty acids vs glucose are also associated with changes in the redox potential of *Mtb*. Our ligand binding assay, metabolomics, ChIP-seq, and expression analysis shed new light on the mechanism of how Rv0158 could function in the presence of propionate to adjust the growth and redox potential of *Mtb*.

– Figure 8 suggests that the deletion mutant is better equipped for growth on lipid carbon sources, possibly affecting metabolism and carbon source use, which impacts redox. The authors state in lines 447-448 that their findings indicate a requirement for rv0158 to calibrate the cytoplasmic redox potential in response to changes in carbon sources. However, it seems equally possible that Rv0158 has some role in carbon metabolism or assimilation of certain carbon sources into central metabolism, and this is having an indirect effect on redox potential. This may be particularly relevant since the initial experiments were performed in media with multiple available carbon sources and there was still a phenotype. The seahorse experiments also suggest issues with carbon source use.

We agree with the Reviewer. Our metabolomics, ChIP-seq, and expression data confirm that Rv0158 deficiency accelerates propionyl-CoA utilization and reduces the pool of toxic metabolites such as glyoxylate, thereby positively influencing the growth and bioenergetics of *rv0158 KO* utilizing propionate as a sole carbon source. We thank the Reviewer for this insightful comment.

– Does the carbon source affect the sensitivity to the conditions used in 10A?

We thank the Reviewer for this comment. We have addressed the Reviewer’s concern by measuring the survival of *rv0158* KO under hydrogen peroxide (H2O2) stress when grown in the presence of acetate/propionate /cholesterol as a sole carbon source (Figure 12—figure supplement 1 in the revised manuscript). We found that the sensitivity of *rv0158* KO towards H2O2 stress was alleviated when we substituted glucose with fatty acids (0.03% acetate or 0.3% propionate or 0.02% cholesterol) as C source (Lines: 503- 508 in the revised manuscript). This aligns with the restoration of MSH redox state (Figure 7F in the revised manuscript) and OXPHOS (relative to WT *Mtb*) in the knockout strain when growing with fatty acids as a C source (Figure 8 in the revised manuscript).

– The growth defect in macrophages could just reflect the same growth defect observed in culture in glucose-containing media. It does not appear to be specific to the macrophages and this should be taken into account when interpreting that data.

We thank the Reviewer for this comment. We showed that the *rv0158* KO exhibits a slow-growth phenotype in glucose-containing media. In the revised manuscript, we addressed the Reviewer’s concern by assessing if the macrophage environment induces slow growth like glucose or killing of the mutant over time (Lines: 532- 536 in the revised manuscript). Immune- activation of macrophages elevates phagosomal oxidative and nitrosative stress to cause redox stress in *Mtb* (Bhaskar et al., 2014). Therefore, we examined the mutant phenotype inside the lipopolysaccharide (LPS), and IFN-γ activated RAW 264.7 macrophages. We observed that *rv0158* KO entered bacteriostasis two days post-infection (p.i.) and showed ≈50% killing at four days p.i. (Figure 12D in the revised manuscript). The killing of the mutant coincides with an overwhelming oxidative shift in the *E_MSH_* of *Mtb* inside immune-activated macrophages (Figure 12C in the revised manuscript). At both these time points, the survival of the mutant remains significantly lower than the WT *Mtb*. The data agree with our findings that *rv0158* KO is exceptionally sensitive to oxidative and nitrosative stress *in vitro*. Altogether, while glucose slows the growth of *rv0158* KO *in vitro*, an oxidatively hostile environment of activated macrophages induces killing.

Reviewer #2 (Recommendations for the authors):Shee and colleagues implemented a clever TnSeq screen to identify mutants with impaired intracellular redox homeostasis. They identified, what they call the redoxosome, which includes functionally diverse genes of which 1/3rd is related to oxidative stress/antioxidant response. Network analysis led to the identification of rv0158, which encodes a putative transcriptional regulator. The authors constructed a deletion mutant and characterized it by RNAseq, seahorse measurements, lipid analyses, and during growth in a macrophage cell line and in mice. The authors conducted many experiments, which revealed that the rv0158 mutant has many phenotypes, but unfortunately the direct regulon of Rv0158 remains unidentified. The mechanistic basis for the attenuation of the rv0158 deletion mutant in the mouse model is therefore unexplained.The identified "redoxosome" includes genes that are functionally very diverse, and I am not convinced that the majority is required for redox homeostasis. Moreover, the authors failed to validate individual transposon mutants. The validation experiments done with the enriched mutant pool (Figure 2) are not informative as the data could be driven by just a few of the mutants. They analyzed EMSH response in a handful of transposon mutants obtained from BEI but did not carry out any further functional analysis that would provide confidence that the identified mutants are in fact required for redox homeostasis. The EMSH response is not sufficient to support this claim. It is not clear why these specific mutants were selected.The authors focused on the putative transcriptional regulator Rv0158 and used RNAseq to identify its regulon. The approach cannot distinguish genes that are directly regulated by Rv0158 from those that are indirectly regulated. This is supported by the large number of genes identified. Accordingly, the follow-up work does not lead to significant insight into the function of Rv0158. The mutant accumulated fewer mycolic acids than wt was more permeable to EtBr, and contained higher NADPH levels than wt Mtb; the underlying mechanisms and specific dysregulated genes that are responsible for these phenotypes remain undetermined. Similarly, the growth defects are difficult to interpret and the conclusion that rv0158 deletion improved growth with fatty acids as a carbon source is not supported by the data; this is only observed with propionate.The conclusion that rv0158 is required to survive in macrophages is incorrect. The mutant survives perfectly well (100% survival); it fails to grow like wt. The slight *in vivo* growth defect and more drastic persistence defect seems inconsistent with the data from the macrophage cell line.

We thank the Reviewer for insightful comments. We have addressed most of the Reviewer’s comments in the revised manuscript. Since our FACS-based strategy did not use exogenous ROS donors (e.g., H2O2 or cumene hydroperoxide) to induce oxidative stress, we expected to unbiasedly sort out mutants that directly or indirectly affect redox by influencing metabolism and respiration. Therefore, our findings are on the anticipated lines. We randomly selected 10 Tn mutants, available at the BEI resource, to validate the screen. To further bolster our validation, we analyzed the Tn mutants (ROS measurement and DNA damage) to confirm that the strains are indeed oxidatively stressed. We have included these findings in the revised manuscript (Lines: 164- 177, 263- 267 in the revised manuscript).

We selected an uncharacterized transcription factor (Rv0158) for further analysis. In the revised manuscript, we performed a detailed characterization of Rv0158. We identified the ligands for Rv1058 by MST. Using ChIP-seq, we identified binding sites for Rv0158 when *Mtb* is grown in glucose vs propionate-medium. Furthermore, we performed a metabolomics study to examine the underlying mechanism of growth improvement on propionate. Finally, we examined the survival of the mutant inside immune-activated macrophages and confirmed poor survival. The mutant showed a slow growth phenotype on glucose and mixed carbon source (glucose, glycerol, and tween-80). However, the growth of the mutant was better than WT *Mtb* on propionate. Overall, data suggest that Rv0158 deficiency accelerates propionyl-CoA utilization and reduces the pool of toxic metabolites such as glyoxylate, positively influencing the growth and bioenergetics of *rv0158 KO* utilizing propionate as a sole carbon source. We also repeated macrophages data to include more time points and showed that *rv0158 KO* entered into bacteriostasis at two days post-infection (p.i.) and showed ≈50% killing at four days p.i. The killing of the mutant coincides with an overwhelming oxidative shift in the *E_MSH_* of *Mtb* inside immune-activated macrophages. At both these time points, the survival of the mutant remains significantly lower than the WT *Mtb*.

The likely reason for attenuation inside animals requires more experimentation. However, the plausible cause could be the exceptional sensitivity of the mutant to oxidative/nitrosative stress and fitness defect on diverse carbon sources or glucose as a sole carbon source. These data are consistent with the fact that *Mtb* persistence in mice relies upon resistance to oxidative/nitrosative stress, efficient metabolization of glucose, and co-catabolism of fatty acids and glucose (Adams et al., 1997; Marrero et al., 2013; Noy et al., 2016). We have included this in the Discussion section of the revised manuscript.

The MIC measurements using resazurin are a concern, given that resazurin reports on respiration, it is reduced in aerobically growing cells. The altered intracellular redox state of the rv0158 mutant could affect the resazurin reduction and confound the data.

We thank the Reviewer for this comment. We have included CFU data to corroborate the resazurin- reduction experiment in the revised manuscript (Figure 12B in the revised manuscript). Furthermore, we confirmed that the sensitivity of *rv0158* KO to H2O2 depends upon the type of carbon source by enumerating CFU (please refer to the response to Reviewer 1). Additionally, in the absence of antibiotics, we did not see any difference in the extent of resazurin reduction to resorufin among the untreated *Mtb* strains (Figure 12—figure supplement 2a in the revised manuscript) (Lines: 512- 514 in the revised manuscript).

There are some inaccurate statements that need to be corrected. For example, the authors state "Mtb secretes antioxidant enzymes (catalase-peroxidase (KatG), superoxide dismutase (Fe-SodA), thioredoxin reductase (trxB2), Alkyl hydroperoxide reductase (AhpC) and redox buffers thioredoxin (trxC), and ergothioneine). Not all of these are secreted; in fact, the majority are not.

We thank the Reviewer for this comment. Based on the published literature, we have included a table (Table 1 in the revised manuscript) in the revised manuscript showing the localization of these systems in *Mtb* (Lines: 69- 72 in the revised manuscript).

They wrote" Mtb displays a low MIC of 3 μM for the endogenous superoxide (O2.-) generator 2, 3-Dihydro-1,4, naphthoquinone and 0.4 μM for the intracellular NO donor PA-824. On what basis do the authors quantify the MICs as "low"?

We thank the Reviewer for this comment. Our quantification is based on the fact that *Mtb* retains viability under high millimolar (mM) concentrations of exogenous ROS and RNI (Voskuil et al., 2011). However, it displays exceptional sensitivity to low μM (μM) concentrations of donors that permeate inside *Mtb* to deliver superoxide (2, 3-Dihydro-1,4, naphthoquinone) or NO (PA-824) (Tyagi et al., 2015), (Singh et al., 2008). We have rephrased the sentence to make this point clearer (Lines: 92- 99 in the revised manuscript).

The experiments shown in Figure 2 should be conducted with isolated transposon mutants.

We thank the Reviewer for this comment. We performed experiments shown in Figure 2 to validate that the oxidative *E_MSH_* displayed by the flow-sorted Tn-mutants pool correlates with other oxidative stress markers under aerobic growth conditions. TraDIS analysis confirmed that the collection of oxidized mutants contains 368 genes with significantly greater Tn insertions than the input library. While experimentally ascribing redox function to 368 genes requires independent investigation, we begin validating our data by confirming an oxidative shift in the *E_MSH_* of strains having Tn-insertions in 10 randomly selected genes that are part of the redoxosome (Figure 3B in the revised manuscript). We agree with the Reviewer that additional experiments are needed to show that these Tn mutants are oxidatively stressed. Therefore, we measured intracellular ROS and DNA damage in the Tn mutants to confirm that the strains are indeed oxidatively stressed leading to damage to biomolecules (DNA) (Lines: 263- 265 in the revised manuscript) (Figure 5: [Figure 3—figure supplement 2 in the revised manuscript]). In agreement with the oxidized *E_MSH_* of *Mtb* mutants (Figure 3B in the revised manuscript), higher eROS and DNA damage as compared to WT *Mtb* was detected.

The genes that are directly regulated by Rv0158 need to be identified to gain mechanistic insight into the many phenotypes of the mutant and potentially explain the *in vivo* attenuation.

We thank the Reviewer for this comment. We found that *rv0158* KO grew better on propionate than glucose as a sole carbon source (Figure 7 in the revised manuscript). Furthermore, Rv0158 directly binds to methylmalonyl-CoA, a crucial intermediary metabolite regulating propionate flux into methyl citrate cycle (MCC), methylmalonyl pathway, and methyl branched lipids (Savvi et al., 2008). Therefore, we propose that Rv0158 facilitates the regulation of genes associated with propionate metabolism in *Mtb*. To examine this, we performed ChIP-sequencing (ChIP-seq) using an anhydrotetracycline (ATc)-inducible flag-tagged Rv0158 (pexCF-*rv0158*) (Minch et al., 2015) (Lines: 468- 474 in the revised manuscript).

When glucose was used as a sole C-source, Rv0158 binds to the promoter regions of 70 *Mtb* genes (Supplementary file 5 in the revised manuscript; Fold change over input control > 1.5; *q value* < 0.0001). Further classification of these genes indicates that Rv0158 binds to several genes involved in methyl-branched lipid biosynthesis (*pks2*, *fadD23*, *rv2958c*, *rv2959c*, *rv2961*), lipid transporter (*rv0987-rv0988*, *mmpL1 mmpS1*, *mmpL4*-*mmpS4*), biotin synthesis (*bioF2*), and surface-associated proteins (*esp* operon, *lpqG*, *esxQ*, *rv1501*-r*v1507A*) (Supplementary file 5 in the revised manuscript) (Lines: 474- 482 in the revised manuscript). Under propionate condition, Rv0158 binds to promoter regions encompassing 32 genes (Fold change over input control > 1.5; *q value* < 0.0001). Out of 32 genes, 24 genes overlap with glucose. The promoter regions of the remaining eight genes Rv0158 exclusively bound in propionate- grown *Mtb* were part of MCC and the glyoxylate cycle (*icl1*, *fadB2*, *prpR*, *and prp* operon) (Figure 11 and Supplementary file 5 in the revised manuscript) (Lines: 483- 488 in the revised manuscript).

We performed RT-qPCR and found that the expression of MCC genes was specifically induced in WT *Mtb* grown in propionate (Figure 10, and Figure 10—figure supplement 2 in the revised manuscript). In contrast to WT *Mtb*, MCC genes were uniformly induced in *rv0158* KO independent of C source (Figure 10, and Figure 10—figure supplement 2 in the revised manuscript). However, propionate resulted in a far greater induction of MCC genes in *rv0158* KO than glucose (Figure 10 and Figure 10—figure supplement 2 in the revised manuscript). In addition to the MCC pathway, *Mtb* mitigates propionate toxicity via the biosynthesis of PDIM (phthiocerol dimycocerosates) (Lee et al., 2013) (Figure 10C in the revised manuscript). We measured the expression of the PDIM biosynthesis cluster in *rv0158* KO. Expression of PDIM genes was broadly unchanged in *rv0158* KO on glucose compared to WT *Mtb* (Figure 10 and Figure 10—figure supplement 2 in the revised manuscript). However, similar to MCC genes, the PDIM cluster was induced ≈10-fold in *rv0158* KO compared to WT *Mtb* on propionate (Figure 10, and Figure 10—figure supplement 2 in the revised manuscript). Since Rv0158 did not bind to any PDIM genes, data indicate that their hyperexpression in *rv0158 KO*, specifically on propionate, was an indirect effect (Lines: 428- 445 in the revised manuscript).

Since Rv0158 binds many genes related to methyl-branched fatty acid metabolism/transport despite growth on glucose, we think that methylmalonyl-CoA is available for binding to Rv0158 and repressing undesirable over-expression of propionate metabolic pathways. Agreeing to this, *Mtb* cultured on glucose maintains carbon flux into MCC and glyoxylate cycle, indicating the availability of propionyl-CoA and methylmalonyl-CoA (Eoh and Rhee, 2014c). Under propionate-rich conditions, increased requirements to mitigate toxicity can be achieved by Rv0158-mediated binding high levels of methylmalonyl-CoA and calibrating carbon flux into MCC by direct binding to *prpR*, *prp* operon, and *icl1,* and indirectly to production of methyl branched lipids. Lack of this control under conditions of mixed carbon sources (sugar and fatty acids) *in vivo* could have contributed to the persistence defect of *rv0158* KO in mice. The exceptional sensitivity of the mutant to oxidative and nitrosative stress is an additional factor responsible for the phenotype of the mutant *in vivo*.

The authors should test growth with additional fatty acids, especially those that are relevant to the *in vivo* environment to substantiate the claim that rv0158 is required for growth with fatty acids.

We thank the Reviewer for this recommendation. As suggested by the Reviewer, we measured the growth of WT *Mtb*, *rv0158* KO, and *rv0158* Comp in 7H9 medium supplemented with either 0.3% glucose or host-relevant fatty acids (0.3% sodium acetate, 0.3% sodium propionate, 0.02% cholesterol, 200 μM palmitate) as a sole carbon (C) source (Lines: 356- 361 in the revised manuscript). Both absorbance (OD 600 nm) and CFU estimation showed a slow growth phenotype with an extended lag phase on glucose (Figure 7A and Figure 7—figure supplement 1 in the revised manuscript). The defective growth phenotype of *rv0158* KO was absent if glucose was replaced by acetate/cholesterol/palmitate (Figure 7B- D in the revised manuscript). Interestingly, we observed a gradual increase in the growth of *rv0158* KO on 0.3% sodium propionate, a concentration bacteriostatic to both WT *Mtb* and *rv0158* Comp (Figure 7B- D in the revised manuscript).

The conclusion that the key rv0158 function is to maintain the flux of lipid metabolism intermediates needs to be substantiated by carbon flux analyses.

We agree with the Reviewer. Therefore, to elucidate metabolic changes associated with the better adaptability of *rv0158 KO* on fatty acids, we performed liquid chromatography-mass spectrometry (LC-MS/MS) metabolomics following exposure of WT *Mtb* and *rv0158 KO* to 0.3% glucose or 0.3% propionate (Lines: 400- 427 in the revised manuscript). We observed a ~100-fold accumulation of 2-methylcitrate (2-MC) in WT *Mtb* and *rv0158* KO under propionate but not glucose (Figure 9 in the revised manuscript), confirming the activation of propionate breakdown by MCC. Furthermore, downstream metabolites of propionate catabolism such as pyruvate, succinate, fumarate, malate, oxaloacetate, and glyoxylate were significantly accumulated (2-6-fold) in propionate-grown WT *Mtb* as compared to glucose (Figure 9 in the revised manuscript). In contrast, these metabolites were either not accumulated (*e.g.,* succinate, malate, and oxaloacetate) or marginally accumulated (*e.g.,* fumarate and glyoxylate) in propionate-grown *rv0158* KO (Figure 9 in the revised manuscript), presumably due to efficient catabolism of propionate via MCC in the mutant relative to WT *Mtb*. We also found evidence of lesser accumulation of pyruvate and lactate in *rv0158* KO (Figure 9 in the revised manuscript). Since the pyruvate and lactate levels depend on glyoxylate shunt, MCC, and gluconeogenesis (Serafini et al., 2019), these findings agree with the better metabolic ability of *rv0158 KO* to consume propionate. Conservation of α-ketoglutarate and glutamate pools via glutamine synthase (GltB/D’s) is an additional metabolic strategy to neutralize propionate toxicity in *Mtb* (Lee et al., 2018). We found that α-ketoglutarate pools are marginally higher in *rv0158* KO compared to WT *Mtb* under propionate (Figure 9 in the revised manuscript), indicating another reason for better mutant growth on propionate. Data suggest that *rv0158* KO mitigates propionate toxicity through efficient MCC and TCA cycle metabolization.

We also accept that only a pulse-label of a metabolite and the estimation of label incorporation in the metabolic products can fully indicate flux, since the steady-state amounts cannot separate production from consumption. However, when observed over entire pathways, or complete sets of metabolites, they do conclusively indicate the overall metabolic state in the cell and, in many cases, correlate with overall flux. However, to be correct in our description, we only indicated pool sizes of these metabolites in the text. We hope the Reviewer would agree that these findings, together with the data showing improved growth and bioenergetics of *rv0158* KO on propionate compared to glucose, support the role of Rv0158 in integrating carbon metabolism with redox balance in *Mtb*.

MIC measurements using growth/OD600 readings should be conducted.

We have supplemented our MIC data with CFU analysis. Treatment with 1X MIC of isoniazid (MIC = 0.06 μg/mL), moxifloxacin (MIC = 0.125 μg/mL), and bedaquiline (MIC = 0.03 μg/mL) decreased the survival of *rv0158* KO by >10- fold relative to WT *Mtb* (Figure 12B In the revised manuscript) (Lines: 517- 519 in the revised manuscript).

Reviewer #3 (Recommendations for the authors):The authors of the study combined the Mrx-roGFP2 redox biosensor with transposon mutagenesis and FACS to identify the Mtb redoxome. From this library, the authors identify a hypothetical protein, RV0158 that modulates intracellular redox balance and lipid homeostasis. The corresponding Mtb knockout strain is implicated in anti-TB drug susceptibility and survival *in vivo*. I am concerned that the claims by the authors are not supported by the data.The strengths of the manuscript are combining roGFP with FACS to identify a set of genes involved in redox balance, implicating a regulatory gene (rv0158) involved in this process, and demonstrating its role in anti-TB drug susceptibility and virulence.

We thank the Reviewer for this comment.

The weaknesses of the manuscript include using pooled libraries of transposon mutants (the redoxome) in their Q-PCR and oxidative stress exposure experiments. Examining pooled libraries of mutant clones complicates the interpretation of data that is based on the average contribution of all the genes and makes it impossible to make reliable conclusions.

We thank the Reviewer for this comment. We relied on pooled transposon (Tn) mutants libraries for developing our biosensor-based strategy. Therefore, individually ascribing redox function to 368 genes redoxosome is unrealistic. However, to begin validating our data, we randomly selected 10 genes that are part of the redoxosome and analyzed *E_MSH_* of *Mtb* strains harboring disruption in these genes. Consistent with the findings with the pooled libraries, the *E_MSH_* of *Mtb* mutants displayed an oxidative shift of +10 mV to +30 mV and exhibited higher eROS and DNA damage than WT *Mtb* (Figure 3B and Figure 3—figure supplement 2 in the revised manuscript). Please see response to Reviewer 2. We propose that the set of 368 genes represents a valuable resource to understand the contribution of *genetic* factors in maintaining redox balance in *Mtb.*

Also, the redoxome was identified using a single oxidant, CHP, which is not a physiologically relevant oxidant. Hence, the redoxome is an artificial signature. An important control, an antioxidant to reverse the response, is lacking.

We thank the Reviewer for this comment. We have *not* used CHP on Tn-library pools expressing Mrx1-roGFP2. CHP was only used on WT Mtb expressing Mrx1-roGFP2 to set the gates for oxidized bacteria in FACS. We clarified this in the Results section. In the revised ms, we mentioned that our strategy allowed gating and isolation of Tn-mutants that were basally oxidized without exposure to oxidants such as CHP (Lines: 164- 177 in the revised manuscript). Therefore, our redoxosome is not artificial but rather a signature of *Mtb* genes required to maintain redox balance under normal aerobic growth conditions.

As the Reviewer suggested, we have included antioxidant control in the revised manuscript. We show that the co-treatment of *E_MSH_-*oxidized Tn mutants pool with antioxidant molecules (combination of catalase (17.5 U/ml) + thiourea (1 mM) + bipyridyl (250 μM)) significantly decreased eROS levels (Figure 1—figure supplement 3b in the revised manuscript) (Lines: 191- 196 in the revised manuscript).

The manuscript lacks a mechanism whereby RV0158 regulates redox homeostasis. Although the lipid metabolism, oxidative stress, and redox couple experiments provide circumstantial evidence, a clear sensing/regulatory mechanism is lacking. The authors favor that rv0158 regulates lipid metabolism as a mechanism for redox homeostasis, but unfortunately, no mechanism is provided.

We thank the Reviewer for this comment**. In** the revised manuscript, we have performed key experiments such as ligand binding assays, ChIP-seq, qRT-PCRs, and metabolomics to understand how Rv0158 regulates growth and redox balance of *Mtb* in response to a carbon source (please see response to essential revisions and Reviewer 2, above).

Lastly, I have concerns about the bioenergetic data and interpretation. Overall, numerous casually linked experiments were performed, and the manuscript appears unfocused.

We hope that with new data, the revised manuscript alleviates several of the Reviewer’s concerns.

There are numerous grammatical issues throughout the manuscript that warrant attention.

We have performed extensive corrections to the manuscript to improve grammatical issues.

Lines 33-34; ambiguous sentences, rephrase.

We have rephrased the sentences for clarity.

Lines 56-61; add references where relevant and not at the end of the sentence.

Since several references support these studies, we have included a table (Table 1 in the revised manuscript; also, please see response to Reviewer 2, above) to match appropriate references with the relevant study.

Line 188; what do the authors mean by nontoxic? How was non-toxicity measured/assessed? This should be provided.

We thank the Reviewer for this comment**.** We do not have any data to state that treatment with 10 mM of CHP for 5 min did not lead to *Mtb*- killing. Therefore, we have rephrased the statement- we used this treatment conditions for only inducing 100% oxidation of the biosensor in *Mtb*. We have cited the appropriate reference to support the findings (Bhaskar et al., 2014; Shee et al., 2022) (Lines: 168-170 in the revised manuscript).

Line 298-301; some of the references are outdated.

We thank the Reviewer for this comment**.** We have included the recent references.

Lines 313-333; this section is unnecessary and should be moved to the supplemental data.

We thank the Reviewer for this comment**.** We have reduced this section and included new data showing the binding affinity of various ligands for Rv0158**.**

Line 423; figure 8A show delayed growth; the growth rate appears to be similar to the wild-type and complemented clones. Also, in figure 8E, the more reducing environment when cultured on fatty acid precursors compared to glucose is expected. However, only in the case of acetate (and glucose) is there a difference between the knockout and corresponding wild-type strain; propionate and cholesterol show no difference. Wise is the case and how does it relate to the role of rv0158? The authors should explain this.

We thank the Reviewer for this comment**.** The Reviewer is correct. Absorbance (OD 600 nm) and CFU estimation showed a slow growth phenotype with an extended lag phase on glucose (Figure 7—figure supplement 1 in the revised manuscript). The defective growth phenotype of *rv0158* KO was absent if glucose was replaced by acetate/cholesterol/palmitate (Figure 7B- E in the revised manuscript). Since fatty acids restored the growth of *rv0158* KO to WT *Mtb* levels, our findings showing comparable *E_MSH_* of *rv0158* KO is likely. We only see the marginal difference in acetate, which did not restore in *rv0158* Comp. We have explained why the mutant shows redox imbalance on glucose (Lines: 641- 653 in the revised manuscript).

We found that Rv0158 is a repressor of the methyl citrate cycle (MCC) and the glyoxylate cycle. The constitutive hyperexpression of *prpR, prpC*, *prpD*, *icl1*, and PDIM genes in *rv0158* KO on glucose could have directed a greater flux of glucose carbon away from energy-generating TCA cycle/oxidative phosphorylation to MCC/glyoxylate cycle and energy-requiring polyketide lipid anabolism. The expression of the MCC pathway is activated by a transcription factor PrpR in *Mtb*, whose activity depends on an oxidation-resistant 4Fe-4S cluster (Tang et al., 2019). Furthermore, PrpD retained activity under high ROS conditions in other bacteria (Grimek and Escalante-Semerena, 2004). Therefore, it is likely that MCC enzymes remained functionally active in *rv0158 KO* on glucose despite higher eROS. These metabolic changes are possible reasons behind altered redox balance, respiration slow down, and impaired growth of *rv0158* KO on glucose.

Lines 459-461; this data is not convincing as panels A and D are not baselined whereas all the other panels are. My suspicion is that if these data sets are baselined, there will be very little difference between the %OCR of the knockout and wild-type/complemented clones. Also, precisely how did the authors calculate the ~two-fold less/lower values?

We thank the Reviewer for this comment**.** We have replotted the %ECAR and %OCR in *Mtb* strains growing over glucose. All the data points are baselined to the third reading obtained just before the addition of C source (Figure 8- Source Data 1 in the revised manuscript). The approximate 2-fold change is determined by comparing the third reading after C- source addition and the first reading after CCCP addition, across groups (Lines: 2010-2014 in the revised manuscript).

Line 457; CCCP is solely an uncoupler and does not increase carbon catabolism and it does not reveal the metabolic flexibility of Mtb; it only does so in response to particular stressors.

We have clarified this point in the revised manuscript (Lines: 386- 387 in the revised manuscript). We thank the Reviewer for pointing this out.

Line 523-526; confusing sentence, rephrase.

We thank the Reviewer for this comment**.** We have removed the confusing statement.

Line 1729; correctly rearrange panels E, F, and G.

We thank the Reviewer for this comment**.** This has been corrected**.**

Line 1824; this heading is probably wrong. Respiration and not energy were measured in these experiments. Indicate that the Agilent Seahorse XFp was used to measure respiration. Also, in line 1826, extracellular acidification is not only due to glycolytic flux. This should be corrected.

We have corrected these issues (Lines: 1997- 1998 in the revised manuscript). We thank the Reviewer for pointing this out.

Line 1835; CCCP does not allow the detection of spare respiratory capacity. The authors confuse spare respiratory capacity with mammalian experiments.

We thank the Reviewer for this comment. We have corrected this issue in the manuscript (Lines: 2006- 2009 in the revised manuscript).

References:

Adams LB, Dinauer MC, Morgenstern DE, Krahenbuhl JL. 1997. Comparison of the roles of reactive oxygen and nitrogen intermediates in the host response to *Mycobacterium tuberculosis* using transgenic mice. *Tubercle and Lung Disease* 78:237–246. doi:10.1016/S0962-8479(97)90004-6

Bhaskar A, Chawla M, Mehta M, Parikh P, Chandra P, Bhave D, Kumar D, Carroll KS, Singh A. 2014. Reengineering Redox Sensitive GFP to Measure Mycothiol Redox Potential of *Mycobacterium tuberculosis* during Infection. *PLoS Pathogens* 10. doi:10.1371/journal.ppat.1003902

Braunstein M, Espinosa BJ, Chan J, Belisle JT, Jacobs WR. 2003. SecA2 functions in the secretion of superoxide dismutase A and in the virulence of *Mycobacterium tuberculosis*. *Molecular Microbiology* 48:453–464. doi:10.1046/j.1365-2958.2003.03438.x

Grimek TL, Escalante-Semerena JC. 2004. The acnD Genes of Shewenella oneidensis and *Vibrio cholerae* Encode a New Fe/S-Dependent 2-Methylcitrate Dehydratase Enzyme That Requires prpF Function *in vivo*. *Journal of Bacteriology* 186:454–462. doi:10.1128/JB.186.2.454-462.2004

Ke N, Landeta C, Wang X, Boyd D, Eser M, Beckwith J. 2018. Identification of the Thioredoxin Partner of Vitamin K Epoxide Reductase in Mycobacterial Disulfide Bond Formation. *Journal of Bacteriology* 200. doi:10.1128/JB.00137-18

Marrero J, Trujillo C, Rhee KY, Ehrt S. 2013. Glucose Phosphorylation Is Required for *Mycobacterium tuberculosis* Persistence in Mice. *PLoS Pathogens* 9. doi:10.1371/journal.ppat.1003116

Nieto LMR, Mehaffy C, Creissen E, Troudt JL, Troy A, Bielefeldt-Ohmann H, Burgos M, Izzo A, Dobos KM. 2016. Virulence of *Mycobacterium tuberculosis* after acquisition of Isoniazid resistance: Individual nature of katG mutants and the possible role of AhpC. *PLoS ONE* 11:e0166807. doi:10.1371/journal.pone.0166807

Noy T, Vergnolle O, Hartman TE, Rhee KY, Jacobs WR, Berney M, Blanchard JS. 2016. Central role of pyruvate kinase in carbon co-catabolism of *Mycobacterium tuberculosis*. *Journal of Biological Chemistry* 291:7060–7069. doi:10.1074/jbc.M115.707430

Sao Emani C, Williams MJ, Wiid IJ, Hiten NF, Viljoen AJ, Pietersen R-DD, van Helden PD, Baker B. 2013. Ergothioneine Is a Secreted Antioxidant in Mycobacterium smegmatis. *Antimicrobial Agents and Chemotherapy* 57:3202–3207. doi:10.1128/AAC.02572-12

Savvi S, Warner DF, Kana BD, McKinney JD, Mizrahi V, Dawes SS. 2008. Functional Characterization of a Vitamin B 12 -Dependent Methylmalonyl Pathway in *Mycobacterium tuberculosis* : Implications for Propionate Metabolism during Growth on Fatty Acids. *Journal of Bacteriology* 190:3886–3895. doi:10.1128/JB.01767-07

Shee S, Singh S, Tripathi A, Thakur C, Kumar AT, Das M, Yadav V, Kohli S, Rajmani RS, Chandra N, Chakrapani H, Drlica K, Singh A. 2022. Moxifloxacin-Mediated Killing of *Mycobacterium tuberculosis* Involves Respiratory Downshift, Reductive Stress, and Accumulation of Reactive Oxygen Species. *Antimicrobial Agents and Chemotherapy* 66. doi:10.1128/aac.00592-22

Tucci P, Portela M, Chetto CR, González-Sapienza G, Marín M. 2020. Integrative proteomic and glycoproteomic profiling of *Mycobacterium tuberculosis* culture filtrate. *PLoS ONE* 15:e0221837. doi:10.1371/journal.pone.0221837

Vargas-Romero F, Guitierrez-Najera N, Mendoza-Hernández G, Ortega-Bernal D, Hernández-Pando R, Castañón-Arreola M. 2016. Secretome profile analysis of hypervirulent *Mycobacterium tuberculosis* CPT31 reveals increased production of EsxB and proteins involved in adaptation to intracellular lifestyle. *Pathogens and Disease* 74:ftv127. doi:10.1093/femspd/ftv127

Wong D, Li W, Chao JD, Zhou P, Narula G, Tsui C, Ko M, Xie J, Martinez-Frailes C, Av-Gay Y. 2018. Protein tyrosine kinase, PtkA, is required for *Mycobacterium tuberculosis* growth in macrophages. *Scientific Reports* 8:1–12. doi:10.1038/s41598-017-18547-9

[Editors' note: further revisions were suggested prior to acceptance, as described below.]

The manuscript has been improved but there are some remaining issues that need to be addressed, as outlined below:The authors have added a tremendous amount of work to the manuscript and strengthened their dataset. They have addressed many prior concerns to yield an impressive story, but there are some concerns with the newly added data that need to be addressed. In addition, the authors still seem to bias their interpretations towards their favored model and should revise the manuscript to consider alternatives.The authors should address the following comments, which they should largely be able to do by not overstating their results, considering other models, and providing a more thorough and transparent analysis of the metabolite measurements.1. The findings that the rv0158 deletion mutant can grow on propionate at a concentration that inhibits growth of the wt and complemented strains and that Rv0158 can bind methylmalonyl-CoA are interesting. However, the authors' interpretations are often based on phenotypic differences whose physiological relevance remains unclear. Their interpretation of the phenotypes seems biased towards the model they propose, but other explanations should be considered and included, at least in the discussion.

We agree with the comment. Our data suggest that Rv0158 likely senses methylmalonyl-CoA and regulates the expression of pathways associated with propionate metabolism. However, several issues remain to be addressed. We need to investigate if methylmalonyl-CoA is perceived as a metabolic signal by Rv0158 in *Mtb* and how this association affects direct interaction with DNA and gene expression. Also, the lack of Rv0158 may induce a physiological state in *Mtb that is* more tolerant to toxicity caused by fatty acids such as propionate. For example, cAMP homeostasis is associated with modulating redox balance, respiration, fatty acid uptake, and fatty acid toxicity in *mycobacteria* (Ko and Oh, 2020; Nambi et al., 2013; Nazarova et al., 2019; Wong et al., 2023)*.* Moreover, *Mycobacterium bovis* BCG elevates cAMP levels in response to propionate, and phosphodiesterases such as Rv0805 regulate cAMP homeostasis to modulate growth on propionate (McDowell et al., 2023). Other evidence suggests that reduced cAMP levels result in overactive fatty acid catabolism (Nambi et al., 2013; Nazarova et al., 2019) as we observed in the case of *rv0158 KO* growing on propionate. While overactive fatty acid catabolism is generally detrimental due to the accumulation of toxic intermediates, mycobacteria have evolved sophisticated regulatory mechanisms (e.g., FdmR) to direct the flux of excess fatty acids away from catabolism to lipid anabolism (Dong et al., 2021). These alterations on the mycobacterial surface might be important to counteract toxicity caused by fatty acid-dependent disruption of *the Mtb* envelope (Kengmo Tchoupa et al., 2022). We observed a higher expression of the PDIM biosynthesis cluster, altered mycolic acids content, and changes in membrane permeability of *rv0158* KO, suggesting overall changes in the cell envelope of the mutant. Therefore, the physiological effects of *rv0158* disruption in *Mtb* could result from multiple effectors such as redox, respiration, and cAMP. Also, propionate metabolism is regulated by multiple transcription factors (*e.g.,* GlnR, PrpR, RamB, PhoP, WhiB3) in *Mtb* (Lee et al., 2013; Micklinghoff et al., 2009; Qi et al., 2021; Singh et al., 2009; Tang et al., 2019). Therefore, future experiments are needed to understand the breadth of Rv0158-mediated gene regulation in collaboration with additional regulators and metabolites (*e.g.,* acyl-CoA, malonyl-CoA, and cAMP) in coordinating fatty acid metabolism. In the revised manuscript, we have included the above explanations in the Discussion section of the revised manuscript (Lines: 737- 767 in the revised manuscript). We thank the editors for these insightful comments.

2. There are concerns regarding the metabolomics data. The authors do not seem to have used metabolite standards. Even the source data fail to show which masses they utilized for the measured metabolites. Oxaloacetate is very unstable and can typically not be measured directly.

We apologize for not adequately describing the metabolomics data. For measurement of the TCA cycle and glyoxylate cycle intermediates, metabolites were derivatized with O-benzylhydroxylamine (OBHA) after extraction and drying down, as mentioned in earlier method papers (Tan et al., 2014; Walvekar et al., 2018). In particular, oxaloacetate is unstable in aqueous solutions, hence was stabilized by OBHA derivatization to prevent degradation (Figure 9- Source Data 1, Figure 9—figure supplement 1- 2 in the revised manuscript) (also see Walvekar et al., 2018 for extensive methodology details). We have used metabolites standards to identify and quantify the TCA and methylcitrate cycle metabolites (please see the source data- Figure 9- Source Data 1). The metabolites were separated using Synergi 4µm Fusion-RP 80 Å LC column (150 x 4.6 612 mm, Phenomenex) on Shimadzu Nexera UHPLC system. Solvents used for the separation of TCA intermediates are the following: 0.1% formic acid in water (solvent A) and 0.1% formic acid in methanol (solvent B). The metabolites were detected using ABSciex QTRAP 5500 mass spectrometer. The detailed parameters used are described in Walvekar et al., 2018. The data was acquired using Analyst 1.6.2 software (Sciex) and analyzed using MultiQuant version 3.0.1 (Sciex). We have incorporated this information in the results (Lines: 412- 419 in the revised manuscript) and methods sections (Lines: 1344- 1355 in the revised manuscript) of the revised manuscript.

The statement "We found that α-ketoglutarate pools are marginally higher in rv0158 KO compared to WT Mtb under propionate (Figure 9; [Figure 9 in the revised manuscript]), indicating another reason for better mutant growth on propionate" is not justified, because in the complemented mutant α-ketoglutarate pools are even higher than that in the KO.

This was an inadvertent error from our side. We removed the statement from the submitted revised manuscript (1-05-2022-RA-*eLife*-80218R1) but did not delete it from the corresponding response letter. This result is not a part of the revised manuscript. We apologize for the confusion.

3. The authors added CFU data to assess susceptibility to antibiotics, but this is different than MIC measurements using growth as readout to assess drug resistance, which can be different than susceptibility to drug induced killing. This should be considered when describing and interpreting the data.

The reason for doing the CFU experiment was to rule out the issue with the MIC assays done using resazurin dye (raised by Reviewer #2). However, in light of the above comment, we have modified the Results section to clearly differentiate between resistance and drug induced killing (Lines- 528- 539 in the revised manuscript).

4. The authors' explanation for the different binding targets of Rv0158 during growth in glucose and in propionate is not clear. It seems that they suggest that methylmalony -CoA binding is required for Rv1058 to bind and repress the prpR promoter, but direct evidence for this is lacking. Overall, their model of Rv0158 activity is very complex and is sometimes based solely on associations. The authors would benefit by more rigorously evaluating their data and being more careful with their interpretations.

We agree that the model of Rv0158 mediated gene regulation is very complex and needs independent investigation. To the extent possible, we have rigorously analysed and interpreted the data. We reanalyzed the ChIP-seq data and made further revision in our interpretations. Based on all the sequences enriched in the ChIP analysis under glucose and propionate conditions, the conserved binding motif for Rv0158 was found to be YGGCGGBGMCGGCGG (where Y represents pyrimidines (C/T), B represents G/T/C, and M represents A/C, enrichment e value = 2.6*10^-109^) (Figure 11D in the revised manuscript). While the binding motif is unique, it is closely located to the binding motifs of other transcription factors involved in regulating methylcitrate cycle (*e.g.,* PrpR, GlnR, PhoP, and RamB) (Supplementary file 6 in the revised manuscript), suggesting overlapping or competing mechanisms to regulate propionate metabolism in *Mtb*. We also made it clear that the direct or indirect nature of Rv0158-mdiated regulation needs future experimentation (Lines- 508- 515 in the revised manuscript).

Additionally, in the Discussion section we made it evident that there several alternate models to explain the findings associated with Rv0158 including the need to understand the breadth of Rv0158-mediated gene regulation in collaboration with additional regulators (GlnR, PrpR, RamB, PhoP, WhiB3) and metabolites (*e.g.,* acyl-CoA, malonyl-CoA, and cAMP) in coordinating fatty acid metabolism (Lines- 743- 767 in the revised manuscript).

References

Dong W, Nie X, Zhu H, Liu Q, Shi K, You L, Zhang Y, Fan H, Yan B, Niu C, Lyu LD, Zhao GP, Yang C. 2021. Mycobacterial fatty acid catabolism is repressed by FdmR to sustain lipogenesis and virulence. *Proceedings of the National Academy of Sciences of the United States of America* 118. doi:10.1073/pnas.2019305118

Kengmo Tchoupa A, Eijkelkamp BA, Peschel A. 2022. Bacterial adaptation strategies to host-derived fatty acids. *Trends in Microbiology* 30:241–253. doi:10.1016/j.tim.2021.06.002

Ko E-M, Oh J-I. 2020. Induction of the cydAB Operon Encoding the bd Quinol Oxidase Under Respiration-Inhibitory Conditions by the Major cAMP Receptor Protein MSMEG_6189 in Mycobacterium smegmatis. *Frontiers in Microbiology* 11:608624. doi:10.3389/fmicb.2020.608624

Lee W, VanderVen BC, Fahey RJ, Russell DG. 2013. Intracellular *Mycobacterium tuberculosis* exploits host-derived fatty acids to limit metabolic stress. *Journal of Biological Chemistry* 288:6788–6800. doi:10.1074/jbc.M112.445056

McDowell JR, Bai G, Lasek‐Nesselquist E, Eisele LE, Wu Y, Hurteau G, Johnson R, Bai Y, Chen Y, Chan J, McDonough KA. 2023. Mycobacterial phosphodiesterase Rv0805 is a virulence determinant and its cyclic nucleotide hydrolytic activity is required for propionate detoxification. *Molecular Microbiology* 119:401–422. doi:10.1111/mmi.15030

Micklinghoff JC, Breitinger KJ, Schmidt M, Geffers R, Eikmanns BJ, Bange FC. 2009. Role of the transcriptional regulator RamB (Rv0465c) in the control of the glyoxylate cycle in *Mycobacterium tuberculosis*. *Journal of Bacteriology* 191:7260–7269. doi:10.1128/JB.01009-09

Nambi S, Gupta K, Bhattacharyya M, Ramakrishnan P, Ravikumar V, Siddiqui N, Thomas AT, Visweswariah SS. 2013. Cyclic AMP-dependent protein lysine acylation in mycobacteria regulates fatty acid and propionate metabolism. *Journal of Biological Chemistry* 288:14114–14124. doi:10.1074/jbc.M113.463992

Nazarova E V., Montague CR, Huang L, La T, Russell D, Vanderven BC. 2019. The genetic requirements of fatty acid import by *Mycobacterium tuberculosis* within macrophages. *eLife* 8:1–12. doi:10.7554/*eLife*.43621

Qi N, She GL, Du W, Ye BC. 2021. Mycobacterium smegmatis GlnR Regulates the Glyoxylate Cycle and the Methylcitrate Cycle on Fatty Acid Metabolism by Repressing icl Transcription. *Frontiers in Microbiology* 12:103. doi:10.3389/fmicb.2021.603835

Singh A, Crossman DK, Mai D, Guidry L, Voskuil MI, Renfrow MB, Steyn AJC. 2009. *Mycobacterium tuberculosis* WhiB3 Maintains Redox Homeostasis by Regulating Virulence Lipid Anabolism to Modulate Macrophage Response. *PLoS Pathogens* 5:e1000545. doi:10.1371/journal.ppat.1000545

Tan B, Lu Z, Dong S, Zhao G, Kuo M-S. 2014. Derivatization of the tricarboxylic acid intermediates with O-benzylhydroxylamine for liquid chromatography–tandem mass spectrometry detection. *Analytical Biochemistry* 465:134–147. doi:10.1016/j.ab.2014.07.027

Tang S, Hicks ND, Cheng Y-S, Silva A, Fortune SM, Sacchettini JC. 2019. Structural and functional insight into the *Mycobacterium tuberculosis* protein PrpR reveals a novel type of transcription factor. *Nucleic acids research* 47:9934–9949. doi:10.1093/nar/gkz724

Walvekar A, Rashida Z, Maddali H, Laxman S. 2018. A versatile LC-MS/MS approach for comprehensive, quantitative analysis of central metabolic pathways. *Wellcome open research* 3:122. doi:10.12688/wellcomeopenres.14832.1

Wong AI, Beites T, Planck KA, Fieweger RA, Eckartt KA, Li S, Poulton NC, VanderVen BC, Rhee KY, Schnappinger D, Ehrt S, Rock J. 2023. Cyclic AMP is a critical mediator of intrinsic drug resistance and fatty acid metabolism in *M. tuberculosis*. *eLife* 12. doi:10.7554/*eLife*.81177